# In-Context Learning with Transformers: Softmax Attention Adapts to Function Lipschitzness

**Liam Collins**[*]
Chandra Family Department of ECE
The University of Texas at Austin
liamc@utexas.edu

**Advait Parulekar**[*]
Chandra Family Department of ECE
The University of Texas at Austin
advaitp@utexas.edu

**Aryan Mokhtari**
Chandra Family Department of ECE
The University of Texas at Austin
mokhtari@austin.utexas.edu

**Sujay Sanghavi**
Chandra Family Department of ECE
The University of Texas at Austin
sanghavi@mail.utexas.edu

**Sanjay Shakkottai**
Chandra Family Department of ECE
The University of Texas at Austin
sanjay.shakkottai@utexas.edu

## Abstract

A striking property of transformers is their ability to perform in-context learning (ICL), a machine learning framework in which the learner is presented with a novel context during inference implicitly through some data, and tasked with making a prediction in that context. As such, that learner must adapt to the context without additional training. We explore the role of *softmax* attention in an ICL setting where each context encodes a regression task. We show that an attention unit learns a window that it uses to implement a nearest-neighbors predictor adapted to the landscape of the pretraining tasks. Specifically, we show that this window widens with decreasing Lipschitzness and increasing label noise in the pretraining tasks. We also show that on low-rank, linear problems, the attention unit learns to project onto the appropriate subspace before inference. Further, we show that this adaptivity relies crucially on the softmax activation and thus cannot be replicated by the linear activation often studied in prior theoretical analyses.

## 1 Introduction

One of the most compelling behaviors of pretrained transformers is their ability to perform *in-context learning (ICL)* [1]: determining how to solve an unseen task simply by making a forward pass on input context tokens. Arguably the most critical innovation enabling ICL is the self-attention mechanism [2], which maps each token in an input sequence to a new token using information from all other tokens. A key design choice in this self-attention architecture is of the activation function that controls how much "attention" a token pays to other tokens. *Softmax*-activated self-attention (i.e. softmax attention) is most commonly, and successfully, used in practice [1, 3–6].

A natural approach to explain ICL adopted by the literature is to equate it with classical machine learning algorithms, primarily variants of gradient descent (GD). Several works have shown that

---

[*]Co-first authors, listed in alphabetical order.

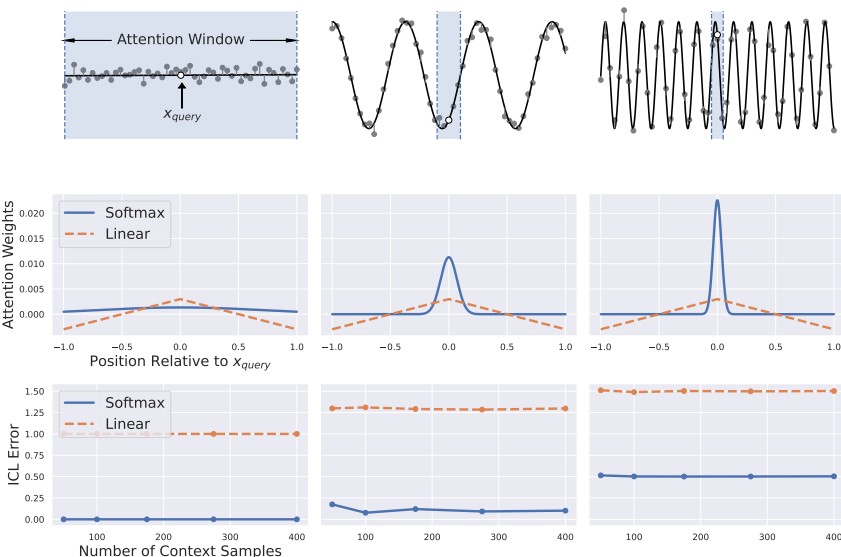

Figure 1: **Top Row:** The black line denotes the target function over a domain (horizontal axis). The gray dots are noisy training data, and the white dot is a query. From left to right, the Lipschitzness of the target function grows and the optimal softmax attention window (shaded blue) shrinks. **Middle Row:** Attention weights – which determine the attention window – as a function of the relative position from the query for softmax and linear attention. The softmax weights adjust to the Lipschitzness. **Bottom Row:** ICL error versus number of context samples for the three settings. **Adapting to function Lipschitzness leads softmax attention to achieve small error**. Please see Remark 2.1 and Appendix J for further discussion and details.

when the ICL tasks are *linear* regressions and the activation in the attention unit is identity (referred to as *linear* attention), transformers that implement preconditioned GD during ICL are global optima of the pretraining loss, which is the population loss on ICL tasks [7–9]. In particular, the prediction of such transformers with $l$ linear attention layers equals the prediction of a regressor trained with $l$ preconditioned GD steps on the context examples. However, since these analyses are limited to linear attention and tasks, they do not explain the widespread success of *softmax* attention at ICL.

More recent work [10] extends these results by showing that for general regression tasks and any attention activation that is a kernel, ICL equates to training a kernel regressor via functional GD in the Reproducing Kernel Hilbert Space (RKHS) induced by the activation. However, this functional GD yields generalization guarantees only when the activation kernel is *identical* to a kernel that generates the labels, which does not apply to the softmax activation, as it is not a kernel. Further, like the aforementioned studies of the linear setting [7–9], this analysis only shows that pretraining leads to learning the *covariate* distribution, while the activation implicitly encodes the *label* distribution needed for accurate predictions. Thus, this line of work has not explained the very fundamental question of what *softmax* attention learns during pretraining that enables it to perform ICL on a wide variety of downstream tasks. Motivated by this gap in the literature, we ask the following question.

*How does **softmax attention** learn to perform ICL?*

To answer this question, we study general settings in which pretraining and evaluation ICL tasks are regressions that share only *Lipschitzness* and *label noise variance*. Specifically, the rate at which their ground-truth labels change along particular directions in the input space, and the variance in the label noise, is similar across tasks. In such settings, we observe that softmax attention acts as a nearest neighbors regressor with an *attention window* – i.e. neighborhood of points around the query that strongly influence, or "attend to", the prediction – that adapts to the pretraining tasks. Specifically, our main result is as follows:

> **Main Claim:** Softmax attention performs ICL by calibrating its *attention window* to the *Lipschitzness* and *label noise variance* of the pretraining tasks.

While this does not contradict the line of work showing that ICL manifests via a "meta-learned" gradient-based algorithm, we show in a general setting that a simpler mechanism can explain the capabilities of a widely accepted model of ICL.

**Outline.** We substantiate the above claim via two streams of analysis. To our knowledge, these are the first results showing that softmax attention pretrained on ICL tasks recovers shared structure among the tasks that facilitates ICL on downstream tasks.

**(1) Attention window *scale* adapts to Lipschitzness and noise variance – Section 3.** We prove that the pretraining-optimal softmax attention estimator scales its attention window *inversely with the task Lipschitzness* and *jointly with the noise level* to optimally trade-off bias and variance in its prediction (Theorem 3.4). This requires tight upper and lower bounds on the pretraining ICL loss. While the upper bounds (Lemma C.8) hold for all $L$-Lipschitz tasks, the lower bounds (Lemma C.9) are more challenging and require considering specific classes of tasks. We consider two classes of generalized linear models (GLMs), and obtain lower bounds via novel concentrations for particular functionals on the distribution of the attention weights for tokens distributed on the hypersphere (Corollary G.5).

**(2) Attention window *directions* adapt to direction-wise Lipschitzness – Section 4.** We prove that when the target function class consists of linear functions that share a common low-dimensional structure, the optimal softmax attention weight matrix from pretraining projects the data onto this subspace (Theorem 4.4). In other words, softmax attention learns to zero-out the zero-Lipschitzness directions in the ambient data space, and thereby reduces the effective dimension of ICL. We prove this via a careful symmetry-based argument to characterize a particular gradient of the ICL loss as positive (Lemmas H.3 and H.4).

**Tightness of results.** Our results highlight the importance of shared Lipschitzness across training and test, as well as the critical role of the softmax activation, to ICL. We show that softmax attention pretrained on the setting from Section 3 in-context learns *any* downstream task with *similar Lipschitzness* to the pretraining tasks, while changing *only the Lipschitzness* of the evaluation tasks degrades performance (Theorem 3.5) – implying *learning Lipschitzness is both sufficient* and *necessary for generalization*. Further, to emphasize the *necessity of the softmax*, we show that the minimum ICL loss achievable by linear attention exceeds that achieved by pretrained softmax attention (Theorem 3.6). We verify all of these results with empirical simulations (Section 3.2 and Appendix J).

**Notations.** We use (upper-, lower-)case boldface for (matrices, vectors), respectively. We denote the (identity, zero) matrix in $\mathbb{R}^{d \times d}$ as $(\mathbf{I}_d, \mathbf{0}_{d \times d})$, respectively, the set of column-orthonormal matrices in $\mathbb{R}^{d \times k}$ as $\mathbb{O}^{d \times k}$, and the (column space, 2-norm) of a matrix $\mathbf{B}$ as $(\text{col}(\mathbf{B}), \|\mathbf{B}\|)$, respectively. We indicate the unit hypersphere in $\mathbb{R}^d$ by $\mathbb{S}^{d-1}$ and the uniform distribution over $\mathbb{S}^{d-1}$ as $\mathcal{U}^d$. We use asymptotic notation $(\mathcal{O}, \Omega)$ to hide constants that depend only on the dimension $d$.

## 1.1 Additional Related Work

Numerous recent works have *constructed* transformers that can implement GD and other machine learning algorithms during ICL [11–15], but it is unclear whether *pretraining* leads to such transformers. [16] and [13] provide generalization bounds for ICL via tools from algorithmic stability and uniform concentration, respectively. [17] investigate the pretraining statistical complexity of learning a Bayes-optimal predictor for ICL on linear tasks with linear attention. [18–20] study the role of the pretraining data distribution, rather than the learning model, in facilitating ICL. [21] studies the dynamics of a softmax attention unit trained with GD on ICL tasks, but this analysis considers only linear tasks and orthogonal inputs. The connection between ICL with softmax attention and non-parametric regression has been noticed by other works that analyze the ICL performance of a softmax-like kernel regressor [22] and aim to improve upon softmax attention [23–27] rather than explain what it learns during pretraining. Please see Appendix A for further discussion of the large body of related works studying the theory of transformers, ICL and kernel regression.

## 2 Preliminaries

**In-Context Learning (ICL) regression tasks.** We study ICL in the regression setting popularized by [28], wherein each task is a regression problem in $\mathbb{R}^d$. The context for task $t$ consists of a set of $n$ feature vectors paired with noisy labels $\{\boldsymbol{x}_i^{(t)}, f^{(t)}(\boldsymbol{x}_i^{(t)}) + \epsilon_i^{(t)}\}_{i=1}^n$, where $f^{(t)} : \mathbb{R}^d \to \mathbb{R}$ generates the ground-truth labels for task $t$ and $\epsilon_i^{(t)}$ is label noise. Given this context, the model solves the task if it accurately predicts the label of a query $\boldsymbol{x}_{n+1}^{(t)}$. During pretraining, the model

observes many such tasks. Then, it is evaluated on a new task with context $\{\boldsymbol{x}_i, f^{(*)}(\boldsymbol{x}_i^{(*)}) + \epsilon_i^{(*)}\}_{i=1}^n$ and query $\boldsymbol{x}_{n+1}^{(*)}$. We emphasize that the model is trained only on the pretraining tasks, not the evaluation context. Unlike traditional supervised learning, which would involve training on the context $\{\boldsymbol{x}_i, f^{(*)}(\boldsymbol{x}_i^{(*)}) + \epsilon_i^{(*)}\}_{i=1}^n$ in order to predict $f^{(*)}(\boldsymbol{x}_{n+1}^{(*)})$, ICL happens *entirely in a forward pass*, so there is no training using labels from $f^{(*)}$. Our inquiry focuses on how ICL is facilitated by the softmax activation in the self-attention unit, which we introduce next.

**The Softmax Attention Unit.** We consider a single softmax attention head $H_{SA}(\cdot; \boldsymbol{\theta})$ : $\mathbb{R}^{(d+1)\times(n+1)} \to \mathbb{R}^{(d+1)\times(n+1)}$ parameterized by $\boldsymbol{\theta} := (\mathbf{W}_K, \mathbf{W}_Q, \mathbf{W}_V)$, where $\mathbf{W}_K, \mathbf{W}_Q, \mathbf{W}_V \in \mathbb{R}^{(d+1)\times(d+1)}$ are known as key, query, and value weight matrices, respectively. Intuitively, for a sequence of tokens $\mathbf{Z} = [\boldsymbol{z}_1, \ldots, \boldsymbol{z}_{n+1}] \in \boldsymbol{z}^{(d+1)\times(n+1)}$, the attention layer creates a "hash map" where the key-value pairs come from key and value embeddings of the input tokens, $\{\mathbf{W}_K \boldsymbol{z}_i : \mathbf{W}_V \boldsymbol{z}_i\}$. Each token $\boldsymbol{z}_i$ is interpreted as a query $\mathbf{W}_Q \boldsymbol{z}_i$, and during a pass through the attention layer, this query is matched with the keys $\{\mathbf{W}_K \boldsymbol{z}_j\}_j$ to return an average over the associated values $\{\mathbf{W}_V \boldsymbol{z}_j\}_j$ with a weight determined by the quality of the match (proportional to $e^{(\mathbf{W}_K \boldsymbol{z}_j)^\top (\mathbf{W}_Q \boldsymbol{z}_i)}$). Specifically, $H_{SA}(\mathbf{Z}; \boldsymbol{\theta}) = [h_{SA}(\mathbf{z}_1, \mathbf{Z}; \boldsymbol{\theta}), \cdots, h_{SA}(\mathbf{z}_{n+1}, \mathbf{Z}; \boldsymbol{\theta})]$, where

$$h_{SA}(\mathbf{z}_i, \mathbf{Z}; \boldsymbol{\theta}) = \frac{\sum_{j=1}^n (\mathbf{W}_V \mathbf{z}_j) \; e^{(\mathbf{W}_K \mathbf{z}_j)^\top (\mathbf{W}_Q \mathbf{z}_i)}}{\sum_{i=1}^n e^{(\mathbf{W}_K \mathbf{z}_j)^\top (\mathbf{W}_Q \mathbf{z}_i)}} \in \mathbb{R}^{d+1}. \tag{ATTN}$$

With slight abuse of notation, we denote $h_{SA}(\mathbf{z}_j) = h_{SA}(\mathbf{z}_j, \mathbf{Z}; \boldsymbol{\theta})$ when it is not ambiguous. To study how this architecture enables ICL, we follow [28] to formalize ICL as a regression problem. Below we define the tokenization, pretraining objective and evaluation task.

**Tokenization for regression.** The learning model encounters token sequences of the form

$$\mathbf{Z} := \begin{bmatrix} \boldsymbol{x}_1 & \boldsymbol{x}_2 & \ldots & \boldsymbol{x}_n & \boldsymbol{x}_{n+1} \\ f(\boldsymbol{x}_1) + \epsilon_1 & f(\boldsymbol{x}_2) + \epsilon_1 & \ldots & f(\boldsymbol{x}_n) + \epsilon_n & 0 \end{bmatrix} \in \mathbb{R}^{(d+1)\times(n+1)}, \tag{1}$$

where the ground-truth labelling function $f$ maps from $\mathbb{R}^d$ to $\mathbb{R}$ and belongs to some class $\mathcal{F}$, each $\epsilon_i$ is mean-zero noise, and the $i$-th input feature vector $\boldsymbol{x}_i \in \mathbb{R}^d$ is jointly embedded in the same token with its noisy label $f(\boldsymbol{x}_i) + \epsilon_i \in \mathbb{R}$. We denote this token $\boldsymbol{z}_i$. The ICL task is to accurately predict this label given the $n$ context tokens $\{(\boldsymbol{x}_i, f(\boldsymbol{x}_i) + \epsilon_i)\}_{i=1}^n$, where $f$ may vary across sequences. The prediction for the label of the $(n+1)$-th feature vector is the $(d+1)$-th element of $h_{SA}(\mathbf{z}_{n+1})$ [10], denoted $h_{SA}(\mathbf{z}_{n+1})_{d+1}$. Ultimately, the goal is to learn weight matrices such that $h_{SA}(\mathbf{z}_{n+1})_{d+1}$ is likely to approximate the $(n + 1)$-th label on a random sequence $\mathbf{Z}$.

**Pretraining protocol.** We study what softmax attention learns when its weight matrices are *pretrained* using sequences of the form of (1). These sequences are randomly generated as follows:

$$f \sim D(\mathcal{F}), \quad \boldsymbol{x}_1, \ldots, \boldsymbol{x}_{n+1} \overset{\text{i.i.d.}}{\sim} D_{\boldsymbol{x}}^{\otimes(n+1)}, \quad \epsilon_1, \ldots, \epsilon_n \overset{\text{i.i.d.}}{\sim} D_\epsilon^{\otimes(n+1)} \tag{2}$$

where $D(\mathcal{F})$ is a distribution over functions in $\mathcal{F}$, $D_{\boldsymbol{x}}$ is a distribution over $\mathbb{R}^d$, and $D_\epsilon$ is a distribution over $\mathbb{R}$ with mean zero and variance $\sigma^2$. The token embedding sequence $\mathbf{Z}$ is then constructed as in (1). Given this generative model, the pretraining loss of the parameters $\boldsymbol{\theta} = (\mathbf{W}_Q, \mathbf{W}_K, \mathbf{W}_V)$ is the expected squared difference between the prediction of softmax attention and the ground-truth label of the $(n+1)$-th input feature vector in each sequence, namely

$$\bar{\mathcal{L}}(\boldsymbol{\theta}) := \mathbb{E}_{f, \{\boldsymbol{x}_i\}_i, \{\epsilon_i\}_i} (h_{SA}(\mathbf{z}_{n+1})_{d+1} - f(\boldsymbol{x}_{n+1}))^2. \tag{3}$$

We next reparameterize the attention weights to make (3) more interpretable. For the last column of $\mathbf{W}_V$, we show in Appendix B that any minimizer of (3) in the settings we consider must have the first $d$ elements of this last column equal to zero. We follow [7, 9, 10] by setting the first $n$ columns of $\mathbf{W}_V$ to zero. As in [10], we fix the $(d+1, d+1)$-th element of $\mathbf{W}_V$, here as 1 for simplicity. In the same vein, we follow [7, 10] by setting the $(d+1)$-th row and column of $\mathbf{W}_K$ and $\mathbf{W}_Q$ equal to zero. To summarize, the reparameterized weights are:

$$\mathbf{W}_V = \begin{bmatrix} \mathbf{0}_{d\times d} & \mathbf{0}_{d\times 1} \\ \mathbf{0}_{1\times d} & 1 \end{bmatrix}, \quad \mathbf{W}_K = \begin{bmatrix} \mathbf{M}_K & \mathbf{0}_{d\times 1} \\ \mathbf{0}_{1\times d} & 0 \end{bmatrix}, \quad \mathbf{W}_Q = \begin{bmatrix} \mathbf{M}_Q & \mathbf{0}_{d\times 1} \\ \mathbf{0}_{1\times d} & 0 \end{bmatrix} \tag{4}$$

where $\mathbf{M}_K, \mathbf{M}_Q \in \mathbb{R}^{d \times d}$. Now, since our goal is to reveal properties of minimizers of the pretraining loss, rather than study the dynamics of optimizing the loss, without loss of generality we can define $\mathbf{M} := \mathbf{M}_K^\top \mathbf{M}_Q$ and re-define the pretraining loss (3) as a function of $\mathbf{M}$. Doing so yields:

$$\mathcal{L}(\mathbf{M}) := \mathbb{E}_{f, \{\boldsymbol{x}_i\}_i, \{\epsilon_i\}_i} \left( \frac{\sum_{i=1}^n (f(\boldsymbol{x}_i) + \epsilon_i)\, e^{\boldsymbol{x}_i^\top \mathbf{M} \boldsymbol{x}_{n+1}}}{\sum_{i=1}^n e^{\boldsymbol{x}_i^\top \mathbf{M} \boldsymbol{x}_{n+1}}} - f(\boldsymbol{x}_{n+1}) \right)^2. \qquad \text{(ICL)}$$

**Interpretation of the pretraining loss.** The loss (ICL) clarifies how softmax attention can be interpreted as a nearest neighbors regressor. When $\boldsymbol{x}_i^\top \mathbf{M} \boldsymbol{x}_{n+1}$ is a proxy for the distance between $\boldsymbol{x}_i$ and $\boldsymbol{x}_{n+1}$ (which we formally show in Section 3 as happening under reasonable assumptions), the softmax attention prediction is a convex combination of the noisy labels with weights determined by the closeness of $\boldsymbol{x}_i$ to $\boldsymbol{x}_{n+1}$, such that the labels of points closer to $\boldsymbol{x}_{n+1}$ have larger weight. Moreover, the decay in weights on points further from $\boldsymbol{x}_{n+1}$ is exponential and controlled by $\mathbf{M}$, which effectively defines a neighborhood, or attention window, of points around $\boldsymbol{x}_{n+1}$ whose labels have non-trivial weight. More formally, we can think of the attention window defined for a query $\boldsymbol{x}_{n+1}$ as the set $\texttt{AttnWindow}(\boldsymbol{x}_{n+1}; \mathbf{M}) := \{\boldsymbol{x} : \boldsymbol{x}^\top \mathbf{M} \boldsymbol{x}_{n+1} = \Omega(1)\}$. As we have observed in Figure 1, our key insight is that pretrained $\mathbf{M}$ **scales this attention window with the Lipschitzness of the function class.** Generally speaking, larger $\mathbf{M}$ entails averaging over a smaller window and incurring less bias due to the function values of distant tokens in the estimate, while smaller $\mathbf{M}$ entails averaging over a larger window, resulting in larger bias due to distant token labels, but a smaller noise variance. Figure 2 further depicts this tradeoff.

**Connection to non-parametric estimation and the Nadaraya-Watson estimator.** A nonparametric estimation technique to interpolate between known values of a function is to use a kernel estimator. The Nadaraya-Watson (NW) estimator [29–31] is one such estimator, and interpolates the data as

$$f_{NW}(\boldsymbol{x}_{n+1}) = \sum_i \frac{K(x_{n+1}, x_i) f(x_i)}{\sum_j K(x_{n+1}, x_j)}$$

where $K(r) = e^{-r^2/h}$ for some bandwidth $h$. In Section B.1 we show that optimizing the pretraining loss (ICL) reduces to meta-learning the bandwidth of an NW estimator on a distribution of pretraining tasks. However, to our knowledge, the literature has not determined the optimal bandwidth for the kernel, as there has been no analysis of non-asymptotic lower bounds on the loss, which we need to characterize the optimal solution. A close work to ours is [32], which considers regression on general $L$-Lipschitz tasks, but this analysis provides only a tight upper bound on the loss.

**Remark 2.1** (Extreme cases). *Consider the following two settings.*

*(i) Constant functions. If each of the functions the attention unit sees in pretraining is constant, as in the **Left** column of Figure 1, it is best to consider an infinite attention window, that is, take $\mathbf{M} = \mathbf{0}_{d \times d}$ as this results in a uniform average over all the noisy token labels.*

*(ii) Rapidly changing functions. If the pretraining functions change rapidly, as in the **Right** column of Figure 1, attending to a distant token might only corrupt the estimate at the target. For example suppose the input tokens are used to construct Voronoi cells on the surface of the hypersphere, and the label for a new token in a cell is the label of the token used to construct that cell. The optimal estimator attends only to the single nearest token since this incurs error only from label noise.*

**Remark 2.2** (Softmax advantage). *To further highlight the utility of the softmax, we compare with linear attention [7, 9, 11], whose estimator can be written as $h_{LA}(\boldsymbol{x}) = \sum_i (f(\boldsymbol{x}_i) + \epsilon_i)\, \boldsymbol{x}_i^\top \mathbf{M} \boldsymbol{x}$, up to a universal scaling due to the value embedding. This is again a weighted combination of labels, but one that does not allow for adapting an attention window – any scaling of $\mathbf{M}$ does not change the relative weights placed on each label – unlike softmax attention. Please see Figure 1 (**Middle Row**) for a comparison of the weights used in the different estimators.*

## 3 Pretraining Learns Scale of Attention Window

One of our observations of the attention estimator $h_{SA}$ is that it computes a nearest neighbours regression. We hypothesize that the role of pretraining is to select a neighbourhood within which to select tokens for use in the estimator. In this section we characterize the radius of this neighborhood.

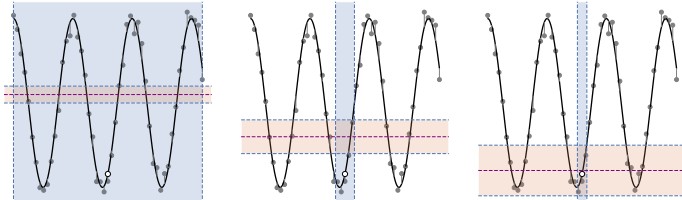

Figure 2: From **left to right**, as we **shrink the attention window** (shaded in blue), the estimator has **lower bias** (the expected value of the estimate, depicted in purple, is closer to the ground-truth label, depicted by the white circle) but **larger variance** (shaded in tan).

**Definition 3.1** (Lipschitzness). *A function $f : \mathcal{X} \to \mathbb{R}$ has Lipschitzness $L$ if $L$ is the smallest number satisfying $f(\boldsymbol{x}) - f(\boldsymbol{x}') \leq L\|\boldsymbol{x} - \boldsymbol{x}'\|$ for all $(\boldsymbol{x}, \boldsymbol{x}') \in \mathcal{X}^2$.*

The general requirement for the function classes to which our results apply is that the class should be invariant to isometries, each function should be Lipschitz, and the function value at two points should be less correlated as those points get further. These are written formally in Assumption B.4. To be concrete, we work with the following two function classes that satisfy these assumptions (this is shown in Lemmas C.3 and C.7) to derive explicit bounds.

**Definition 3.2** (Affine and ReLU Function Classes). *The function classes $\mathcal{F}_L^{aff}$ and $\mathcal{F}_L^+$ are respectively defined as:*

$$\mathcal{F}_L^{aff} := \{f : f(\boldsymbol{x}) = l\,\mathbf{w}^\top \boldsymbol{x} + b,\ \mathbf{w} \in \mathbb{S}^{d-1}, b, l \in [-L, L]\},$$
$$\mathcal{F}_L^+ := \{f : f(\boldsymbol{x}) = l_1(\mathbf{w}^\top \boldsymbol{x})_+ + l_2(-\mathbf{w}^\top \boldsymbol{x})_+ + b,\ \mathbf{w} \in \mathbb{S}^{d-1}, (b, l_1, l_2) \in [-L, L]^2\}.$$

*$D(\mathcal{F}_L^{aff}), D(\mathcal{F}_L^+)$ are induced by drawing $\mathbf{w} \sim \boldsymbol{\Sigma}\mathcal{U}^d$ and $b, l, l_1, l_2 \overset{i.i.d.}{\sim} \text{Unif}([-L, L])$ for some $\boldsymbol{\Sigma} \succ \mathbf{0}_{d \times d}$. Note that the max Lipschitzness of any function in these classes is $L$, and $(z)_+ := \max(z, 0)$.*

Next, we make the following assumption, similar to [7], on the covariate distribution.

**Assumption 3.3** (Covariate Distribution). *The covariate distribution satisfies $D_{\boldsymbol{x}} = \boldsymbol{\Sigma}^{-1}\mathcal{U}^d$.*

Now we are ready to state our main theorem that characterizes minimizers of (ICL).

**Theorem 3.4.** *Let Assumption 3.3 hold and tasks $f$ be drawn from **(Case 1)** $D(\mathcal{F}_L^{aff})$ or **(Case 2)** $D(\mathcal{F}_L^+)$. For $n = \Omega(1)$ and $\Omega(n^{-d/2}) \leq \sigma^2 \leq \mathcal{O}(nL^2)$, any minimizer of the pretraining loss (ICL) satisfies[2] $\mathbf{M}^* = w_{KQ}\boldsymbol{\Sigma}$, where for $\Lambda := \frac{nL^2}{\sigma^2}$, $\alpha := \frac{1}{d+4}$ and $\beta := \frac{1}{d+2}$:*

**(Case 1)** $\Omega\left(\Lambda^\alpha\right) \leq |w_{KQ}| \leq \mathcal{O}\left(\Lambda^{\frac{2\alpha}{1-\beta}}\right), \quad$ **(Case 2)** $\Omega\left(\Lambda^\beta\right) \leq |w_{KQ}| \leq \mathcal{O}\left(\Lambda^{2\beta}\right).$

Theorem 3.4 shows that optimizing the pretraining population loss in Equation (ICL) leads to attention key-query parameters that scale with the Lipschitzness of the function class, as well as the noise level and number of in-context samples. These bounds align with our observations from Figures 1 and 2 that softmax attention selects an attention window that shrinks with the function class Lipschitzness, recalling that larger $w_{KQ}$ results in a smaller window. Further, the dependencies of the bounds on $\sigma^2$ and $n$ are also intuitive, since larger noise should encourage wider averaging to average out the noise, and larger $n$ should encourage a smaller window since more samples makes it more likely that there are samples very close to the query. To our knowledge, this is the *first result showing that softmax attention learns properties of the task distribution during pretraining that facilitate ICL*.

**Learning Lipschitzness is critical to generalization.** We next give the following generalization result for downstream tasks.

**Theorem 3.5.** *Suppose softmax attention is first pretrained on tasks drawn from $D(\mathcal{F}_L^+)$ and then tested on an arbitrary $L-$Lipschitz task, then the loss on the new task is upper bounded as $\mathcal{L} \leq$*

---

[2]We further show in Appendix B that $\mathbf{M}^* = w_{KQ}\boldsymbol{\Sigma}$ for scalar $w_{KQ}$ holds for a broad family of rotationally-invariant function classes.

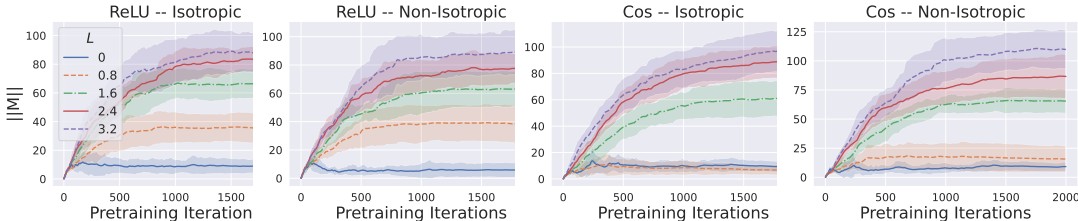

Figure 3: Spectral norm of $\mathbf{M}$ during pretraining with varying $L$. Each plot shows results for different task and covariate distributions, with (tasks, covariates) drawn from **(Left)** $(D(\mathcal{F}_L^+), \mathcal{U}^d)$, **(Middle-Left)** $(D(\mathcal{F}_L^+), \tilde{\mathcal{U}}^d)$, **(Middle-Right)** $(D(\mathcal{F}_L^{\cos}), \mathcal{U}^d)$, **(Right)** $(D(\mathcal{F}_L^{\cos}), \tilde{\mathcal{U}}^d)$, where $\tilde{\mathcal{U}}^d$ is a non-isotropic distribution on $\mathbb{S}^{d-1}$ (see Section 3.2 for its definition).

$\mathcal{O}(\frac{L^2}{\Lambda^\beta})$. Furthermore, if the new task is instead drawn from $D(\mathcal{F}_{L'}^+)$, the loss is lower bounded as $\mathcal{L} \geq \Omega(\frac{L'^2}{\Lambda^{2\beta}})$ for $L' > L$ and $\mathcal{L} \geq \Omega(\frac{\Lambda^{\beta d/2}}{n})$ for $L' < L$.

Theorem 3.5 shows that pretraining on $D(\mathcal{F}_L^+)$ yields a model that can in-context learn downstream tasks *if and only if* they have similar Lipschitzness as $L$. Thus, learning Lipschitzness is *both sufficient and necessary* for ICL. If the evaluation task Lipschitzness is much larger than that seen in pretraining, the pretrained model will give highly biased estimates. Conversely, if the evaluation Lipschitzness is much lower, the pretrained model will not optimally average the label noise.

**Necessity of Softmax.** To further emphasize the importance of the softmax in Theorem 3.4, we next study the performance of an analogous model with the softmax removed. We consider *linear self-attention* [7, 9, 11], which replaces the softmax activation with an identity operation. In particular, in the in-context regression setting we study, the prediction of $f(\boldsymbol{x}_{n+1})$ by linear attention and the corresponding pretraining loss are given by:

$$h_{LA}(\boldsymbol{x}_{n+1}) := \sum_{i=1}^{n}(f(\boldsymbol{x}_i) + \epsilon_i)\boldsymbol{x}_i^\top \mathbf{M} \boldsymbol{x}_{n+1},$$

$$\mathcal{L}_{\text{LA}}(\mathbf{M}) := \mathbb{E}_{f, \{\boldsymbol{x}_i\}_i, \{\epsilon_i\}_i}\left(h_{LA}(\boldsymbol{x}_{n+1}) - f(\boldsymbol{x}_{n+1})\right)^2.$$

As discussed in Remark 2.1, $h_{LA}(\boldsymbol{x}_{n+1})$ cannot adapt an attention window to the problem setting. We show below that this leads it to large ICL loss when tasks are drawn from $D(\mathcal{F}_L^+)$.

**Theorem 3.6** (Lower Bound for Linear Attention). *Consider pretraining on $\mathcal{L}_{LA}$ with tasks $f$ drawn from $D(\mathcal{F}_L^+)$ and covariates drawn from $\mathcal{U}^d$. Then for all $\mathbf{M} \in \mathbb{R}^{d \times d}$, $\mathcal{L}_{LA}(\mathbf{M}) = \Omega(L^2)$.*

This lower bound on $\mathcal{L}_{LA}$ is strictly larger than the upper bound on $\mathcal{L}$ from Theorem 3.5, up to factors in $d$, as long as $\frac{\sigma^2}{n} \leq 1$, which holds in all reasonable cases. Please see Appendix F for the proof.

### 3.1 Proof Sketch

To highlight the key insights of our analysis, in this section we consider a modification of the softmax attention that exhibits important properties of the original. Note that this approximation is for illustration only; the above results use the original softmax attention – see Appendices C, D, E. For now, consider a function class $\mathcal{F}_L := \{f : f(\boldsymbol{x}) = L\mathbf{w}^\top \boldsymbol{x}, \ \mathbf{w} \in \mathbb{S}^{d-1}\}$ of linear functions.

**(Temporary) modification of the softmax attention.** Rather than averaging over every token with a weight that decays exponentially with distance, we consider a modification which *uniformly* averages all tokens within a distance specified by $w_{KQ} = \|\mathbf{M}\|$. From Lemma B.5, without loss of generality (WLOG) we can consider $\mathbf{M} = w_{KQ}\mathbf{I}_d$. This means that, ignoring normalization, the weight assigned to $f(\boldsymbol{x}_i)$ by the true soft-max attention is $e^{-w_{KQ}\|\boldsymbol{x} - \boldsymbol{x}_i\|^2}$. That is, for all $\boldsymbol{x}_i$ satisfying $\|\boldsymbol{x} - \boldsymbol{x}_i\| < 1/\sqrt{w_{KQ}}$, the assigned weights within a constant factor of each other. Meanwhile, for $\boldsymbol{x}_i$ satisfying $\|\boldsymbol{x} - \boldsymbol{x}_i\| = \sqrt{c}/\sqrt{w_{KQ}}$ for $c > 1$, the weights are $e^{-c}$, decaying exponentially in $c$. This motivates us to consider a "modified softmax attention" given by $h_{MSA}(\boldsymbol{x}) := \sum_i \frac{f(\boldsymbol{x}_i)\mathbb{1}_i}{\sum_j \mathbb{1}_j}$, where $\mathbb{1}_j := \mathbb{1}\{\|\boldsymbol{x} - \boldsymbol{x}_j\| < 1/\sqrt{w_{KQ}}\}$.

**The In-context Loss.** The pretraining loss in Equation ICL can be decomposed as:

$$\mathcal{L}(w_{KQ}\mathbf{I}_d) = \underbrace{\mathbb{E}_{f,\{\boldsymbol{x}_i\}_i}\left(\sum_j \frac{(f(\boldsymbol{x}_{n+1}) - f(\boldsymbol{x}_j))\mathbb{1}_j}{\sum_j \mathbb{1}_j}\right)^2}_{=:\mathcal{L}_{\text{signal}}(w_{KQ})} + \underbrace{\mathbb{E}_{\{\boldsymbol{x}_i\}_i,\{\epsilon_i\}_i}\left(\sum_i \frac{\epsilon_i \mathbb{1}_i}{\sum_j \mathbb{1}_j}\right)^2}_{=:\mathcal{L}_{\text{noise}}(w_{KQ})}.$$

We first upper and lower bound each of these terms separately, starting with $\mathcal{L}_{\text{signal}}(w_{KQ})$.

**Noiseless Estimator Bias.** (Please see Appendix C) This term is the squared difference between an unweighted average of the token labels within a radius of $\boldsymbol{x}$, and the true label. Take $w_{KQ} = \Omega(1)$. Then for large $d$, most of the points $\boldsymbol{x}_i$ satisfying $\|\boldsymbol{x} - \boldsymbol{x}_i\| \leq 1/\sqrt{w_{KQ}}$ lie on the boundary of the cap, that is, $\|\boldsymbol{x} - \boldsymbol{x}_i\| < 1/\sqrt{w_{KQ}} \implies \|\boldsymbol{x} - \boldsymbol{x}_i\| \approx 1/\sqrt{w_{KQ}}$. This motivates us to approximate the set of points $\boldsymbol{x}_i$ satisfying the above as coming from a uniform distribution over just the boundary of the cap. The center of mass of a ring of radius $1/\sqrt{w_{KQ}}$ embedded on the surface of a hyper-sphere, is $\mathcal{O}(1/w_{KQ})$ from the boundary of a sphere, so the squared bias is $\Theta(L^2/w_{KQ}^2)$.

**Noise.** (Please see Appendix D for details) Since the noise is independent across tokens, we can write $\mathcal{L}_{\text{noise}}(w_{KQ}) = \frac{\sigma^2}{\sum_j \mathbb{1}_j}$, which is related to the number of tokens found within a $1/\sqrt{w_{KQ}}$ radius of $\boldsymbol{x}$. In Lemma G.1, we derive bounds for the measure of this region. For now, we replace the sum in the denominator with its expectation. We can bound $\frac{1}{\sum_j \mathbb{1}_j} = \Theta\left(w_{KQ}^{\frac{d}{2}}/n\right)$ as long as $w_{KQ} \lesssim n^{2/d}$.

**Combining the $\mathcal{L}_{\text{signal}}$ and $\mathcal{L}_{\text{noise}}$ terms.** (Please see Appendix E for details) Overall, we have $\mathcal{L} = \mathcal{L}_{\text{signal}} + \mathcal{L}_{\text{noise}}$ with $\mathcal{L}_{\text{signal}} = \Theta\left(L^2/w_{KQ}\right)$ and $\mathcal{L}_{\text{noise}} = \Theta\left(w_{KQ}^{\frac{d}{2}}\sigma^2/n\right)$. Minimizing this sum reveals that the optimal $w_{KQ}$ satisfies $w_{KQ} = \Theta\left(\left(nL^2/\sigma^2\right)^{\frac{2}{d+2}}\right)$.

### 3.2 Experiments

We next empirically verify our intuitions and results regarding learning the scale of the attention window. In all cases we use the Adam optimizer with one task sampled per round, use the noise distribution $D_\epsilon = \mathcal{N}(0, \sigma^2)$, and run 10 trials and plot means and standard deviations over these 10 trials. Please see Appendix J for full details as well as additional results.

**Ablations over $L$, $\sigma$ and $n$.** We verify whether the relationship between the attention window scale – i.e. $\|\mathbf{M}\|^{-1}$ – and $L$, $\sigma$ and $n$ matches our bounds in Theorem 3.4 for the case when tasks are drawn from $D(\mathcal{F}_L^+)$ and the covariates are drawn from $\mathcal{U}^d$, as well as whether these relationships generalize to additional function classes and covariate distributions. We train on tasks drawn from $D(\mathcal{F}_L^+)$ and $D(\mathcal{F}_L^{\cos})$, where $\mathcal{F}_L^{\cos} := \{f : f(\boldsymbol{x}) = \cos(L\mathbf{w}^\top \boldsymbol{x}), \mathbf{w} \in \mathbb{S}^{d-1}\}$ and $D(\mathcal{F}_L^{\cos})$ is induced by sampling $\mathbf{w} \sim \mathcal{U}^d$. In all cases we set $d = 5$, and use $(L, \sigma, n) = (1, 0.01, 20)$ if not ablating over these parameters, and vary only one of $\{L, \sigma, n\}$ and no other hyperparameters within each plot.

**Attention window scales inversely with $L$.** Figure 3 shows that $\|\mathbf{M}\|$ increases with $L$ in various settings. In Figure 3(Left, Middle-Left), tasks are drawn from $D(\mathcal{F}_L^+)$, and in Figure 3(Middle-Right, Right), they are drawn $D(\mathcal{F}_L^{\cos})$. In Figure 3(Left, Middle-Right), each $\boldsymbol{x}_i$ is drawn from $\mathcal{U}^d$, whereas in Figure 3(Middle-Left, Right), each $\boldsymbol{x}_i$ is drawn from a non-isotropic distribution $\tilde{\mathcal{U}}^d$ on $\mathbb{S}^{d-1}$ defined as follows. First, let $\mathbf{S}_d := \text{diag}([1, \ldots, d]) \in \mathbb{R}^{d \times d}$, then $\boldsymbol{x} \sim \tilde{\mathcal{U}}^d$ is generated by sampling $\hat{\boldsymbol{x}} \sim \mathcal{N}(\mathbf{0}_d, \mathbf{I}_d)$, then computing $\boldsymbol{x} = \frac{\mathbf{S}_d^{1/2}\hat{\boldsymbol{x}}}{\|\mathbf{S}_d^{1/2}\hat{\boldsymbol{x}}\|}$. Although larger $L$ implies larger $\|\nabla_{\boldsymbol{x}} f(\boldsymbol{x})\|$ on average across $f$, it is not clear that it implies larger $\|\nabla_{\mathbf{M}_K}\mathcal{L}(\mathbf{W}_K^\top \mathbf{W}_Q)\|$ nor $\|\nabla_{\mathbf{M}_Q}\mathcal{L}(\mathbf{W}_K^\top \mathbf{W}_Q)\|$, so it is surprising that larger $L$ implies larger pretrained $\mathbf{M}$ (although it is consistent with our results).

**Attention window scales with $\sigma$, inversely with $n$.** Figure 4 shows that the dependence of $\|\mathbf{M}\|$ on $\sigma$ and $n$ also aligns with Theorem 3.4. As expected, $\|\mathbf{M}\|$ increases slower during pretraining for larger $\sigma$ (shown in Figures 4(Left, Middle-Left)), since more noise encourages more averaging over a larger window to cancel out the noise. Likewise, $\|\mathbf{M}\|$ increases faster during pretraining for larger $n$ (shown in Figures 4(Middle-Right, Right)), since larger $n$ increases the likelihood that there is a highly informative sample in a small attention window. Here always the covariate distribution is $\mathcal{U}^d$.

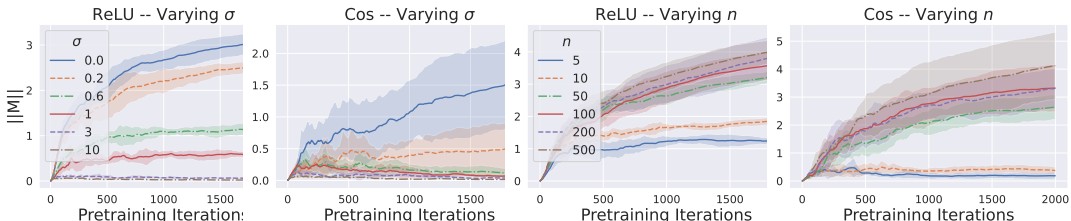

Figure 4: Spectral norm of $\mathbf{M}$ during pretraining on tasks drawn from $D(\mathcal{F}_1^+)$ in **Left, Middle-Right** and $D(\mathcal{F}_1^{\mathrm{cos}})$ in **Middle-Left, Right**. **Left, Middle-Left** show ablations over the noise standard deviation $\sigma$ and **Middle-Right, Right** show ablations over the number of context samples $n$.

**Learning new tasks in-context.** An implication of our work is that for the function classes we consider, the **softmax attention estimator does not adapt to the function class beyond its Lipschitzness**. We have already seen in Figures 3 and 4 that the growth of $\|\mathbf{M}\|$ during pretraining is similar across different function classes with the same Lipschitzness, as long as $\sigma$ and $n$ are fixed. Here we verify the conclusion from Theorem 3.5 that for fixed $n$ and $\sigma$, the necessary and sufficient condition for downstream generalization, measured by small ICL error, is that the pretraining and downstream tasks have similar Lipschitzness. Figure 5 supports this conclusion. Here we set $d = 5, n = 200, \sigma = 0.01$ and draw each $\boldsymbol{x}_i$ i.i.d. from $\mathcal{U}^d$. In Figure 5(Left, Middle-Left, Middle-Right), we train three attention units on tasks drawn from the 1-Lipschitz affine ($D(\mathcal{F}_1^{\mathrm{aff}})$), ReLU ($D(\mathcal{F}_1^+)$), and cosine ($D(\mathcal{F}_1^{\mathrm{cos}})$) task distributions. Each plot shows the test ICL error on tasks drawn from a distribution in $\{D(\mathcal{F}_1^{\mathrm{aff}}), D(\mathcal{F}_1^+), D(\mathcal{F}_1^{\mathrm{cos}})\}$. Performance is similar regardless of the pairing of pretraining and test distributions, as the Lipschitzness is the same in all cases, demonstrating that **pretraining on tasks with appropriate Lipschitzness is sufficient for generalization**.

Moreover, Figure 5(Right) shows that when the Lipschitzness of the pretraining tasks does *not* match that of the test tasks, ICL performance degrades sharply, even when the tasks otherwise share similar structure. Here the test task distribution is $D(\mathcal{F}_1^{\mathrm{cos}})$, and the pretraining task distributions are $D(\mathcal{F}_1^{\mathrm{aff}})$, $D(\mathcal{F}_{0.1}^{\mathrm{cos}})$, and $D(\mathcal{F}_{10}^{\mathrm{cos}})$. The only pretraining distribution that leads to downstream generalization is $D(\mathcal{F}_1^{\mathrm{aff}})$ since its Lipschitzness matches that of the downstream tasks, despite the fact that it is not a distribution over cosine functions, unlike the other distributions. Thus, these results lend credence to the idea that in addition to being sufficient, **pretraining on tasks with appropriate Lipschitzness is necessary for generalization**.

## 4 Softmax Attention Learns Direction of Attention Window

Thus far, we have considered distributions over tasks that treat the value of the input data in all directions within the ambient space as equally relevant to its label. However, in practice the ambient dimension of the input data is often much larger than its information content – the labels may change very little with many features of the data, meaning that such features are spurious. This is generally true of embedded language tokens, whose embedding dimension is typically far larger than the minimum dimension required to store them (logarithmic in the vocabulary size) [1]. Motivated by this, we define a notion of "direction-wise Lipschitzness" of a function class to allow for analyzing classes that may depend on some directions within the ambient input data space more than others.

**Definition 4.1** (Direction-wise Lipschitzness of Function Class). *The Lipschitzness of a function class $\mathcal{F}$ with domain $\mathcal{X} \subseteq \mathbb{R}^d$ in the direction $\mathbf{w} \in \mathbb{S}^{d-1}$ is defined as as the largest Lipschitz constant of all functions in $\mathcal{F}$ over the domain $\mathcal{X}$ projected onto $\mathbf{w}$, that is:*

$$Lip_{\mathbf{w}}(\mathcal{F}, \mathcal{X}) := \inf_{L \in \mathbb{R}}\{L : f(\mathbf{w}\mathbf{w}^\top \boldsymbol{x}) - f(\mathbf{w}\mathbf{w}^\top \boldsymbol{x}') \le L|\mathbf{w}^\top \boldsymbol{x} - \mathbf{w}^\top \boldsymbol{x}'| \ \forall \ (\boldsymbol{x}, \boldsymbol{x}') \in \mathcal{X}^2, f \in \mathcal{F}\}.$$

Using this definition, we analyze function classes consisting of linear functions with parameters lying in a subspace of $\mathbb{R}^d$, as follows:

**Definition 4.2** (Low-rank Linear Function Class). *The function class $\mathcal{F}_{\mathbf{B}}^{lin}$ is defined as $\mathcal{F}_{\mathbf{B}}^{lin} := \{f : f(\boldsymbol{x}) = \mathbf{a}^\top \mathbf{B}^\top \boldsymbol{x}, \ \mathbf{a} \in \mathbb{R}^k\}$, and $D(\mathcal{F}_{\mathbf{B}}^{lin})$ is induced by drawing $\mathbf{a} \sim \mathcal{U}^k$.*

where $\mathbf{B} \in \mathbb{O}^{d \times k}$ is a column-wise orthonormal matrix. Since our motivation is settings with low-dimensional structure, we can think of $k \ll d$. Let $\mathbf{B}_\perp \in \mathbb{O}^{d \times (d-k)}$ denote a matrix whose columns

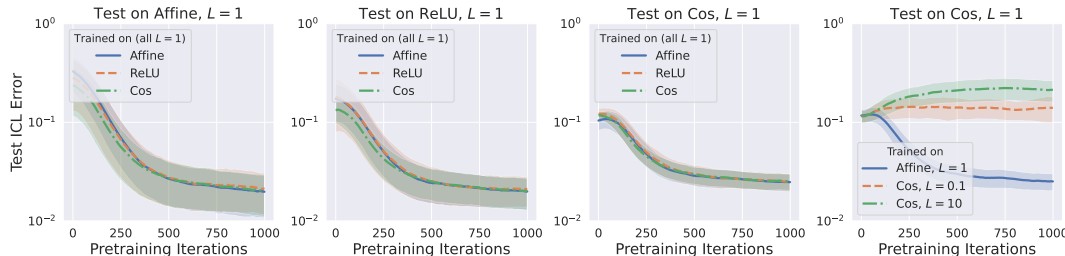

Figure 5: **Left, Middle-Left, Middle-Right:** The test error for softmax attention as it is trained on the distributions over 1-Lipschitz affine, ReLU, and cosine function ($D(\mathcal{F}_1^{\text{aff}})$, $D(\mathcal{F}_1^{+})$, and $D(\mathcal{F}_1^{\text{cos}})$, respectively), where the error is evaluated at each pretraining iteration on 5 tasks drawn from the distributions over the 1-Lipschitz (affine, ReLU, cosine) function classes in (**Left, Middle-Left, Middle-Right**), respectively. **Right:** The test error evaluated on tasks drawn from $D(\mathcal{F}_1^{\text{cos}})$ for three softmax attention trained on tasks drawn from $D(\mathcal{F}_1^{\text{aff}})$, $D(\mathcal{F}_{0.1}^{\text{cos}})$, and $D(\mathcal{F}_{10}^{\text{cos}})$, respectively.

form an orthonormal basis for the subspace perpendicular to $\text{col}(\mathbf{B})$, and note that the Lipschitzness of $\mathcal{F}_{\mathbf{B}}^{\text{lin}}$ in the direction $\mathbf{w}$ is $L$ if $\mathbf{w} \in \text{col}(\mathbf{B})$ and 0 if $\mathbf{w} \in \text{col}(\mathbf{B}_\perp)$. Observe that any function in $\mathcal{F}_{\mathbf{B}}^{\text{lin}}$ can be learned by projecting the input onto the non-zero Lipschitzness directions, i.e. $\text{col}(\mathbf{B})$, then solving a $k \ll d$-dimensional regression. To formally study whether softmax attention recovers $\text{col}(\mathbf{B})$, we assume the covariates are generated as follows.

**Assumption 4.3** (Covariate Distribution). *There are fixed constants $c_{\mathbf{u}} \neq 0$ and $-\infty < c_{\mathbf{v}} < \infty$ s.t. sampling $\boldsymbol{x}_i \sim D_{\boldsymbol{x}}$ is equivalent to $\boldsymbol{x}_i = c_{\mathbf{u}} \mathbf{B} \mathbf{u}_i + c_{\mathbf{v}} \mathbf{B}_\perp \mathbf{v}_i$ where $\mathbf{u}_i \sim \mathcal{U}^k$ and $\mathbf{v}_i \sim \mathcal{U}^{d-k}$.*

Assumption 4.3 entails that the data is generated by latent variables $\mathbf{u}_i$ and $\mathbf{v}_i$ that determine label-relevant and spurious features. This may be interpreted as a continuous analogue of dictionary learning models studied in feature learning works [33, 34]. We require no finite upper bound on $|c_{\mathbf{v}}|$ nor $\frac{1}{|c_{\mathbf{u}}|}$, so the data may be dominated by spurious features.

**Theorem 4.4.** *Let $\mathbf{B} \in \mathbb{O}^{d \times k}$ and consider the pretraining population loss (ICL) with $f \sim D(\mathcal{F}_{\mathbf{B}}^{\text{lin}})$. Suppose Assumption 4.3 holds, as well as at least one of two cases: (Case 1) $\sigma = 0$, or (Case 2) $n = 2$. Then among all $\mathbf{M} \in \mathcal{M} := \{\mathbf{M} \in \mathbb{R}^{d \times d} : \mathbf{M} = \mathbf{M}^\top, \|\mathbf{B}^\top \mathbf{M} \mathbf{B}\| \leq \frac{1}{c_{\mathbf{u}}^2}\}$, the minimizer of the pretraining population loss (ICL) is $\mathbf{M}^* = c\mathbf{B}\mathbf{B}^\top$ for some $c \in (0, \frac{1}{c_{\mathbf{u}}^2}]$.*

Theorem 4.4 shows that softmax attention can achieve dimensionality reduction during ICL on any downstream task that has non-zero Lipschitzness only in $\text{col}(\mathbf{B})$ by removing the zero-Lipschitzness features while pretraining on $\mathcal{F}_{\mathbf{B}}^{\text{lin}}$. Removing the zero-Lipschitzness features entails that the nearest neighbor prediction of pretrained softmax attention uses a neighborhood, i.e. attention window, defined strictly by projections of the input onto $\text{col}(\mathbf{B})$. To our knowledge, this is the *first result showing that softmax attention pretrained on ICL tasks recovers a shared low-dimensional structure among the tasks*. Please see Appendix J for empirical results verifying that softmax attention indeed recovers low-dimensional structure, even for tasks consisting of (nonlinear) generalized linear models.

## 5 Conclusion

We have presented, to our knowledge, the first results showing that softmax attention learns shared structure among pretraining tasks that facilitates downstream ICL. Moreover, we have provided empirical evidence suggesting that our conclusions about what softmax attention learns during pretraining generalize to function classes beyond those considered in our analysis.

**Limitations and Future Work. 1.** The model we use in this work is an attempt to understand a phenomenon that emerges in LLMs, which is that the output of the model can be 'primed' with some examples provided in the context that resembles few-shot learning, even though they are only trained on next token prediction. Establishing a mathematical framework for this remains an interesting question. **2.** We consider the output of a single layer of attention. Studying the nature of the solution when this is iterated over multiple trained layers is an interesting future prospect.

# 6 Acknowledgments

This work was supported in part by NSF Grants 2127697, 2019844, 2107037, and 2112471, ARO Grant W911NF2110226, ONR Grant N00014-19-1-2566, the Machine Learning Lab (MLL) at UT Austin, and the Wireless Networking and Communications Group (WNCG) Industrial Affiliates Program.

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

# Contents

# A  Additional Related Work

**Empirical study of ICL.** Several works have studied ICL of linear tasks in the framework introduced by [28], and demonstrated that pretrained transformers can mimic the behavior of gradient descent [11–13, 28], Newton's method [14], and certain algorithm selection approaches [13, 16]. [35] studied the same linear setting with the goal of understanding the role of pretraining task diversity, while [36] argued via experiments on general auto-regressive tasks that ICL implicitly constructs a learning objective and optimizes it within one forward pass. Other empirical works have both directly supported [37] and contradicted [38] the hypothesis that ICL is a gradient-based optimization algorithm via experiments on real ICL tasks, while [39] empirically concluded that induction heads with softmax attention are the key mechanism that enables ICL in transformers. Lastly, outside of the context of ICL, [40] noticed that the attention parameter matrices of trained transformers are often close to scaled identities in practice, consistent with our findings on the importance of learning a scale to softmax attention training.

**Transformer training dynamics.** [21] and [41] studied the dynamics of softmax attention trained with gradient descent, but assumed orthonormal input features and either linear tasks [21] or that the softmax normalization is a fixed constant [41]. [42] proved that softmax attention with diagonal weight matrices incrementally learns features during gradient-based training. Other work has shown that trained transformers can learn topic structure [43], spatial structure [44], visual features [45] and support vectors [46, 47] in specific settings disjoint from ICL.

**Expressivity of transformers.** Multiple works have shown that transformers with linear [11, 36], ReLU [13, 14, 48], and softmax [12, 15] attention are expressive enough to implement general-purpose machine learning algorithms during ICL, including gradient descent. A series of works have shown the existence of transformers that recover sparse functions of the input data [49–52]. [53] studied the statistical complexity the learning capabilities of attention with random weights. More broadly, [54–59] have analyzed various aspects of the expressivity of transformers.

**Other studies of softmax attention.** [60] hypothesized that the role of the softmax in attention is to facilitate a mixture-of-experts algorithm amenable to unstructured training data. [27] formulated a softmax regression problem and analyzed the convergence of a stylized algorithm to solve it. [22] showed that in a setting with ICL regression tasks a la [28], a kernel regressor akin to softmax attention with $\mathbf{M}$ equal to the inverse covariance of $\mathbf{x}$ converges to the Bayes posterior for a new ICL task – in this setting the conditional distribution of the label given the query and $n$ labelled context samples – polynomially with the number of context samples, but did not study what softmax attention learns during pretraining. [61] also compared softmax and linear attention, but focused on softmax's greater capacity to separate data from two classes. [62] and [63] investigate the Lipschitz constant of attention rather than what attention learns.

**Non-parametric regression.** Our results imply that pretraining softmax attention reduces to the problem of meta-learning the bandwidth of a Nadaraya-Watson estimator with a Gaussian kernel. However, to our knowledge, the non-parametric regression literature has not addressed this problem. The closest work is [32], which only upper bounds the noiseless loss, and only in the limit $n \to \infty$, whereas our result characterizes the optimal bandwidth, which requires upper and lower bounds on the noisy loss.

# B  Preliminaries

We first justify our claim that the first $d$ rows of the last column of $\mathbf{W}_V$ can be set to $\mathbf{0}_d$ for any optimal choice of parameters.

**Lemma B.1.** *If under the function distribution, a function $f$ is equally likely as likely as $-f$, then any optimal solution to $\mathcal{L}(\mathbf{W}_V, \mathbf{W}_K, \mathbf{W}_Q)$ in 3 satisfies $\mathbf{W}_V = \begin{pmatrix} \mathbf{0}_{d \times d} & \mathbf{0}_{d \times 1} \\ \mathbf{0}_{1 \times d} & c \end{pmatrix}$.*

*Proof.* For readability we write $\beta_i = e^{-w_{KQ}\|\boldsymbol{x}_i - \boldsymbol{x}_{n+1}\|^2} \sum_j e^{-w_{KQ}\|\boldsymbol{x}_j - \boldsymbol{x}_{n+1}\|^2}$ Suppose $\mathbf{W}_V = \begin{pmatrix} \mathbf{0}_{d\times d} & \mathbf{v} \\ \mathbf{0}_{1\times d} & c \end{pmatrix}$ was optimal, then the loss can be written

$$\mathcal{L} = \mathbb{E}_{f,\{\boldsymbol{x}_i\}}\left[\left(\sum_i c\left(f(\boldsymbol{x}_i) + \epsilon_i\right)\beta_i + \sum_i \mathbf{v}^\top \boldsymbol{x}_i \beta_i - f(\boldsymbol{x}_{n+1})\right)^2\right].$$

But because $f$ and $-f$ are equally likely, and because the noise is also symmetric about 0, we can write this as

$$\mathcal{L} = \frac{1}{2}\mathbb{E}_{f,\{\boldsymbol{x}_i\},\{\epsilon_i\}}\left[\left(\sum_i c\left(f(\boldsymbol{x}_i) + \epsilon_i\right)\beta_i + \sum_i \mathbf{v}^\top \boldsymbol{x}_i \beta_i - f(\boldsymbol{x}_{n+1})\right)^2\right]$$

$$+ \frac{1}{2}\mathbb{E}_{f,\{\boldsymbol{x}_i\},\{\epsilon_i\}}\left[\left(\sum_i c\left((-f)(\boldsymbol{x}_i) - \epsilon_i\right)\beta_i + \sum_i \mathbf{v}^\top \boldsymbol{x}_i \beta_i - (-f)(\boldsymbol{x}_{n+1})\right)^2\right]$$

We can couple the noise $\{\epsilon_i\}$ and the data $\{\boldsymbol{x}_i\}$ in the two summands above to write this as

$$\mathbb{E}\left[(A + B + C)^2 + (-A + B - C)^2\right],$$

where $A = \sum_i cf(\boldsymbol{x}_i)\beta_i - f(\boldsymbol{x}) = -\left(\sum_i c(-f)(\boldsymbol{x}_i)\beta_i\right)$, $B = \sum_i \mathbf{v}^\top \boldsymbol{x}_i \beta_i$, and $C = \sum_i c\epsilon_i\beta_i$. We can set $B = 0$ simply by setting $\mathbf{v} = \mathbf{0}_{d\times 1}$, and this has loss

$$\mathcal{L} = \mathbb{E}_{f,\{\boldsymbol{x}_i\}}\left[\left(\sum_i c\left(f(\boldsymbol{x}_i) + \epsilon_i\right)\beta_i - f(\boldsymbol{x}_{n+1})\right)^2\right]$$

$$= \frac{1}{2}\left(\mathbb{E}\left[(A + C)^2 + (-A - C)^2\right]\right) \leq \frac{1}{2}\left(\mathbb{E}\left[(A + B + C)^2 + (-A + B - C)^2\right]\right)$$

$\square$

In all of the distributions over functions we consider for pretraining, $f$ is equally likely as $-f$, so without loss of generality we set all elements of $\mathbf{W}_V$ besides the $(d+1, d+1)$-th to 0. For simplicity, we set the $(d+1, d+1)$-th element to 1.

**Assumption B.2** (Covariate Distribution). *For each token $\boldsymbol{x}$, first we draw $\tilde{\boldsymbol{x}}$ as $\tilde{\boldsymbol{x}} \sim \mathcal{U}^d$. Then $\boldsymbol{x}$ is constructed as $\boldsymbol{x} = \boldsymbol{\Sigma}^{1/2}\tilde{\boldsymbol{x}}$.*

**Definition B.3** (Linear and 2-ReLU Function Classes). *The function classes $\mathcal{F}_L^{lin}$ and $\mathcal{F}_L^+$ are respectively defined as:*

$$\mathcal{F}_L^{lin} := \{f_{\mathbf{w}} : f_{\mathbf{w}}(\boldsymbol{x}) = l\mathbf{w}^\top \boldsymbol{x} + b, \ \mathbf{w} \in \mathbb{S}^{d-1}, \ l \in [-L, L]\}, \tag{5}$$

$$\mathcal{F}_L^+ := \{f_{\mathbf{w}} : f_{\mathbf{w}}(\boldsymbol{x}) = l_1 \, ReLU(\mathbf{w}^\top \boldsymbol{x}) + l_2 \, ReLU(-\mathbf{w}^\top \boldsymbol{x}) + b, \ \mathbf{w} \in \mathbb{S}^{d-1}\}. \tag{6}$$

*$D(\mathcal{F}_L^{lin}), D(\mathcal{F}_L^+)$ are induced by drawing $\mathbf{w} \sim \mathcal{N}(\mathbf{0}, \boldsymbol{\Sigma}^{-1})$ and $b, l, l_1, l_2 \sim Unif([-L, L])$. We say that these classes are $L-$Lipschitz, because the maximum Lipschitz constant for any function in the class is $L$.*

Note that because $\|\boldsymbol{\Sigma}^{-1/2}\boldsymbol{x}_i\| = 1$ always, we have

$2\,\boldsymbol{x}_i\,\mathbf{M}\,\boldsymbol{x}_{n+1}$

$= \|\boldsymbol{\Sigma}^{-1/2}\boldsymbol{x}_i\|^2 + \|\boldsymbol{\Sigma}^{1/2}\mathbf{M}\boldsymbol{\Sigma}^{1/2}\boldsymbol{\Sigma}^{-1/2}\boldsymbol{x}_{n+1}\|^2 - \|\boldsymbol{\Sigma}^{-1/2}\boldsymbol{x}_i - \boldsymbol{\Sigma}^{1/2}\mathbf{M}\boldsymbol{\Sigma}^{1/2}\boldsymbol{\Sigma}^{-1/2}\boldsymbol{x}_{n+1}\|^2.$

Let $\mathbf{M}' = \boldsymbol{\Sigma}^{1/2}\mathbf{M}\boldsymbol{\Sigma}^{1/2}$. This means the attention estimator can be rewritten as

$$h_{SA}(\boldsymbol{x}) := \sum_i \frac{f(\boldsymbol{x}_i)e^{\boldsymbol{x}_i^\top \mathbf{M}\boldsymbol{x}_{n+1}}}{\sum_j e^{\boldsymbol{x}_j^\top \mathbf{M}\boldsymbol{x}_{n+1}}} = \sum_i \frac{f(\boldsymbol{x}_i)e^{-\|\boldsymbol{\Sigma}^{-1/2}\boldsymbol{x}_i - \mathbf{M}'\boldsymbol{\Sigma}^{-1/2}\boldsymbol{x}_{n+1}\|^2}}{\sum_j e^{-\|\boldsymbol{\Sigma}^{-1/2}\boldsymbol{x}_j - \mathbf{M}'\boldsymbol{\Sigma}^{-1/2}\boldsymbol{x}_{n+1}\|^2}} \tag{7}$$

So the attention a token $\boldsymbol{x}_{n+1}$ places on another $\boldsymbol{x}_i$ is related to the distance between $\mathbf{M}'\boldsymbol{\Sigma}^{-1/2}\boldsymbol{x}_{n+1}$ and $\boldsymbol{\Sigma}^{-1/2}\boldsymbol{x}_i$. It is natural to suppose under some symmetry conditions that $\mathbf{M}'$ is best chosen to be a scaled identity matrix so that the attention actually relates to a distance between tokens. Below we discus sufficient conditions for this.

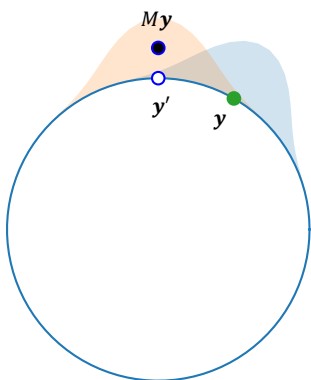

Figure 6: Comparison between using $\mathbf{M}$ and $\omega$ in Lemma B.5. Here we denote $\boldsymbol{y} := \boldsymbol{y}_{n+1}$. Under the attention induced by $\mathbf{M}$, the center of attention for $\boldsymbol{y}$ is actually $\boldsymbol{y}'$, and the attention weights are depicted by the light orange shading. Under the attention induced by $\omega$, the center of attention for $\boldsymbol{y}$ is $\boldsymbol{y}$ and the weights are depicted by the light blue shading. Naturally, using the blue shaded attention should lead to a better estimate of $f(\mathbf{y})$ under mild regularity conditions.

**Assumption B.4.** *The function class $\mathcal{F}$ and distribution $D(\mathcal{F})$ satisfy*

1. $|f(\boldsymbol{x}) - f(\boldsymbol{y})| \leq L \|\boldsymbol{x} - \boldsymbol{y}\|_{\boldsymbol{\Sigma}^{-1}} \ \forall \boldsymbol{x}, \boldsymbol{y} \in \mathcal{X}^2, f \in \mathcal{F}$

2. $\mathbb{E}_{f \sim D(\mathcal{F})}[f(\boldsymbol{x})f(\boldsymbol{y})] = \rho(\boldsymbol{x}^\top \boldsymbol{y}) \ \forall \boldsymbol{x}, \boldsymbol{y} \in \mathcal{X}^2$, *for some monotonically increasing $\rho$.*

3. *For any isometry $\phi$ preserving the unit sphere, and $f \in \mathcal{F}$, we have $f \circ \phi \in \mathcal{F}$.*

**Lemma B.5.** *Under Assumption B.4, any minimizer of Equation ICL satisfies $\mathbf{M}^* = w_{KQ}\boldsymbol{\Sigma}^{-1}$ for some scalar $w_{KQ} \geq 0$.*

*Proof.* Let $\{\boldsymbol{y}_i\} = \{\boldsymbol{\Sigma}^{-1/2}\boldsymbol{x}_i\}$. Suppose $\mathbf{M}\boldsymbol{y}_{n+1} \neq c\,\boldsymbol{y}_{n+1}$ for any $c > 0$ for some $\boldsymbol{y}_{n+1}$. Take $c_{\boldsymbol{y}_{n+1}} = \|\mathbf{M}\boldsymbol{y}_{n+1}\|$ and $\boldsymbol{y}'_{n+1} = \frac{\mathbf{M}\boldsymbol{y}_{n+1}}{c_{\boldsymbol{y}_{n+1}}}$ (the projection of $\boldsymbol{y}$ onto the sphere). Consider a function $\omega : \mathbb{R}^d \to \mathbb{R}^d$ satisfying $\omega(\boldsymbol{y}_{n+1}) = c_{\boldsymbol{y}_{n+1}}\,\boldsymbol{y}_{n+1}$. Note that this need not be linear. Let $\phi$ denote a rotation that sends $\boldsymbol{y}'_{n+1}$ to $\boldsymbol{y}_{n+1}$.

We show that $\mathcal{L}(\mathbf{M}) > \mathcal{L}(\omega)$, that is, it is favorable to *not* rotate $\boldsymbol{y}_{n+1}$. We have

$$\mathcal{L}(\mathbf{M}) = \mathbb{E}_{f, \boldsymbol{y}_{n+1}, \{\boldsymbol{y}_i\}}\left[\left(f(\boldsymbol{y}_{n+1}) - \frac{\sum_i f(\boldsymbol{y}_i)e^{-\|\boldsymbol{y}_i - \mathbf{M}\boldsymbol{y}_{n+1}\|^2}}{\sum_j e^{-\|\boldsymbol{y}_j - \mathbf{M}\boldsymbol{y}_{n+1}\|^2}}\right)^2\right]$$

$$= \mathbb{E}_{f, \boldsymbol{y}_{n+1}, \{\boldsymbol{y}_i\}}\, f(\boldsymbol{y}_{n+1})^2 + \mathbb{E}_{f, \boldsymbol{y}_{n+1}, \{\boldsymbol{y}_i\}}\left[\left(\frac{\sum_i f(\boldsymbol{y}_i)e^{-\|\boldsymbol{y}_i - \mathbf{M}\boldsymbol{y}_{n+1}\|^2}}{\sum_j e^{-\|\boldsymbol{y}_j - \mathbf{M}\boldsymbol{y}_{n+1}\|^2}}\right)^2\right]$$

$$- 2\,\mathbb{E}_{f, \boldsymbol{y}_{n+1}, \{\boldsymbol{y}_i\}}\left[\sum_i \frac{f(\boldsymbol{y}_{n+1})f(\boldsymbol{y}_i)e^{-\|\boldsymbol{y}_i - \mathbf{M}\boldsymbol{y}_{n+1}\|^2}}{\sum_j e^{-\|\boldsymbol{y}_j - \mathbf{M}\boldsymbol{y}_{n+1}\|^2}}\right]$$

Lets compare this with the loss of $\omega$. For a depiction of this, please see Figure 6

$$\mathcal{L}(\omega) = \mathbb{E}_{f, \boldsymbol{y}_{n+1}, \{\boldsymbol{y}_i\}}\left[\left(f(\boldsymbol{y}_{n+1}) - \frac{\sum_i f(\boldsymbol{y}_i)e^{-\|\boldsymbol{y}_i - \omega(\boldsymbol{y}_{n+1})\|^2}}{\sum_j e^{-\|\boldsymbol{y}_j - \omega(\boldsymbol{y}_{n+1})\|^2}}\right)^2\right]$$

$$= \mathbb{E}_{f, \boldsymbol{y}_{n+1}, \{\boldsymbol{y}_i\}}\, f(\boldsymbol{y}_{n+1})^2 + \mathbb{E}_{f, \boldsymbol{y}_{n+1}, \{\boldsymbol{y}_i\}}\left[\left(\frac{\sum_i f(\boldsymbol{y}_i)e^{-\|\boldsymbol{y}_i - \omega(\boldsymbol{y}_{n+1})\|^2}}{\sum_j e^{-\|\boldsymbol{y}_j - \omega(\boldsymbol{y}_{n+1})\|^2}}\right)^2\right]$$

$$-2\,\mathbb{E}_{f,\boldsymbol{y}_{n+1},\{\boldsymbol{y}_i\}}\left[\sum_i \frac{f(\boldsymbol{y}_{n+1})f(\boldsymbol{y}_i)e^{-\|\boldsymbol{y}_i-\omega(\boldsymbol{y}_{n+1})\|^2}}{\sum_j e^{-\|\boldsymbol{y}_j-\omega(\boldsymbol{y}_{n+1})\|^2}}\right]$$

There are three terms to compare. The first in each is identical. The second is also the same:

$$\mathbb{E}_{f,\boldsymbol{y}_{n+1},\{\boldsymbol{y}_i\}}\left[\left(\frac{\sum_i f(\boldsymbol{y}_i)e^{-\|\boldsymbol{y}_i-\mathbf{M}\boldsymbol{y}_{n+1}\|^2}}{\sum_j e^{-\|\boldsymbol{y}_j-\mathbf{M}\boldsymbol{y}_{n+1}\|^2}}\right)^2\right]$$

$$=\mathbb{E}_{\boldsymbol{y}_{n+1}}\mathbb{E}_{f,\{\boldsymbol{y}_i\}}\left[\left(\frac{\sum_i f(\boldsymbol{y}_i)e^{-\|\boldsymbol{y}_i-\mathbf{M}\boldsymbol{y}_{n+1}\|^2}}{\sum_j e^{-\|\boldsymbol{y}_j-\mathbf{M}\boldsymbol{y}_{n+1}\|^2}}\right)^2\right]$$

$$=\mathbb{E}_{\boldsymbol{y}_{n+1}}\mathbb{E}_{f,\{\boldsymbol{y}_i\}}\left[\left(\frac{\sum_i f(\boldsymbol{y}_i)e^{-\|\boldsymbol{y}_i-c_{\boldsymbol{y}_{n+1}}\boldsymbol{y}'_{n+1}\|^2}}{\sum_j e^{-\|\boldsymbol{y}_j-c_{\boldsymbol{y}_{n+1}}\boldsymbol{y}'_{n+1}\|^2}}\right)^2\right]$$

$$=\mathbb{E}_{\boldsymbol{y}_{n+1}}\mathbb{E}_{f,\{\boldsymbol{y}_i\}}\left[\left(\frac{\sum_i f(\phi(\boldsymbol{y}_i))e^{-\|\phi(\boldsymbol{y}_i)-c_{\boldsymbol{y}_{n+1}}\boldsymbol{y}_{n+1}\|^2}}{\sum_j e^{-\|\phi(\boldsymbol{y}_j)-c_{\boldsymbol{y}_{n+1}}\boldsymbol{y}_{n+1}\|^2}}\right)^2\right]\qquad\text{rotational symmetry of }\{\boldsymbol{y}_i\},\boldsymbol{y}_{n+1}$$

$$=\mathbb{E}_{\boldsymbol{y}_{n+1}}\mathbb{E}_{f,\{\boldsymbol{y}_i\}}\left[\left(\frac{\sum_i f(\boldsymbol{y}_i)e^{-\|\boldsymbol{y}_i-c_{\boldsymbol{y}_{n+1}}\boldsymbol{y}_{n+1}\|^2}}{\sum_j e^{-\|\boldsymbol{y}_j-c_{\boldsymbol{y}_{n+1}}\boldsymbol{y}_{n+1}\|^2}}\right)^2\right]\qquad\text{rotational symmetry of }\{\boldsymbol{y}_i\}$$

The third takes some more work. For any choice of $\{\boldsymbol{y}_i\}$, let

$$\alpha_{\boldsymbol{y}_{n+1},\{\boldsymbol{y}_i\}}(\boldsymbol{y}_*)=\frac{e^{-\|\boldsymbol{y}_{n+1}-\boldsymbol{y}_*\|^2}}{e^{-\|\boldsymbol{y}_{n+1}-\boldsymbol{y}_*\|^2}+\sum_j e^{-\|\boldsymbol{y}_{n+1}-\boldsymbol{y}_i\|^2}}.$$

We see that $\alpha_{\boldsymbol{y}_{n+1},\{\boldsymbol{y}_i\}}(\boldsymbol{y}_*)$ varies monotonically with $\boldsymbol{y}_{n+1}^\top\boldsymbol{y}_*$ for all $\boldsymbol{y}_{n+1},\{\boldsymbol{y}_i\}$. That is,

$$\boldsymbol{y}_*^\top\boldsymbol{y}_{n+1}>\boldsymbol{y}_*'^\top\boldsymbol{y}_{n+1}\implies\alpha_{\boldsymbol{y}_{n+1},\{\boldsymbol{y}_i\}}(\boldsymbol{y}_*)>\alpha_{\boldsymbol{y}_{n+1},\{\boldsymbol{y}_i\}}(\boldsymbol{y}_*'),$$

$$\mathbb{E}_{f,\boldsymbol{y}_{n+1},\{\boldsymbol{y}_i\}}\left[\sum_i\frac{f(\boldsymbol{y}_{n+1})f(\boldsymbol{y}_i)e^{-\|\boldsymbol{y}_i-\mathbf{M}\boldsymbol{y}_{n+1}\|^2}}{\sum_j e^{-\|\boldsymbol{y}_j-\mathbf{M}\boldsymbol{y}_{n+1}\|^2}}\right]$$

$$=\mathbb{E}_{\boldsymbol{y}_{n+1},\{\boldsymbol{y}_i\}}\left[\sum_i\frac{\mathbb{E}_f\left[f(\boldsymbol{y}_{n+1})f(\boldsymbol{y}_i)\right]e^{-\|\boldsymbol{y}_i-\mathbf{M}\boldsymbol{y}_{n+1}\|^2}}{\sum_j e^{-\|\boldsymbol{y}_j-\mathbf{M}\boldsymbol{y}_{n+1}\|^2}}\right]$$

$$=\mathbb{E}_{\boldsymbol{y}_{n+1},\{\boldsymbol{y}_i\}}\left[\sum_i\frac{\rho(\boldsymbol{y}_{n+1}^\top\boldsymbol{y}_i)e^{-\|\boldsymbol{y}_i-\mathbf{M}\boldsymbol{y}_{n+1}\|^2}}{\sum_j e^{-\|\boldsymbol{y}_j-\mathbf{M}\boldsymbol{y}_{n+1}\|^2}}\right]$$

$$=n\,\mathbb{E}_{\boldsymbol{y}_{n+1},\boldsymbol{y}_*,\{\boldsymbol{y}_i\}_{i=[n-1]}}\left[\frac{\rho(\boldsymbol{y}_{n+1}^\top\boldsymbol{y}_*)e^{-\|\boldsymbol{y}_*-\mathbf{M}\boldsymbol{y}_{n+1}\|^2}}{e^{-\|\boldsymbol{y}_*-\mathbf{M}\boldsymbol{y}_{n+1}\|^2}+\sum_j e^{-\|\boldsymbol{y}_j-\mathbf{M}\boldsymbol{y}_{n+1}\|^2}}\right]$$

$$=n\,\mathbb{E}_{\boldsymbol{y}_{n+1},\boldsymbol{y}_*,\{\boldsymbol{y}_i\}_{i=[n-1]}}\left[\rho(\boldsymbol{y}_{n+1}^\top\boldsymbol{y}_*)\alpha_{\mathbf{M}\boldsymbol{y}_{n+1},\{\boldsymbol{y}_i\}}(\boldsymbol{y}_*)\right]$$

$$=n\,\mathbb{E}_{\boldsymbol{y}_{n+1},\boldsymbol{y}_*,\{\boldsymbol{y}_i\}_{i=[n-1]}}\left[\rho(\boldsymbol{y}_{n+1}^\top\boldsymbol{y}_*)\alpha_{c_{\boldsymbol{y}_{n+1}}\boldsymbol{y}',\{\boldsymbol{y}_i\}}(\boldsymbol{y}_*)\right]$$

$$=n\,\mathbb{E}_{\boldsymbol{y}_{n+1},\boldsymbol{y}_*,\{\boldsymbol{y}_i\}_{i=[n-1]}}\left[\rho(\boldsymbol{y}_{n+1}^\top\boldsymbol{y}_*)\alpha_{c_{\boldsymbol{y}_{n+1}}\boldsymbol{y}_{n+1},\{\phi^{-1}(\boldsymbol{y}_i)\}}(\phi^{-1}(\boldsymbol{y}_*))\right]$$

$$=n\,\mathbb{E}_{\boldsymbol{y}_{n+1},\boldsymbol{y}_*,\{\boldsymbol{y}_i\}_{i=[n-1]}}\left[\rho(\boldsymbol{y}_{n+1}^\top\boldsymbol{y}_*)\alpha_{c_{\boldsymbol{y}_{n+1}}\boldsymbol{y}_{n+1},\{\boldsymbol{y}_i\}}(\phi^{-1}(\boldsymbol{y}_*))\right]$$

Similarly, we have

$$\mathbb{E}_{f,\boldsymbol{y}_{n+1},\{\boldsymbol{y}_i\}}\left[\sum_i\frac{f(\boldsymbol{y}_{n+1})f(\boldsymbol{y}_i)e^{-\|\boldsymbol{y}_i-\omega(\boldsymbol{y}_{n+1})\|^2}}{\sum_j e^{-\|\boldsymbol{y}_j-\omega(\boldsymbol{y}_{n+1})\|^2}}\right]$$

$$= \mathbb{E}_{\boldsymbol{y}_{n+1}, \{\boldsymbol{y}_i\}} \left[ \sum_i \frac{\mathbb{E}_f \left[ f(\boldsymbol{y}_{n+1}) f(\boldsymbol{y}_i) \right] e^{-\| \boldsymbol{y}_i - c_{\boldsymbol{y}_{n+1}} \boldsymbol{y}_{n+1} \|^2}}{\sum_j e^{-\| \boldsymbol{y}_j - c_{\boldsymbol{y}_{n+1}} \boldsymbol{y}_{n+1} \|^2}} \right]$$

$$= \mathbb{E}_{\boldsymbol{y}_{n+1}, \{\boldsymbol{y}_i\}} \left[ \sum_i \frac{\rho(\boldsymbol{y}_{n+1}^\top \boldsymbol{y}_i) e^{-\| \boldsymbol{y}_i - c_{\boldsymbol{y}_{n+1}} \boldsymbol{y}_{n+1} \|^2}}{\sum_j e^{-\| \boldsymbol{y}_j - c_{\boldsymbol{y}_{n+1}} \boldsymbol{y}_{n+1} \|^2}} \right]$$

$$= n \, \mathbb{E}_{\boldsymbol{y}_{n+1}, \boldsymbol{y}_*, \{\boldsymbol{y}_i\}_{i=[n-1]}} \left[ \frac{\rho(\boldsymbol{y}_{n+1}^\top \boldsymbol{y}_*) e^{-\| \boldsymbol{y}_* - c_{\boldsymbol{y}_{n+1}} \boldsymbol{y}_{n+1} \|^2}}{e^{-\| \boldsymbol{y}_* - c_{\boldsymbol{y}_{n+1}} \boldsymbol{y}_{n+1} \|^2} + \sum_j e^{-\| \boldsymbol{y}_j - c_{\boldsymbol{y}_{n+1}} \boldsymbol{y}_{n+1} \|^2}} \right]$$

$$= n \, \mathbb{E}_{\boldsymbol{y}_{n+1}, \boldsymbol{y}_*, \{\boldsymbol{y}_i\}_{i=[n-1]}} \left[ \rho(\boldsymbol{y}_{n+1}^\top \boldsymbol{y}_*) \alpha_{c_{\boldsymbol{y}_{n+1}} \boldsymbol{y}_{n+1}, \{\boldsymbol{y}_i\}}(\boldsymbol{y}_*) \right]$$

Critically, for a given $\boldsymbol{y}_{n+1}$, $\alpha_{\boldsymbol{y}, \{\boldsymbol{y}_i\}}(\boldsymbol{y}_*)$ can be re-parameterized as
$\alpha_{\boldsymbol{y}_{n+1}, \{\boldsymbol{y}_i\}}(\boldsymbol{y}_*) = \alpha'_{\{\boldsymbol{y}_i\}}(\boldsymbol{y}_* - \boldsymbol{y}_{n+1})$ where $\alpha'_{\{\boldsymbol{y}_i\}}$ is symmetric about 0 and decreasing. Similarly,
$\rho(\boldsymbol{y}_{n+1}^\top \boldsymbol{y}_*)$ can be re-parameterized as $\rho(\boldsymbol{y}_{n+1}^\top \boldsymbol{y}_*) = \rho'(\boldsymbol{y}_* - \boldsymbol{y}_{n+1})$ where $\alpha', \rho'$ are symmetric decreasing rearrangement (that is, the set of points $\boldsymbol{z}$ such that $\rho(\boldsymbol{x}) > r$ is a ball about the origin). From Lemma I.2 we then have

$$\mathbb{E}_{\boldsymbol{y}_{n+1}} \mathbb{E} \, \boldsymbol{y}_*, \{\boldsymbol{y}_i\}_{i=[n-1]} \left[ \rho(\boldsymbol{y}_{n+1}^\top \boldsymbol{y}_*) \alpha_{c_{\boldsymbol{y}_{n+1}} \boldsymbol{y}_{n+1}, \{\boldsymbol{y}_i\}}(\phi^{-1}(\boldsymbol{y}_*)) \right]$$

$$= \mathbb{E}_{\boldsymbol{y}_{n+1}} \mathbb{E} \, \boldsymbol{y}_*, \{\boldsymbol{y}_i\}_{i=[n-1]} \left[ \rho'(\| \boldsymbol{y}_{n+1} - \boldsymbol{y}_* \|) \alpha_{\{\boldsymbol{y}_i\}}(\| \boldsymbol{y}_{n+1} - \phi^{-1} \boldsymbol{y}_* \|) \right]$$

$$< \mathbb{E}_{\boldsymbol{y}_{n+1}} \mathbb{E} \, \boldsymbol{y}_*, \{\boldsymbol{y}_i\}_{i=[n-1]} \left[ \rho'(\| \boldsymbol{y}_{n+1} - \boldsymbol{y}_* \|) \alpha_{\{\boldsymbol{y}_i\}}(\| \boldsymbol{y}_{n+1} - \boldsymbol{y}_* \|) \right]$$

$$= \mathbb{E}_{\boldsymbol{y}_{n+1}} \mathbb{E} \, \boldsymbol{y}_*, \{\boldsymbol{y}_i\}_{i=[n-1]} \left[ \rho(\boldsymbol{y}_{n+1}^\top \boldsymbol{y}_*) \alpha_{c_{\boldsymbol{y}_{n+1}} \boldsymbol{y}_{n+1}, \{\boldsymbol{y}_i\}}(\boldsymbol{y}_*) \right]$$

So $\mathcal{L}(\omega) < \mathcal{L}(\mathbf{M})$. Let

$$q(c_{\boldsymbol{y}_{n+1}}) = \mathbb{E}_{f, \{\boldsymbol{y}_i\}} \left[ \left( f(\boldsymbol{y}_{n+1}) - \frac{\sum_i f(\boldsymbol{y}_i) e^{-\| \boldsymbol{y}_i - c_{\boldsymbol{y}_{n+1}} \boldsymbol{y}_{n+1} \|^2}}{\sum_j e^{-\| \boldsymbol{y}_j - c_{\boldsymbol{y}_{n+1}} \boldsymbol{y}_{n+1} \|^2}} \right)^2 \right].$$

Observe that $\mathcal{L}(\omega) = \mathbb{E}_{\boldsymbol{y}_{n+1}} q(c_{\boldsymbol{y}_{n+1}})$. We might as well set $\omega$ to be such that $c_{\boldsymbol{y}_{n+1}}$ is the same for all $\boldsymbol{y}_{n+1}$ and a minimizer of $q$, so we have $\omega(\boldsymbol{y}_{n+1}) = c \, \boldsymbol{y}_{n+1}$ for all $\boldsymbol{y}_{n+1}$ which implies $\omega = c \mathbf{I}_d$ for some $c$. Because the optimal $\mathbf{M}'$ is identity, the corresponding optimal $\mathbf{M}$ is $\boldsymbol{\Sigma}^{-1}$. $\qquad \square$

## B.1 Rewriting the Loss

As a result of this, we can take $\mathbf{M} = w_{KQ} \boldsymbol{\Sigma}^{-1}$ and write the attention estimator as

$$h_{SA}(\boldsymbol{x}) = \sum_i \frac{f(\boldsymbol{x}_i) e^{-w_{KQ} \| \boldsymbol{\Sigma}^{-1/2} \boldsymbol{x}_i - \boldsymbol{\Sigma}^{-1/2} \boldsymbol{x}_{n+1} \|^2}}{\sum_j e^{-w_{KQ} \| \boldsymbol{\Sigma}^{-1/2} \boldsymbol{x}_j - \boldsymbol{\Sigma}^{-1/2} \boldsymbol{x}_{n+1} \|^2}} \tag{8}$$

This allows us to make the transformation $\mathcal{X} \to \boldsymbol{\Sigma}^{-1/2} \mathcal{X}$. This has the effect of making both the data covariance and the induced function class covariance equal to the identity. Essentially, WLOG we will henceforth consider $\boldsymbol{\Sigma} = \mathbf{I}_d$. Henceforth, the estimator will be taken to be

$$h_{SA}(\boldsymbol{x}) = \sum_i \frac{f(\boldsymbol{x}_i) e^{-w_{KQ} \| \boldsymbol{x}_i - \boldsymbol{x}_{n+1} \|^2}}{\sum_j e^{-w_{KQ} \| \boldsymbol{x}_j - \boldsymbol{x}_{n+1} \|^2}} \tag{9}$$

and the loss will be parameterized by $w_{KQ}$ as

$$\mathcal{L}(w_{KQ}) = \mathbb{E}_{f, \{\boldsymbol{x}_i\}} \left[ \left( \sum_i \frac{(f(\boldsymbol{x}_i) + \epsilon_i) \, e^{-w_{KQ} \| \boldsymbol{x}_i - \boldsymbol{x}_{n+1} \|^2}}{\sum_j e^{-w_{KQ} \| \boldsymbol{x}_j - \boldsymbol{x}_{n+1} \|^2}} - f(\boldsymbol{x}_{n+1}) \right)^2 \right].$$

Because the noise $\epsilon_i$ is independent of everything else, we can decompose this into two terms, a signal term and a noise term as follows

$$\mathcal{L}(w_{KQ}) = \mathbb{E}_{f, \{\boldsymbol{x}_i\}} \underbrace{\left[ \left( \sum_i \frac{(f(\boldsymbol{x}_{n+1}) - f(\boldsymbol{x}_i)) \, e^{-w_{KQ} \| \boldsymbol{x}_i - \boldsymbol{x}_{n+1} \|^2}}{\sum_j e^{-w_{KQ} \| \boldsymbol{x}_j - \boldsymbol{x}_{n+1} \|^2}} \right)^2 \right]}_{\mathcal{L}_{\text{signal}}(w_{KQ})}$$

$$+ \mathbb{E}_{f,\{x_i\}} \underbrace{\left[ \left( \sum_i \frac{\epsilon_i e^{-w_{KQ} \| \, x_i - x_{n+1} \, \|^2}}{\sum_j e^{-w_{KQ} \| \, x_j - x_{n+1} \, \|^2}} - f(x_{n+1}) \right)^2 \right]}_{\mathcal{L}_{\text{noise}}(w_{KQ})}$$

We bound the first term in Appendix C and the second in Appendix D. A useful function that we bound in Lemma G.4 and Corrolary G.5 in Appendix G is

$$g_p(r) = \sum_{i=1}^n \| \, x_i - x \, \|^p e^{-r \| \, x_i^\top - x^2 \, \|}.$$

We will use this function, particularly for $p = 0$ and $1$.

## C   The Signal Term

The purpose of this section of the Appendix is to obtain upper and lower bounds on $\mathcal{L}_{\text{signal}}(w_{KQ})$. Because we work with two different distributions over functions, and because the bounds depend on the distributions, we will make the distribution explicit in the argument to the function

$$\mathcal{L}_{\text{signal}}(w_{KQ}; D(\mathcal{F})) = \mathbb{E}_{f,\{x\}} \left( f(x_i) - \sum_i \frac{f(x_i) e^{-w_{KQ} \| \, x_i - x_{n+1} \, \|^2}}{\sum_j e^{-w_{KQ} \| \, x_j - x_{n+1} \, \|^2}} \right)^2$$

As a reminder, we consider the following two distributions over functions. Please see section B.1 to see why we have set the covariance of $\mathbf{w}$ to be identity.

**Definition C.1** (Affine and 2-ReLU Function Classes). *The function classes $\mathcal{F}_L^{aff}$ and $\mathcal{F}_L^+$ are respectively defined as:*

$$\mathcal{F}_L^{aff} := \{ f : f(x) = l\mathbf{w}^\top x + b, \ \mathbf{w} \in \mathbb{S}^{d-1} \},$$
$$\mathcal{F}_L^+ := \{ f : f(x) = l_1 \, ReLU(\mathbf{w}^\top x) + l_2 \, ReLU(-\mathbf{w}^\top x) + b, \ \mathbf{w} \in \mathbb{S}^{d-1} \}.$$

$D(\mathcal{F}_L^{aff}), D(\mathcal{F}_L^+)$ *are induced by taking* $\mathbf{w} \sim \mathcal{U}^d$, $b, l, l_1, l_2 \sim Unif[-L, L]$.

First we have the following trivial bound on $\mathcal{L}_{\text{signal}}(w_{KQ})$.

**Lemma C.2.** *For all $w_{KQ}$ we have $\mathcal{L}_{signal}(w_{KQ}) \leq 4L^2$.*

*Proof.* We have $\mathcal{L}_{\text{signal}}(w_{KQ}) \leq \mathbb{E} \left[ \left( \sum \frac{f(x_i) - f(x_{n+1})\gamma_i}{\sum \gamma_i} \right)^2 \right]$ for some positive $\{\gamma_i\}$. By Lipschitz-ness, $f(x_i) - f(x_{n+1}) \leq L \| \, x_i - x_{n+1} \, \| \leq 2L$. $\qquad \square$

### C.1   Affine functions

Here we consider the affine function class $\mathcal{F}_L^{\text{aff}}$. First, we note that this class satisfies Assumption B.4.

**Lemma C.3.** *The affine class $\mathcal{F}_L^{aff}$ in Definition 3.2 satisfies Assumption B.4.*

*Proof.*      1. We have $|f(x) - f(y)| = |\mathbf{w}^\top (x - y)| \leq \|\mathbf{w}\| \| \, x - y \, \|$ by Cauchy-Schwarz.

2. Because $b$ is independent of $w$, we have

$$\mathbb{E}_f [f(x)f(y)] = \mathbb{E}_\mathbf{w} \left[ l^2 x^\top \mathbf{w}\mathbf{w}^\top y + b^2 \right] = \mathbb{E} \, l^2 \frac{\mathbb{E}_\mathbf{w} \|\mathbf{w}\|^2}{d} x^\top y + \frac{L^2}{3}.$$

3. $\mathbf{w}$ is isotropic, so $\phi(\mathbf{w})$ is also supported by the distribution on $\mathbf{w}$.

$\qquad \square$

**Lemma C.4.** *For affine functions, the signal term is upper bounded as*

$$
\mathcal{L}_{signal}(w_{KQ}; D(\mathcal{F}_L^{aff})) \leq
\begin{cases}
L^2\, \mathcal{O}\left( \frac{1}{w_{KQ}^2} + \frac{w_{KQ}^{\frac{d}{2}-1}}{n} + \frac{1}{n} \right) & w_{KQ} \geq \frac{d+\sqrt{d}}{2} \\[2ex]
4L^2 & w_{KQ} < \frac{d+\sqrt{d}}{2}
\end{cases}
$$

*Proof.* In the interest of readability, we will denote $\boldsymbol{x}_{n+1}$ as $\boldsymbol{x}$. Consider $\tilde{\boldsymbol{x}}$ such that $\tilde{\boldsymbol{x}} = \sum_i \boldsymbol{x}_i \frac{e^{-2w_{KQ}\,\boldsymbol{x}_i^\top \boldsymbol{x}}}{\sum_j e^{-2w_{KQ}\,\boldsymbol{x}_j^\top \boldsymbol{x}}}$. Then our loss is given by $\mathbb{E}\left[ l^2 \mathbf{w}^\top (\boldsymbol{x} - \tilde{\boldsymbol{x}}) \right]^2$. First, since $\mathbf{w}$ is independent of $\boldsymbol{x}, \{\boldsymbol{x}_i\}$, we have $\mathbb{E}\, l^2 \left( \mathbf{w}^\top (\boldsymbol{x} - \tilde{\boldsymbol{x}}) \right)^2 = \mathbb{E}\, l^2 \mathbf{w}\mathbf{w}^\top (\boldsymbol{x} - \tilde{\boldsymbol{x}})(\boldsymbol{x} - \tilde{\boldsymbol{x}})^\top$, Now $\mathbf{w}$ has a uniformly randomly chosen direction, so its covariance is a multiple of the identity. We have $\mathbb{E}\,\mathrm{Tr}(\mathbf{w}\mathbf{w}^\top) = \mathbb{E}\,\|\mathbf{w}\|^2 = \frac{L^2}{3}$, so $\mathbb{E}\, l^2 \mathbf{w}\mathbf{w}^\top = \frac{L^2}{3d}\mathbf{I}_d$. Continuing, $\mathbb{E}\left( \mathbf{w}^\top (\boldsymbol{x} - \tilde{\boldsymbol{x}}) \right)^2 = \frac{L^2}{3d}\mathbb{E}\,\| \boldsymbol{x} - \tilde{\boldsymbol{x}} \|^2$. Take any $\boldsymbol{x}' \perp \boldsymbol{x}$, we have

$$
\mathbb{E}\,\tilde{\boldsymbol{x}}^\top \boldsymbol{x}' = \mathbb{E}\sum_i \boldsymbol{x}_i^\top \boldsymbol{x}' \, \frac{e^{-2w_{KQ}\,\boldsymbol{x}_i^\top \boldsymbol{x}}}{\sum_j e^{-2w_{KQ}\,\boldsymbol{x}_j^\top \boldsymbol{x}}}
$$

$$
= \mathbb{E}\sum_i \mathbb{E}[\boldsymbol{x}_i^\top \boldsymbol{x}' \mid \boldsymbol{x}_i^\top] \, \frac{e^{-2w_{KQ}\,\boldsymbol{x}_i^\top \boldsymbol{x}}}{\sum_j e^{-2w_{KQ}\,\boldsymbol{x}_j^\top \boldsymbol{x}}} = 0 \qquad \text{iterated expectation and symmetry}
$$

Decomposing $\tilde{\boldsymbol{x}}$ into an orthogonal and a parallel component, we have $\mathbb{E}\,\| \boldsymbol{x} - \tilde{\boldsymbol{x}} \|^2 = \mathbb{E}\,\| \boldsymbol{x} - \boldsymbol{x}\boldsymbol{x}^\top \tilde{\boldsymbol{x}} - \boldsymbol{x}'\boldsymbol{x}'^\top \tilde{\boldsymbol{x}} \|^2$ for some $\boldsymbol{x}' \perp \boldsymbol{x}$ with $\|\boldsymbol{x}'\| = 1$. But

$$
\mathbb{E}\,\| \boldsymbol{x} - \boldsymbol{x}\boldsymbol{x}^\top \tilde{\boldsymbol{x}} - \boldsymbol{x}'\boldsymbol{x}'^\top \tilde{\boldsymbol{x}} \|^2
$$

$$
= \mathbb{E}\,\| \boldsymbol{x}(1 - \boldsymbol{x}^\top \tilde{\boldsymbol{x}}) \|^2 + \mathbb{E}\,\| \boldsymbol{x}'\boldsymbol{x}'^\top \tilde{\boldsymbol{x}} \|^2 - 2\,\mathbb{E}\,\boldsymbol{x}(1 - \boldsymbol{x}^\top \tilde{\boldsymbol{x}})\tilde{\boldsymbol{x}}^\top \boldsymbol{x}' \boldsymbol{x}'^\top
$$

$$
= \mathbb{E}\,\| \boldsymbol{x}(1 - \boldsymbol{x}^\top \tilde{\boldsymbol{x}}) \|^2 + \mathbb{E}\,\| \boldsymbol{x}'\boldsymbol{x}'^\top \tilde{\boldsymbol{x}} \|^2 \qquad \because \boldsymbol{x}^\top \boldsymbol{x}' = 0 \implies 2\,\mathbb{E}\,\boldsymbol{x}(1 - \boldsymbol{x}^\top \tilde{\boldsymbol{x}})\tilde{\boldsymbol{x}}^\top \boldsymbol{x}' \boldsymbol{x}'^\top = 0
\tag{10}
$$

**Case 1:** $w_{KQ} \geq \frac{d+\sqrt{d}}{2}$.

Consider first the term $\mathbb{E}\,\| \boldsymbol{x}(1 - \boldsymbol{x}^\top \tilde{\boldsymbol{x}}) \|^2 = \mathbb{E}(1 - \boldsymbol{x}^\top \tilde{\boldsymbol{x}})^2$. Here we have with probability $1 - \frac{1}{n}$

$$
1 - \boldsymbol{x}^\top \tilde{\boldsymbol{x}} = \frac{\sum (1 - \boldsymbol{x}^\top \boldsymbol{x}_i)e^{-w_{KQ}\| \boldsymbol{x} - \boldsymbol{x}_i \|^2}}{\sum e^{-w_{KQ}\| \boldsymbol{x} - \boldsymbol{x}_i \|^2}} = \frac{g_2(w_{KQ})}{2g_0(w_{KQ})}
$$

$$
\leq \frac{\overline{C_b}\, n \left( \frac{1}{w_{KQ}} \right)^{\frac{d}{2}+1}}{2\underline{C_b}\, n \left( \frac{1}{w_{KQ}} \right)^{\frac{d}{2}}} \leq \frac{\overline{C_b}}{\underline{C_b}} \frac{1}{w_{KQ}} \qquad\qquad \text{Corollary G.5} \tag{11}
$$

The other term $\mathbb{E}\,\| \boldsymbol{x}'\boldsymbol{x}'^\top \tilde{\boldsymbol{x}} \|^2 = \mathbb{E}(\boldsymbol{x}'^\top \tilde{\boldsymbol{x}})^2$ is the component of the bias in the direction orthogonal to $\boldsymbol{x}$.

$$
(\boldsymbol{x}'^\top \tilde{\boldsymbol{x}})^2 = \left( \frac{\sum_i \boldsymbol{x}'^\top \boldsymbol{x}_i\, e^{-w_{KQ}\| \boldsymbol{x}_i - \boldsymbol{x} \|^2}}{\sum_i e^{-w_{KQ}\| \boldsymbol{x}_i - \boldsymbol{x} \|^2}} \right)^2
$$

$$
\leq \left( \frac{\sum_i \boldsymbol{x}'^\top \boldsymbol{x}_i\, e^{-w_{KQ}\| \boldsymbol{x}_i - \boldsymbol{x} \|^2}}{\sum_i e^{-w_{KQ}\| \boldsymbol{x}_i - \boldsymbol{x} \|^2}} \right)^2
$$

$$
\leq \left( \frac{\sum_i \boldsymbol{x}'^\top \boldsymbol{x}_i\, e^{-w_{KQ}\| \boldsymbol{x}_i - \boldsymbol{x} \|^2}}{\sum_i e^{-w_{KQ}\| \boldsymbol{x}_i - \boldsymbol{x} \|^2}} \right)^2
$$

$$
\leq \frac{\sum_i \left( 1 - (\boldsymbol{x}^\top \boldsymbol{x}_i)^2 \right) e^{-2w_{KQ}\| \boldsymbol{x}_i - \boldsymbol{x} \|^2}}{\left( \sum_i e^{-w_{KQ}\| \boldsymbol{x}_i - \boldsymbol{x}_n \|^2} \right)^2} \qquad\qquad \text{Popoviciu's Variance inequality}
$$

$$
\leq \frac{\sum_i 2 \left( 1 - \boldsymbol{x}^\top \boldsymbol{x}_i \right) e^{-2w_{KQ}\| \boldsymbol{x}_i - \boldsymbol{x} \|^2}}{\left( \sum_i e^{-w_{KQ}\| \boldsymbol{x}_i - \boldsymbol{x} \|^2} \right)^2}
$$

$$\leq \frac{\sum_i \|\boldsymbol{x}_i - \boldsymbol{x}\|^2 e^{-2w_{KQ}\|\boldsymbol{x}_i - \boldsymbol{x}\|^2}}{\left(\sum_i e^{-w_{KQ}\|\boldsymbol{x}_i - \boldsymbol{x}\|^2}\right)^2} = \frac{g_2(2w_{KQ})}{g_0^2(w_{KQ})}$$

With probability $1 - \frac{1}{n}$, when $w_{KQ} \geq d + \sqrt{d}$ we have

$$\frac{g_2(2w_{KQ})}{g_0(w_{KQ})^2} \leq \frac{\overline{c_g}n\left(\frac{1}{2w_{KQ}}\right)^{\frac{d}{2}+1}}{\left(\underline{c_g}n\left(\frac{1}{w_{KQ}}\right)^{\frac{d}{2}}\right)^2} \leq \frac{\overline{c_g}w_{KQ}^{\frac{d}{2}-1}}{\underline{c_g}^2 2^{\frac{d}{2}+1}n} \tag{12}$$

Putting together Equations 11 and 12, we have with probability $1 - \frac{1}{n}$,

$$\mathcal{L}_{\text{signal}}(w_{KQ}; D(\mathcal{F}_L)) \leq \mathcal{O}\left(\frac{L^2}{3d}\left(\frac{1}{w_{KQ}} + \frac{w_{KQ}^{\frac{d}{2}-1}}{n}\right)\right).$$

The signal bias is upper bounded by $4L^2$ always (Lemma C.2). The overall upper-bound on the expectation is

$$\mathcal{L}_{\text{signal}}(w_{KQ}; D(\mathcal{F}_L)) \leq \mathcal{O}\left(\frac{L^2}{3d}\left(\frac{1}{w_{KQ}} + \frac{w_{KQ}^{\frac{d}{2}-1}}{n} + 4\right)\right).$$

**Case 2:** $w_{KQ} < \frac{d+\sqrt{d}}{2}$. We always have $\mathcal{L}(w_{KQ}) \leq 4L^2$ from Lemma C.2. $\qquad\square$

**Lemma C.5.** *For affine functions, the signal term is lower bounded as*

$$\mathcal{L}_{signal}(w_{KQ}; D(\mathcal{F}_L^{aff})) \geq \begin{cases} \Omega\left(\frac{L^2}{w_{KQ}^2}\right) & w_{KQ} > \frac{d+\sqrt{d}}{2} \\ \Omega(1) & w_{KQ} < \frac{d+\sqrt{d}}{2} \end{cases}.$$

*Proof.* Similar to Equation (10), for $\tilde{\boldsymbol{x}} = \sum_i \boldsymbol{x}_i \frac{e^{-2w_{KQ}\boldsymbol{x}_i^\top \boldsymbol{x}}}{\sum_j e^{-2w_{KQ}\boldsymbol{x}_i^\top \boldsymbol{x}}}$, we have

$$\mathcal{L}_{\text{signal}}(w_{KQ}; D(\mathcal{F}_L^{\text{aff}})) \geq \frac{L^2}{3d}\mathbb{E}\|\boldsymbol{x}(1 - \boldsymbol{x}^\top \tilde{\boldsymbol{x}})\|^2 = \frac{L^2}{3d}\mathbb{E}(1 - \boldsymbol{x}^\top \tilde{\boldsymbol{x}})^2$$

Now consider the term $1 - \boldsymbol{x}^\top \tilde{\boldsymbol{x}}$. We have

$$\frac{\sum(1 - \boldsymbol{x}^\top \boldsymbol{x}_i)e^{-w_{KQ}\|\boldsymbol{x} - \boldsymbol{x}_i\|^2}}{\sum e^{-w_{KQ}\|\boldsymbol{x} - \boldsymbol{x}_i\|^2}} \geq \frac{g_2(w_{KQ})}{2g_0(w_{KQ})}$$

**Case 1:** $w_{KQ} \geq \frac{d+\sqrt{d}}{2}$. Here we have from Corollary G.5, with probability $1 - 1/n$

$$\frac{\sum(1 - \boldsymbol{x}^\top \boldsymbol{x}_i)e^{-w_{KQ}\|\boldsymbol{x} - \boldsymbol{x}_i\|^2}}{\sum e^{-w_{KQ}\|\boldsymbol{x} - \boldsymbol{x}_i\|^2}} \geq \frac{\underline{C_b}n\left(\frac{1}{w_{KQ}}\right)^{\frac{d}{2}+1}}{2\overline{C_b}n\left(\frac{1}{w_{KQ}}\right)^{\frac{d}{2}}} \geq \frac{\underline{C_b}}{2\overline{C_b}}\frac{1}{w_{KQ}}.$$

With probability $1/n \leq \frac{1}{2}$ the lowest we can have is $\mathcal{L}_{\text{signal}}(w_{KQ}) = 0$, so overall we have

$$\mathcal{L}_{\text{signal}}(w_{KQ}) \geq \frac{L^2}{24d}\left(\frac{\underline{C_b}}{\overline{C_b}}\frac{1}{w_{KQ}}\right)^2$$

**Case 2:** $\frac{d+\sqrt{d}}{4} \leq w_{KQ} \leq \frac{d+\sqrt{d}}{2}$. From Corollary G.5, with probability $1 - \frac{1}{n}$

$$\frac{\sum(1 - \boldsymbol{x}^\top \boldsymbol{x}_i)e^{-w_{KQ}\|\boldsymbol{x} - \boldsymbol{x}_i\|^2}}{\sum e^{-w_{KQ}\|\boldsymbol{x} - \boldsymbol{x}_i\|^2}} \geq \frac{\underline{C_b}n\left(\frac{1}{w_{KQ}}\right)^{\frac{d}{2}+1}}{2\overline{C_b}ne^{-2w_{KQ}}} \geq \frac{\underline{C_b}}{2\overline{C_b}}\frac{e^{2w_{KQ}}}{w_{KQ}^{\frac{d}{2}+1}}.$$

With probability $1/n \leq \frac{1}{2}$ the lowest we can have is $\mathcal{L}_{\text{signal}}(w_{KQ}; D(\mathcal{F}_L^{\text{aff}})) = 0$, so overall we have

$$\mathcal{L}_{\text{signal}}(w_{KQ}; D(\mathcal{F}_L^{\text{aff}})) \geq \frac{L^2}{24d} \left( \frac{\underline{C_b}}{\overline{\overline{C_b}}} \frac{e^{2w_{KQ}}}{w_{KQ}^{\frac{d}{2}+1}} \right)^2$$

**Case 3:** $\frac{d+\sqrt{d}}{4} > w_{KQ}$. From Corollary G.5, with probability $1 - \frac{1}{n}$

$$\frac{\sum(1 - \boldsymbol{x}^\top \boldsymbol{x}_i)e^{-w_{KQ}\|\boldsymbol{x} - \boldsymbol{x}_i\|^2}}{\sum e^{-w_{KQ}\|\boldsymbol{x} - \boldsymbol{x}_i\|^2}} \geq \frac{\underline{C_b}ne^{-4w_{KQ}}}{2\overline{\overline{C_b}}ne^{-2w_{KQ}}} \geq \frac{\underline{C_b}}{2\overline{\overline{C_b}}}e^{-2w_{KQ}}.$$

With probability $1/n \leq \frac{1}{2}$ the lowest we can have is $\mathcal{L}_{\text{signal}}(w_{KQ}; D(\mathcal{F}_L^{\text{aff}})) = 0$, so overall we have

$$\mathcal{L}_{\text{signal}}(w_{KQ}; D(\mathcal{F}_L^{\text{aff}})) \geq \frac{L^2}{24d} \left( \frac{\underline{C_b}}{\overline{\overline{C_b}}}e^{-2w_{KQ}} \right)^2$$

$\square$

**Corollary C.6.** *Combining the above, we have*

$$L^2 \, \mathcal{O}\left( \frac{1}{(w_{KQ} + 1)^2} \right) \leq \mathcal{L}_{signal}(w_{KQ}; D(\mathcal{F}_L^{aff})) \leq L^2 \, \mathcal{O}\left( \frac{1}{w_{KQ}^2} + \frac{w_{KQ}^{\frac{d}{2}-1}}{n} + \frac{1}{n} \right). \qquad (13)$$

We can now perturb these bounds in the case of the ReLU-based function class $\mathcal{F}_L^+$.

## C.2 ReLU-based functions

Consider the function class

$$\mathcal{F}_L^+ = \{l_1 \text{ReLU}(\mathbf{w}^\top \boldsymbol{x}) + l_2 \text{ReLU}(-\mathbf{w}^\top \boldsymbol{x}) + b : \mathbf{w} \in \mathbb{S}^{d-1}, b, l_1, l_2 \in [-L, L]\},$$

where $\text{ReLU}(z) := (z)_+ := \max(z, 0)$. Consider a distributions on $\mathcal{F}_L^+$, namely $D(\mathcal{F}_L^+)$. Let $D(\mathcal{F}_L^+)$ be induced by $\mathbf{w} \sim \mathcal{U}^d, b, l_1, l_2 \sim \text{Unif}[-L, L]$. That is, a vector $\mathbf{w}$ is drawn uniformly on the unit hypersphere. Then two norms are selected, $l_1, l_2$, and the overall function is given by

$$f_{\mathbf{w}, l_1, l_2}(\boldsymbol{x}) = l_1 \text{ReLU}(\mathbf{w}^\top \boldsymbol{x}) + l_2 \text{ReLU}(-\mathbf{w}^\top \boldsymbol{x}) + b,$$

so that it follows one affine rule in one halfspace, and another affine rule in the opposite halfspace. Please see section B.1 to see why we have set the covariance of $\mathbf{w}$ to be identity.

**Lemma C.7.** *The class $\mathcal{F}_L^+$ and distribution $D(\mathcal{F}_L^+)$ defined above satisfy Assumption B.4.*

*Proof.*  1. Each function is defined as being piece-wise $L$-Lipschitz, and it is continuous, so it is also $L-$Lipschitz overall.

2. With probability $1 - 2\frac{\arccos(\boldsymbol{x}^\top \boldsymbol{y})}{\pi}$ the points $\boldsymbol{x}$ and $\boldsymbol{y}$ are such that $(\mathbf{w}^\top \boldsymbol{x})(\mathbf{w}^\top \boldsymbol{y}) < 0$ (that is, they are on opposite sides of the hyperplane defining the two pieces of the ReLU). Because the bias $b$ is independent of the other parameters, we have as in the proof of Lemma C.3

$$\begin{aligned}
\mathbb{E}_f\left[f(\boldsymbol{x})f(\boldsymbol{y})\right] &= \frac{L^2}{3} + \mathbb{E}_\mathbf{w}\left[l_1^2 \boldsymbol{x}^\top \mathbf{w}\mathbf{w}^\top \boldsymbol{y} \big| (\mathbf{w}^\top \boldsymbol{x})(\mathbf{w}^\top \boldsymbol{y}) \geq 0\right] \mathbb{P}[(\mathbf{w}^\top \boldsymbol{x})(\mathbf{w}^\top \boldsymbol{y}) \geq 0] \\
&\quad + \mathbb{E}_\mathbf{w}\left[l_1 l_2 \boldsymbol{x}^\top \mathbf{w}\mathbf{w}^\top \boldsymbol{y} \big| (\mathbf{w}^\top \boldsymbol{x})(\mathbf{w}^\top \boldsymbol{y}) < 0\right] \mathbb{P}[(\mathbf{w}^\top \boldsymbol{x})(\mathbf{w}^\top \boldsymbol{y}) < 0] \\
&= \frac{L^2}{3} + \mathbb{E}_\mathbf{w}\left[l_1^2 \boldsymbol{x}^\top \mathbf{w}\mathbf{w}^\top \boldsymbol{y} \big| \boldsymbol{x}^\top \mathbf{w}\mathbf{w}^\top \boldsymbol{y} > 0\right] \left( 2\frac{\arccos(\boldsymbol{x}^\top \boldsymbol{y})}{\pi} \right) \quad \because l_1 \perp l_2
\end{aligned}$$

Let $\overline{\boldsymbol{x}} = \frac{\boldsymbol{x}}{\|\boldsymbol{x}\|}$ for any vector $\boldsymbol{x}$. Consider a re-parameterization of the pair $(\boldsymbol{x}, \boldsymbol{y})$ as $\xi_\theta(\boldsymbol{x}, \boldsymbol{y}) \to (\overline{\boldsymbol{x} + \boldsymbol{y}}, \overline{\boldsymbol{x} - \boldsymbol{y}})$. Because $\boldsymbol{x}$ and $\boldsymbol{y}$ are on the unit sphere, this is a bijection as

$$\xi_\theta^{-1}(\boldsymbol{x}, \boldsymbol{y}) = \left( \frac{1+\theta}{2} \boldsymbol{x} + \frac{1-\theta}{2} \boldsymbol{y}, \frac{1+\theta}{2} \boldsymbol{x} - \frac{1-\theta}{2} \boldsymbol{y} \right).$$

That is, for any $\boldsymbol{x}, \boldsymbol{y}$, $\xi^{-1}_{\boldsymbol{x}^\top \boldsymbol{y}}(\xi_{\boldsymbol{x}^\top \boldsymbol{y}}(\boldsymbol{x}, \boldsymbol{y})) = (\boldsymbol{x}, \boldsymbol{y})$. The push-forward of $\xi$ is also uniform, that is for $\boldsymbol{x}, \boldsymbol{y}$ satisfying $\boldsymbol{x}^\top \boldsymbol{y} = \theta$, $\xi_\theta(\boldsymbol{x}, \boldsymbol{y})$ is distributed as $\mathcal{U}^d \times \mathcal{U}^{d-1}$. For any $\boldsymbol{x}, \boldsymbol{y}$, let $\xi^{-1}_\theta(\boldsymbol{x}, \boldsymbol{y}) = (\boldsymbol{x}_\theta, \boldsymbol{y}_\theta)$. Then we have $\mathbb{E}_f[f(\boldsymbol{x}_\theta)f(\boldsymbol{y}_\theta)]$ is a decreasing function of $\theta$. Finally, for $\theta \le \theta'$, $L^2 \boldsymbol{x}_\theta^\top \mathbf{w}\mathbf{w}^\top \boldsymbol{y}_\theta > L^2 \boldsymbol{x}_{\theta'}^\top \mathbf{w}\mathbf{w}^\top \boldsymbol{y}_{\theta'}$ so $\boldsymbol{x}_\theta^\top \mathbf{w}\mathbf{w}^\top \boldsymbol{y}_\theta < 0 \implies \boldsymbol{x}_{\theta'}^\top \mathbf{w}\mathbf{w}^\top \boldsymbol{y}_{\theta'} < 0$. The product of two positive increasing functions is itself non-increasing. Since we have both $\mathbb{E}_{\mathbf{w}}\left[L^2 \boldsymbol{x}^\top \mathbf{w}\mathbf{w}^\top \boldsymbol{y} \,\middle|\, \boldsymbol{x}^\top \mathbf{w}\mathbf{w}^\top \boldsymbol{y} > 0\right]$ and $\frac{2\arccos(\boldsymbol{x}^\top \boldsymbol{y})}{\pi}$ are increasing functions of $\boldsymbol{x}^\top \boldsymbol{y}$, we also have

$$\mathbb{E}_{\mathbf{w}}\left[L^2 \boldsymbol{x}^\top \mathbf{w}\mathbf{w}^\top \boldsymbol{y} \,\middle|\, \boldsymbol{x}^\top \mathbf{w}\mathbf{w}^\top \boldsymbol{y} > 0\right]\left(\frac{2\arccos(\boldsymbol{x}^\top \boldsymbol{y})}{\pi}\right)$$

is an increasing function of $\boldsymbol{x}^\top \boldsymbol{y}$ since $\mathbb{E}_{\mathbf{w}}\left[L^2 \boldsymbol{x}^\top \mathbf{w}\mathbf{w}^\top \boldsymbol{y} \,\middle|\, \boldsymbol{x}^\top \mathbf{w}\mathbf{w}^\top \boldsymbol{y} > 0\right] \ge 0$ and $\left(\frac{2\arccos(\boldsymbol{x}^\top \boldsymbol{y})}{\pi}\right) \ge 0$.

3. $\mathbf{w}$ is distributed uniformly on the hypersphere, so $\phi(\mathbf{w})$ is also also distributed uniformly on the hypersphere for any isometry $\phi$ that preserves the origin.

$\square$

**Lemma C.8.** *The signal term is upper bounded as*

$$\mathcal{L}_{signal}(w_{KQ}; D(\mathcal{F}_L^+)) \le \begin{cases} L^2 \mathcal{O}\left(\frac{1}{w_{KQ}} + \frac{1}{n}\right) & w_{KQ} \ge \frac{d+\sqrt{d}}{2} \\ 4L^2 & w_{KQ} < \frac{d+\sqrt{d}}{2} \end{cases}$$

*Proof.* We have

$$\mathcal{L}_{signal}(w_{KQ}; D) = \mathbb{E}_{f, \{\boldsymbol{x}_i\}}\left(\frac{\sum_i (f(\boldsymbol{x}_i) - f(\boldsymbol{x}_n)) e^{-w_{KQ}\|\boldsymbol{x}_i - \boldsymbol{x}_n\|^2}}{\sum_i e^{-w_{KQ}\|\boldsymbol{x}_i - \boldsymbol{x}_n\|^2}}\right)^2$$

$$\le \mathbb{E}_{f, \{\boldsymbol{x}_i\}}\left(\frac{\sum_i L\|\boldsymbol{x}_i - \boldsymbol{x}_n\| e^{-w_{KQ}\|\boldsymbol{x}_i - \boldsymbol{x}_n\|^2}}{\sum_i e^{-w_{KQ}\|\boldsymbol{x}_i - \boldsymbol{x}_n\|^2}}\right)^2$$

$$\le \left(L\frac{g_1(w_{KQ})}{g_0(w_{KQ})}\right)^2$$

With probability $1 - \frac{1}{n}$, when $w_{KQ} \ge \frac{d+\sqrt{d}}{2}$ we have

$$\frac{g_1(w_{KQ})}{g_0(w_{KQ})} \le \frac{\overline{C_b} n \left(\frac{1}{w_{KQ}}\right)^{\frac{d+1}{2}}}{\underline{C_b} n \left(\frac{1}{w_{KQ}}\right)^{\frac{d}{2}}} \le \frac{\overline{C_b}}{\underline{C_b}}\left(\frac{1}{w_{KQ}}\right)^{\frac{1}{2}}$$

We always have $\mathcal{L}_{signal}(w_{KQ}) \le 4L^2$ from Lemma C.2. So the overall upper bound is

$$\mathcal{L}_{signal}(w_{KQ}; D) \le L^2\left(\frac{1}{w_{KQ}} + \frac{4}{n}\right)$$

For $w_{KQ} \ge \frac{d+\sqrt{d}}{2}$, as before, we always have $\mathcal{L}_{signal}(w_{KQ}; D) \le 4L^2$. $\square$

**Lemma C.9.** *The signal term is lower bounded as*

$$\mathcal{L}_{signal}(w_{KQ}; D(\mathcal{F}_L^+)) \ge \mathcal{L}_{signal}(w_{KQ}; D(\mathcal{F}_L^{aff}))/2$$

*Proof.* Again for readability we will write $\boldsymbol{x}_{n+1}$ as $\boldsymbol{x}$. For any $f \in \mathcal{F}_L^+$ let $f_{\boldsymbol{x},\text{aff}}$ denote the corresponding affine function that is equal to $f$ in the halfspace containing $\boldsymbol{x}$, that is if $f(\boldsymbol{x}') = l_1\text{ReLU}(\mathbf{w}^\top \boldsymbol{x}') + l_2\text{ReLU}(-\mathbf{w}^\top \boldsymbol{x}') + b$, and WLOG $\mathbf{w}^\top \boldsymbol{x}' > 0$, then $f_{\boldsymbol{x},\text{aff}}(\boldsymbol{x}') = l_1\mathbf{w}^\top \boldsymbol{x}' + b$. Note that $f_{\boldsymbol{x},\text{aff}}$ comes from a $\mathbf{w}$ selected from the unit sphere and $b, l \in [-L, L]$ exactly as $f \sim$

$D(\mathcal{F}_L)$, so it is actually statistically indistinguishable from a sample from $D(\mathcal{F}_L^{\mathrm{aff}})$, the distribution over affine functions in Definition 3.2 (and the object of Lemma C.5). The error of the nonlinear estimator can be written as

$$\mathbb{E}_{f,\boldsymbol{x},\{\boldsymbol{x}_i\}}\left[\left(\sum_i f(\boldsymbol{x}_i)\gamma_i - f(\boldsymbol{x}_n)\right)^2\right]$$

where $\gamma_i = \dfrac{e^{-w_{KQ}\|\boldsymbol{x}-\boldsymbol{x}_i\|_{\Sigma^{-1}}^2}}{\sum_j e^{-w_{KQ}\|\boldsymbol{x}-\boldsymbol{x}_j\|_{\Sigma^{-1}}^2}}$ Let us compare the two errors due to the two functions. Let $A = \{i : (\boldsymbol{x}_i^\top \mathbf{w})(\boldsymbol{x}^\top \mathbf{w}) < 0\}$ denote the set of points on the opposite side to $\boldsymbol{x}$ of the hyperplane defining the function.

$\mathcal{L}_{\mathrm{signal}}(w_{KQ}; D(\mathcal{F}_L^+))$

$$= \mathbb{E}_{f,\boldsymbol{x},\{\boldsymbol{x}_i\}}\left[\left(\sum_i f(\boldsymbol{x}_i)\gamma_i - f(\boldsymbol{x})\right)^2\right]$$

$$= \mathbb{E}_{f,\boldsymbol{x},\{\boldsymbol{x}_i\}}\left[\left(\sum f_{\boldsymbol{x},\mathrm{aff}}(\boldsymbol{x}_i)\gamma_i + \sum_{i\in A}\left(f(\boldsymbol{x}_i) - f_{\boldsymbol{x},\mathrm{aff}}(\boldsymbol{x}_i)\right)\gamma_i - f(\boldsymbol{x})\right)^2\right]$$

$$= \mathbb{E}_{\boldsymbol{x},\{\boldsymbol{x}_i\}}\,\mathbb{E}_f\left[\left(\sum_{i\notin A} f_{\boldsymbol{x},\mathrm{aff}}(\boldsymbol{x}_i)\gamma_i - f_{\boldsymbol{x},\mathrm{aff}}(\boldsymbol{x})\right)^2\right] + \mathbb{E}_f\left[\left(\sum_{i\in A} f(\boldsymbol{x}_i)\gamma_i\right)^2\right]$$

$$= \mathbb{E}_{\boldsymbol{x},\{\boldsymbol{x}_i\}}\,\mathbb{E}_f\left[\left(\sum_i f_{\boldsymbol{x},\mathrm{aff}}(\boldsymbol{x}_i)\gamma_i - f_{\boldsymbol{x},\mathrm{aff}}(\boldsymbol{x})\right)^2\right] + \mathbb{E}_f\left[\left(\sum_{i\in A} f_{\boldsymbol{x},\mathrm{aff}}(\boldsymbol{x}_i)\gamma_i\right)^2\right]$$

$$- 2\,\mathbb{E}_f\left(\sum f_{\boldsymbol{x},\mathrm{aff}}(\boldsymbol{x}_i)\gamma_i - f_{\boldsymbol{x},\mathrm{aff}}(\boldsymbol{x})\right)\left(\sum_{i\in A} f_{\boldsymbol{x},\mathrm{aff}}(\boldsymbol{x}_i)\gamma_i\right) + \mathbb{E}_f\left[\left(\sum_{i\in A} f(\boldsymbol{x}_i)\gamma_i\right)^2\right]$$

$$\geq \mathbb{E}_{f,\boldsymbol{x},\{\boldsymbol{x}_i\}}\left[\left(\sum f_{\boldsymbol{x},\mathrm{aff}}(\boldsymbol{x}_i)\gamma_i - f_{\boldsymbol{x},\mathrm{aff}}(\boldsymbol{x})\right)^2\right] + \mathbb{E}_{f,\boldsymbol{x},\{\boldsymbol{x}_i\}}\left(\sum_{i\in A} f_{\boldsymbol{x},\mathrm{aff}}(\boldsymbol{x}_i)\gamma_i\right)^2$$

$$- 2\sqrt{\mathbb{E}_{f,\boldsymbol{x},\{\boldsymbol{x}_i\}}\left[\left(\sum f_{\boldsymbol{x},\mathrm{aff}}(\boldsymbol{x}_i)\gamma_i - f_{\boldsymbol{x},\mathrm{aff}}(\boldsymbol{x})\right)^2\right]\mathbb{E}_{f,\boldsymbol{x},\{\boldsymbol{x}_i\}}\left[\left(\sum_{i\in A} f_{\boldsymbol{x},\mathrm{aff}}(\boldsymbol{x}_i)\gamma_i\right)^2\right]}$$

$$+ \mathbb{E}_{f,\boldsymbol{x},\{\boldsymbol{x}_i\}}\left(\sum_{i\in A} f(\boldsymbol{x}_i)\gamma_i\right)^2$$

Here the third equality holds because $f(\boldsymbol{x}_i)$ is independent of $f_{\boldsymbol{x},\mathrm{aff}}(\boldsymbol{x}_j)$ if $i \in A, j \notin A$.

Let $q = \mathbb{E}_{f,\boldsymbol{x},\{\boldsymbol{x}_i\}}\left(\sum_{i\in A} f(\boldsymbol{x}_i)\gamma_i\right)^2 = \mathbb{E}_{f,\boldsymbol{x},\{\boldsymbol{x}_i\}}\left(\sum_{i\in A} f_{\boldsymbol{x},\mathrm{aff}}(\boldsymbol{x}_i)\gamma_i\right)^2$. Then from the above we have

$$\mathbb{E}_{f,\boldsymbol{x},\{\boldsymbol{x}_i\}}\left[\left(\sum_i f(\boldsymbol{x}_i)\gamma_i - f(\boldsymbol{x})\right)^2\right] \geq (\mathcal{L}_{\mathrm{signal}}(w_{KQ}; D(\mathcal{F}_L))(w_{KQ}) - q)^2 + q^2,$$

which has minimum at $q = \mathcal{L}_{\mathrm{signal}}(w_{KQ}; D(\mathcal{F}_L))/2$, completing the proof. $\qquad\square$

## D   Bounds on Noise Variance

In this section we obtain upper and lower bounds on the variance of the estimator due to label noise. There are three relevant parameters: $d$, the ambient dimension of the data; $w_{KQ}$, the scaling induced

by the attention layer; and $n$, the number of tokens. Recall that the noise term is

$$\mathcal{L}_{\text{noise}}(w_{KQ}) = \mathbb{E}_{f,\{\boldsymbol{x}_i\}} \left[ \left( \sum_i \frac{\epsilon_i e^{-w_{KQ}\|\boldsymbol{x}_i - \boldsymbol{x}_{n+1}\|^2}}{\sum_j e^{-w_{KQ}\|\boldsymbol{x}_j - \boldsymbol{x}_{n+1}\|^2}} \right)^2 \right]$$

Because the $\epsilon_i$ are independent, this can further be simplified as

$$\mathcal{L}_{\text{noise}}(w_{KQ}) = \sigma^2 \, \mathbb{E}_{\{\boldsymbol{x}_i\}} \left[ \sum_i \frac{e^{-2w_{KQ}\|\boldsymbol{x}_i - \boldsymbol{x}_{n+1}\|^2}}{\left( \sum_j e^{-w_{KQ}\|\boldsymbol{x}_j - \boldsymbol{x}_{n+1}\|^2} \right)^2} \right]$$

**Lemma D.1.** *The noise term is bounded for $d + \sqrt{d} \leq w_{KQ} \leq \left( \frac{n}{45\sqrt{d}\log n} \right)^{\frac{2}{d}}$ as*

$$\Omega\left( \frac{\sigma^2 w_{KQ}^{\frac{d}{2}}}{n} \right) \leq \mathcal{L}_{noise}(w_{KQ}) \leq \mathcal{O}\left( \frac{\sigma^2 \left( 1 + w_{KQ}^{\frac{d}{2}} \right)}{n} \right).$$

*Proof.* We have

$$\mathcal{L}_{\text{noise}}(w_{KQ}) = \sigma^2 \, \mathbb{E} \left[ \sum_i \frac{e^{-2w_{KQ}\|\boldsymbol{x}_i - \boldsymbol{x}_n\|^2}}{\left( \sum_j e^{-w_{KQ}\|\boldsymbol{x}_j - \boldsymbol{x}_n\|^2} \right)^2} \right] = \sigma^2 \, \mathbb{E} \left[ \frac{g_0(2w_{KQ})}{g_0(w_{KQ})^2} \right].$$

Using Lemma G.5, we have with probability at least $1 - \frac{1}{n}$

$$\frac{g_0(2w_{KQ})}{g_0(w_{KQ})^2} \leq \frac{\overline{c_n} n \left( \frac{1}{w_{2KQ}} \right)^{\frac{d}{2}}}{\left( \underline{c_n} n \left( \frac{1}{w_{KQ}} \right)^{\frac{d}{2}} \right)^2} \leq \frac{\overline{c_n}}{\underline{c_n}^2} \frac{w_{KQ}^{\frac{d}{2}}}{n}$$

and similarly

$$\frac{g_0(2w_{KQ})}{g_0(w_{KQ})^2} \geq \frac{\underline{c_n} n \left( \frac{1}{w_{2KQ}} \right)^{\frac{d}{2}}}{\left( \overline{c_n} n \left( \frac{1}{w_{KQ}} \right)^{\frac{d}{2}} \right)^2} \leq \frac{\underline{c_n}}{\overline{c_n}^2} \frac{w_{KQ}^{\frac{d}{2}}}{n}$$

Finally, in the worst case, we have $0 \leq \mathcal{L}_{\text{noise}}(w_{KQ}) \leq 1$. $\qquad\square$

Finally, we show that the noise term is monotonic in $w_{KQ}$.

**Lemma D.2.** $\mathcal{L}_{noise}(w) > \mathcal{L}_{noise}(w') \iff w > w'$

*Proof.* Let $a_i = e^{-w'\|x_i - x_{n+1}\|^2}, b_i = e^{-(w-w')\|x_i - x_{n+1}\|^2}$. The result follows from Lemma I.3 because $\{a_i\}$ and $\{b_i\}$ satisfy $a_i > a_j \iff b_i > b_j \iff \|x_i - x_{n+1}\| < \|x_j - x_{n+1}\|$. $\qquad\square$

## E   Optimizing the Loss

For the nonlinear function class $\mathcal{F}_L^+$, we have the following.

**Theorem E.1.** *Suppose the functions seen in pretraining are drawn from $D(\mathcal{F}_L^+)$ as in Definition 3.2, the covariates are drawn as Assumption 3.3, $n = \Omega\left( \frac{L\log n}{\sigma} \right)^d$ and $n^{\frac{2}{d+2}} = \Omega(1)$, then the optimal $\mathbf{M}$ satisfies*

$$\mathbf{M} = w_{KQ}\mathbf{I}_d \tag{14}$$

*where $w_{KQ}$ satisfies*

$$\Omega\left( (nL^2)^{\frac{1}{d+2}} \right) \leq w_{KQ} \leq \mathcal{O}\left( (nL^2)^{\frac{2}{d+2}} \right). \tag{15}$$

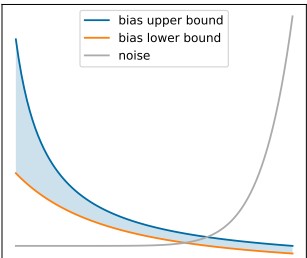 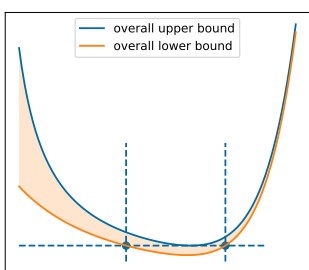

Figure 7: **Left:** Rough upper and lower bounds for the bias term (shaded region), along with the noise variance (gray). **Right:** Overall upper and lower bound for the in-context loss. The horizontal dashed line establishes an upper bound for the optimal loss, while the vertical dashed lines establish lower and upper bounds for the parameter $w_{KQ}$ that can attain the optimal loss.

*Proof.* We consider three regions in which the optimal value could potentially lie and see that only the third region is viable.

**Case 1.** $w_{KQ} \leq d + \sqrt{d}$: In this case, the signal term lower bounds the optimal loss by Lemma C.5 as $\Omega(1)$.

**Case 2.** $w_{KQ} > \Omega\left(\frac{n}{\log n}\right)^{\frac{2}{d}}$. In this case, the noise term lower bounds the optimal loss. From Lemma D.2 we know that the noise term is non-decreasing in $w_{KQ}$ so in the range $w_{KQ} > \Omega\left(\left(\frac{n}{\log n}\right)^{\frac{2}{d}}\right)$ is lower bounded by $\mathcal{L}_{\text{noise}}(w_{KQ})$ at $w_{KQ} = \Omega\left(\left(\frac{n}{\log n}\right)^{\frac{2}{d}}\right)$, which is $\Omega\left(\frac{\sigma^2}{\log n}\right)$.

**Case 3.** $d + \sqrt{d} \leq w_{KQ} \leq \Omega\left(\frac{n}{\log n}\right)^{\frac{2}{d}}$ By combining Lemmas C.8, C.9, and D.1 , we obtain the following overall bound on the loss:

$$\underline{c}\left(\frac{L^2}{(w_{KQ}+1)^2} + \frac{\sigma^2 w_{KQ}^{\frac{d}{2}}}{n}\right) \leq \mathcal{L}(w_{KQ}) \leq \overline{c}\left(\frac{L^2}{w_{KQ}} + \sigma^2\frac{w_{KQ}^{\frac{d}{2}}}{n} + \frac{\sigma^2 + L^2}{n}\right)$$

for some constants $\overline{c}, \underline{c}$ that only depend on $d$. In the range $w_{KQ} \geq d + \sqrt{d}$, we have $w_{KQ} > 1$ and $w_{KQ} \leq n$, so the upper bound can be relaxed as $\mathcal{L}_{\text{noise}}(w_{KQ}) \leq 2\overline{c}\left(\frac{L^2}{n} + \frac{\sigma^2 w_{KQ}^{\frac{d}{2}}}{n}\right)$, which is minimized at $w_{KQ} = \left(\frac{nL^2}{\sigma^2 d}\right)^{\frac{2}{d+2}}$. Here it is upper bounded by $4\overline{c}\left(\frac{dL^d\sigma^2}{n}\right)^{\frac{2}{d+2}}$. We note first of all that for large enough $n$ (as long as $n = \Omega\left(\frac{\sigma \log n}{L}\right)^d$ and $n^{\frac{2}{d+2}} = \Omega(1)$) this is lower than the lower bounds we got in **Case 1** and **Case 2**, so this is indeed the region of global optimal solution. From Lemma C.9 we have $\mathcal{L}_{\text{noise}}(w_{KQ}) \geq \frac{L^2}{w_{KQ}^2} + \sigma^2\frac{w_{KQ}^{\frac{d}{2}}}{n} \geq \frac{L^2}{w_{KQ}^2}$ which gives

$$\underline{c}\frac{L^2}{w_{KQ}^2} \leq \mathcal{L}_{\text{noise}}(w_{KQ}) \leq 4\overline{c}L^2\left(\frac{\sigma^2 d}{nL^2}\right)^{\frac{2}{d+2}}$$

$$\implies \left(\frac{nL^2}{d\sigma^2}\right)^{\frac{1}{d+2}}\sqrt{\frac{\underline{c}}{4\overline{c}}} \leq w_{KQ}$$

for the upper bound, we similarly also have $\mathcal{L}_{\text{noise}}(w_{KQ}) \geq \frac{L^2}{w_{KQ}^2} + \sigma^2\frac{w_{KQ}^{\frac{d}{2}}}{n} \geq \sigma^2\frac{w_{KQ}^{\frac{d}{2}}}{n}$ which gives

$$4\overline{c}\left(\frac{dL^d\sigma^2}{n}\right)^{\frac{2}{d+2}} \geq \sigma^2\frac{w_{KQ}^{\frac{d}{2}}}{n}$$

$$\implies w_{KQ} \le \left(\frac{nL^2}{\sigma^2}\right)^{\frac{2}{d+2}} \left(4\frac{\overline{c}}{\underline{c}} d^{\frac{2}{d+2}}\right)^{\frac{2}{d}}$$

Of course, for this to not be vacuous we need

$$\left(\frac{nL^2}{\sigma^2}\right)^{\frac{2}{d+2}} \left(4\frac{\overline{c}}{\underline{c}} d^{\frac{2}{d+2}}\right)^{\frac{2}{d}} \le \left(\frac{1}{45\sqrt{d}}\frac{n}{\log n}\right)^{\frac{2}{d}}.$$

We will again hide constants that depend only on $d$ and write this as

$$c_1 \left(\frac{nL^2}{\sigma^2}\right)^{\frac{2}{d+2}} \le c_2 \left(\frac{n}{\log n}\right)^{\frac{2}{d}}$$

which is true as long as $n > \left(\frac{L\log n}{\sigma}\right)^d$. $\qquad\square$

For the affine function class $\mathcal{F}_L^{\text{aff}}$, we have the following

**Theorem E.2.** *If the functions seen in pretraining are drawn from $D(\mathcal{F}_L^{\text{aff}})$ as in Definition 3.2, and the noise variance $\sigma^2$ and Liphscitz constant $L$ satisfies $n \ge \left(\frac{L\log^2 n}{\sigma}\right)^{d+2}$, and $n^{\frac{2}{d}} \ge \Omega(1)$, and the covariates are drawn as Assumption 3.3, the optimal $\mathbf{M}$ satisfies*

$$\mathbf{M} = w_{KQ}\mathbf{I}_d \tag{16}$$

*where $w_{KQ}$ satisfies*

$$\Omega\left((nL^2)^{\frac{1}{d+4}}\right) \le w_{KQ} \le \mathcal{O}\left((nL^2)^{\frac{2(d+2)}{d(d+4)}}\right). \tag{17}$$

*Proof.* Again we work with three cases.

**Case 1.** $w_{KQ} \le d + \sqrt{d}$. Again in this case we have a lower bound to the signal term of $\Omega(1)$.

**Case 2.** $w_{KQ} \ge \Omega\left(\frac{n}{\log n}\right)^{\frac{2}{d}}$. Again we have a lower bound of $\Omega\left(\frac{\sigma^2}{\log n}\right)$

**Case 3.** $d + \sqrt{d} \le w_{KQ} \le \Omega\left(\frac{n}{\log n}\right)^{\frac{2}{d}}$ Combining Lemmas C.4, C.5, D.1 is

$$\underline{c}\left(\frac{L^2}{(w_{KQ}+1)^2} + \sigma^2\frac{w_{KQ}^{\frac{d}{2}}}{n}\right) \le \mathcal{L}(w_{KQ}) \le \overline{c}\left(\frac{L^2}{w_{KQ}^2} + \sigma^2\frac{w_{KQ}^{\frac{d}{2}}}{n} + L^2\frac{w_{KQ}^{\frac{d}{2}-1}}{n} + \frac{L^2 + \sigma^2}{n}\right)$$

We will minimize the upper bound. First suppose $\frac{L^2}{\sigma^2} \ge w_{KQ}$ for the $w_{KQ}$ that minimizes the upper bound. Then we have

$$\mathcal{L}(w_{KQ}) \le \overline{c}\left(\frac{L^2}{w_{KQ}^2} + \frac{\sigma^2}{n} + 2L^2\frac{w_{KQ}^{\frac{d}{2}-1}}{n}\right)$$

This upper bound is minimized at $w_{KQ} = n^{\frac{2}{d+2}}$. However, this contradicts the constraint that $w_{KQ} \le \frac{L^2}{\sigma^2}$, when $n^{\frac{2}{d+2}} \ge \frac{L^2}{\sigma^2}$, as we assume. So we have $w_{KQ} \ge \frac{L^2}{\sigma^2}$ for the minimizer. This means the upper bound is no more than

$$\mathcal{L}(w_{KQ}) \le \overline{c}\left(\frac{L^2}{w_{KQ}^2} + \sigma^2\frac{2w_{KQ}^{\frac{d}{2}}}{n} + \frac{\sigma^2 + L^2}{n}\right)$$

This upper bound is minimized at $w_{KQ} = \left(\frac{nL^2}{\sigma^2 d}\right)^{\frac{2}{d+4}}$ where it is upper bounded by

$$\mathcal{L}_{\text{noise}}(w_{KQ}) \le 4L^2\overline{c}\left(\frac{\sigma^2 d}{nL^2}\right)^{\frac{2}{d+4}} + \frac{L^2}{n} \le 5L^2\overline{c}\left(\frac{\sigma^2 d}{nL^2}\right)^{\frac{2}{d+4}}.$$

whenever $n \geq \frac{L^2}{\sigma^2}$. We see that

$$\mathcal{L}(w_{KQ}) \geq \underline{c} \left( \frac{L^2}{w_{KQ}^2} + \sigma^2 \frac{w_{KQ}^{\frac{d}{2}}}{n} \right) \geq \underline{c} \frac{L^2}{w_{KQ}^2}$$

$$\implies \underline{c} \frac{L^2}{w_{KQ}^2} \leq 5L^2 \bar{c} \left( \frac{\sigma^2 d}{nL^2} \right)^{\frac{2}{d+4}}$$

$$\implies \left( \frac{nL^2}{\sigma^2} \right)^{\frac{1}{d+4}} \sqrt{\frac{\underline{c}}{5\bar{c}}} \left( \frac{1}{d} \right)^{\frac{1}{d+4}} \leq w_{KQ}$$

for the upper bound, we similarly also have

$$\mathcal{L}(w_{KQ}) \geq \underline{c} \left( \frac{L'^2}{w_{KQ}^2} + \sigma^2 \frac{w_{KQ}^{\frac{d}{2}}}{n} \right) \geq \underline{c} \sigma^2 \frac{w_{KQ}^{\frac{d}{2}}}{n}$$

$$\implies w_{KQ} \leq \left( \frac{nL^2}{\sigma^2} \right)^{\frac{2(d+2)}{d(d+4)}} \left( 5 \frac{\bar{c}}{\underline{c}} d^{\frac{2}{d+4}} \right)^{\frac{2}{d}}$$

Of course, for this to not be vacuous we need

$$\left( \frac{nL^2}{\sigma^2} \right)^{\frac{2(d+2)}{d(d+4)}} \left( 5 \frac{\bar{c}}{\underline{c}} d^{\frac{2}{d+4}} \right)^{\frac{2}{d}} \leq \left( \frac{1}{45\sqrt{d}} \frac{n}{\log n} \right)^{\frac{2}{d}}.$$

We will again hide constants that depend only on $d$ and write this as

$$c_1 \left( \frac{nL^2}{\sigma^2} \right)^{\frac{2(d+2)}{d(d+4)}} \leq c_2 \left( \frac{n}{\log n} \right)^{\frac{2}{d}}$$

which again is true as long as $n = \Omega \left( \frac{L \log^2 n}{\sigma} \right)^{d+2}$ □

### E.1 Generalization Bounds

We conclude this section with a proof of the generalization error on a new $L-$Lipschitz task.

**Theorem E.3.** *Suppose our attention is first pretrained on tasks drawn from $D(\mathcal{F}_L^+)$ and then tested on an arbitrary $L-$Lipschitz task, then the loss on the new task is upper bounded as $\mathcal{L} \leq \mathcal{O} \left( \frac{L^2}{\Lambda^\beta} \right)$. Furthermore, if the new task is instead drawn from $D(\mathcal{F}_{L'}^+)$, the loss is lower bounded as $\mathcal{L} \geq \min\{\Omega(\frac{L'^2}{\Lambda^{2\beta}}), \Omega(\frac{\Lambda^{\beta d/2}}{n})\}$*

*Proof.* We know from Theorem E.2 that $\Omega(\Lambda^\beta) \leq w_{KQ} \leq \mathcal{O}(\Lambda^{2\beta})$. The upper bound for $\mathcal{L}(w_{KQ})$, which is $\mathcal{O}(\frac{L^2}{w_{KQ}} + \frac{w_{KQ}^{\frac{d}{2}}}{n})$, is a convex function for $d \geq 2$, so in any range it attains its maximum value at the extreme points. We can check the cases to see that this is $\mathcal{O}(\max\{\frac{L^2}{\Lambda^\beta} + \frac{\Lambda^{d\beta/2}}{n}, \frac{L^2}{\Lambda^{2\beta}} + \frac{\Lambda^{d\beta}}{n}\}) = \mathcal{O}(\frac{L^2}{\Lambda^\beta} + \frac{\Lambda^{d\beta/2}}{n} + \frac{L^2}{\Lambda^{2\beta}} + \frac{\Lambda^{d\beta}}{n}) = \mathcal{O}(\frac{L^2}{\Lambda^\beta})$ for large enough $n$.

Now consider testing on a new task from $D(F_{L'}^+)$. The ICL loss for $\Omega(\Lambda^\beta) \leq w_{KQ} \leq \mathcal{O}(\Lambda^{2\beta})$ is bounded below as $\Omega(\frac{L'^2}{\Lambda^{2\beta}})$ and $\Omega(\frac{\Lambda^{\beta d/2}}{n})$. □

The implication of this is that if $L' \gg L$, the error scales as $(L')^2$ rather than $(L')^{\frac{2d}{d+2}}$ while for $L' \ll L$, the error is lower bounded by a constant.

# F  Lower Bound for Linear Attention

In this section we prove Theorem 3.6.

**Lemma F.1.** *Consider the function distributions $D(\mathcal{F}_L)$ and $D(\mathcal{F}_L^+)$ described in Definition 3.2. We have $\mathcal{L}_{LA} \geq \Omega(L^2)$, that is, the ICL error is lower bounded as $\Omega(L^2)$.*

*Proof.* We start by decomposing the ICL loss into a bias dependent term and a cenetered term. For $f \in \mathcal{F}_L \in \{\mathcal{F}_L^{\text{aff}}, \mathcal{F}_L^+\}$, let $\overline{f}$ denote the centered function $f - \mathbb{E}_{\boldsymbol{x}} f$. Let $f'$ denote the flip of $f$ about its expected value, so $f' = \mathbb{E}_x f - \overline{f}$. We observe that $\overline{f}$ is independent of $\mathbb{E}_x f$. For linear attention, we have, for $f \sim D(\mathcal{F}_L)$

$$\mathcal{L}_{\text{LA}}(\mathbf{M}) = \mathbb{E}_{f, \{\boldsymbol{x}_i\}_i, \{\epsilon_i\}_i}\left[\left(h_{LA}(\boldsymbol{x}_{n+1}) - f(\boldsymbol{x}_{n+1})\right)^2\right]$$

$$= \mathbb{E}_{f, \{\boldsymbol{x}_i\}_i, \{\epsilon_i\}_i}\left[\left(\sum_{i=1}^n \left((f(\boldsymbol{x}_i) + \epsilon_i)\boldsymbol{x}_i^\top \mathbf{M}\boldsymbol{x}_{n+1}\right) - f(\boldsymbol{x}_{n+1})\right)^2\right]$$

$$= \mathbb{E}_{f, \{\boldsymbol{x}_i\}_i, \{\epsilon_i\}_i}\left[\left(\sum_{i=1}^n \left(\overline{f}(\boldsymbol{x}_i)\boldsymbol{x}_i^\top \mathbf{M}\boldsymbol{x}_{n+1} + \epsilon_i\boldsymbol{x}_i^\top \mathbf{M}\boldsymbol{x}_{n+1} + \mathbb{E}_x f\boldsymbol{x}_i^\top \mathbf{M}\boldsymbol{x}_{n+1}\right) - f(\boldsymbol{x}_{n+1})\right)^2\right]$$

$$= \mathbb{E}_{f, \{\boldsymbol{x}_i\}_i, \{\epsilon_i\}_i}\left[\left(\sum_{i=1}^n \left(\overline{f}(\boldsymbol{x}_i)\boldsymbol{x}_i^\top \mathbf{M}\boldsymbol{x}_{n+1} + \epsilon_i\boldsymbol{x}_i^\top \mathbf{M}\boldsymbol{x}_{n+1}\right) - f(\boldsymbol{x}_{n+1})\right)^2\right] \tag{18}$$

$$+ \mathbb{E}_{f, \boldsymbol{x}, \{\boldsymbol{x}_i\}}\left[\left(\sum_i \left(\mathbb{E}_x f\right)\boldsymbol{x}_i^\top \mathbf{M}\boldsymbol{x}_{n+1}\right)^2\right]$$

$$\geq \mathbb{E}_{f, \{\boldsymbol{x}_i\}_i, \{\epsilon_i\}_i}\left[\left(\sum_{i=1}^n \left(\overline{f}(\boldsymbol{x}_i)\boldsymbol{x}_i^\top \mathbf{M}\boldsymbol{x}_{n+1} + \epsilon_i\boldsymbol{x}_i^\top \mathbf{M}\boldsymbol{x}_{n+1}\right) - \overline{f}(\boldsymbol{x}_{n+1}) - \mathbb{E}_x f\right)^2\right]$$
$$\tag{19}$$

By symmetry, this is also equal to the same expression using $f'$ instead of $f$, since $f$ and $f'$ are distributed identically. Besides, $\mathbb{E}_x f = \mathbb{E}_x f'$ and $\epsilon$ is symmetric about the origin, so

$$\mathcal{L}_{\text{LA}}(\mathbf{M}) \geq \mathbb{E}_{f, \{\boldsymbol{x}_i\}_i, \{\epsilon_i\}_i}\left[\left(\sum_{i=1}^n \left(f'(\boldsymbol{x}_i)\boldsymbol{x}_i^\top \mathbf{M}\boldsymbol{x}_{n+1} + \epsilon_i\boldsymbol{x}_i^\top \mathbf{M}\boldsymbol{x}_{n+1}\right) - f'(\boldsymbol{x}_{n+1}) - \mathbb{E}_x f'\right)^2\right]$$

$$= \mathbb{E}_{f, \{\boldsymbol{x}_i\}_i, \{\epsilon_i\}_i}\left[\left(\sum_{i=1}^n \left(f'(\boldsymbol{x}_i)\boldsymbol{x}_i^\top \mathbf{M}\boldsymbol{x}_{n+1} - \epsilon_i\boldsymbol{x}_i^\top \mathbf{M}\boldsymbol{x}_{n+1}\right) - f'(\boldsymbol{x}_{n+1}) - \mathbb{E}_x f\right)^2\right]$$

$$= \mathbb{E}_{f, \{\boldsymbol{x}_i\}_i, \{\epsilon_i\}_i}\left[\left(-\left(\sum_{i=1}^n \left(\overline{f}(\boldsymbol{x}_i)\boldsymbol{x}_i^\top \mathbf{M}\boldsymbol{x}_{n+1} + \epsilon_i\boldsymbol{x}_i^\top \mathbf{M}\boldsymbol{x}_{n+1}\right) - \overline{f}(\boldsymbol{x}_{n+1})\right) - \mathbb{E}_x f\right)^2\right]$$

Let $A = \sum_{i=1}^n \left(\overline{f}(\boldsymbol{x}_i)\boldsymbol{x}_i^\top \mathbf{M}\boldsymbol{x}_{n+1} + \epsilon_i\boldsymbol{x}_i^\top \mathbf{M}\boldsymbol{x}_{n+1}\right) - \overline{f}(\boldsymbol{x}_{n+1})$ and $B = \mathbb{E}_x f$. Then we see that $\mathcal{L}_{LA}(\mathbf{M}) \geq \frac{1}{2}\mathbb{E}(A+B)^2 + \frac{1}{2}\mathbb{E}(-A+B)^2 = \mathbb{E}\,A^2 + \mathbb{E}\,B^2$. Meanwhile, $\mathbb{E}\left(\mathbb{E}_x f\right)^2$ is just the variance of the signal term in $D(\mathcal{F}_L^{\text{aff}})$ or $D(\mathcal{F}_L^+)$, which is $\frac{L^2}{3}$. So $\mathcal{L}_{LA}(\mathbf{M}) \geq \frac{L^2}{3}$ $\qquad\square$

# G  Bounds for $g_p(r)$

The purpose of this section is to obtain upper and lower bounds on

$$g_p(r) = \sum_{i=1}^n \|\boldsymbol{x}_i - \boldsymbol{x}\|^p e^{-r\|\boldsymbol{x}_i^\top - \boldsymbol{x}\|^2}$$

for $p = 0, 1/2, 1$. For this, we will need high probability upper and lower bounds on the number of points in a spherical cap under a uniform distribution over the hypersphere. Consider $n$ points $\{x_i\}$ drawn uniformly from $\sigma_{d-1}$, the uniform measure over $S_{d-1}$, the $d-$dimensional hypersphere. The measure of the $\epsilon-$ spherical cap around $x \in S_{d-1}$, $C(\epsilon, x) = \{x' : x'^\top x > 1 - \epsilon\}$ is denoted by $\sigma_\epsilon$.

## G.1 Bounds on Spherical Caps

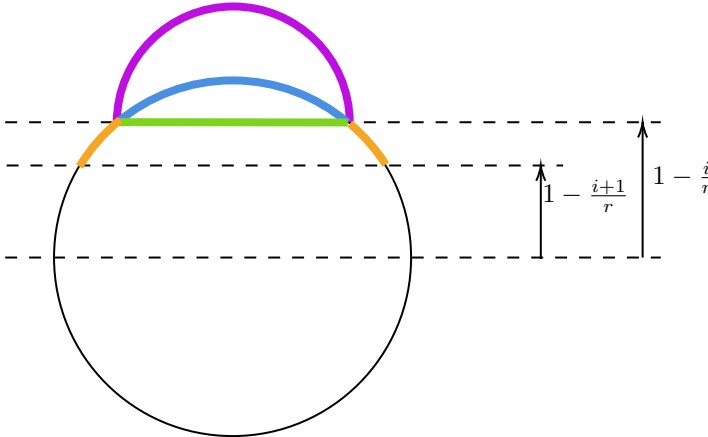

Figure 8: The surface area of the purple hemisphere is used to upper bound the surface area of $C(\frac{i}{r})$, while the *volume* of the green hypersphere is used as a lower bound. Points in the orange region are $S_{i+1} \setminus S_i$, and their count is $N_{i+1} - N_i$.

**Lemma G.1.** *The area of the spherical cap $C(\epsilon)$, $\sigma_\epsilon$ is bounded as*

$$\frac{(2\epsilon - \epsilon^2)^{\frac{d-1}{2}}}{\sqrt{2d\pi}} \leq \sigma_\epsilon \leq (2\epsilon - \epsilon^2)^{\frac{d}{2}} \leq (2\epsilon)^{\frac{d-1}{2}} e^{-\epsilon d/4}$$

*Proof.* We derive a lower bound as follows. We replace the surface area of a spherical cap in $S_{d-1}$ with a $d-1$ dimensional ball of the same boundary. Let $V_d$ denote the volume of a $d$ dimensional ball (that is, $V_3(r) = \frac{4}{3}\pi r^3$), and let $A_d$ denote the surface area of a $d$ dimensional sphere (so $A_3(a) = 4\pi r^2$). It is known that

$$V_d(r) = \frac{\pi^{\frac{d}{2}}}{\Gamma(\frac{d}{2} + 1)} r^d, \text{ and } A_d(r) = \frac{2\pi^{\frac{d}{2}}}{\Gamma(\frac{d}{2})} r^{d-1}.$$

Then we have

$$\sigma_\epsilon \geq \frac{V_{d-1}\left((1 - (1-\epsilon)^2)^{\frac{1}{2}}\right)}{A_d(1)}$$

$$= \frac{(1 - (1-\epsilon)^2)^{\frac{d-1}{2}}}{2\sqrt{\pi}} \frac{\Gamma(\frac{d}{2})}{\Gamma(\frac{d+1}{2})}$$

$$\geq \frac{(1 - (1-\epsilon)^2)^{\frac{d}{2}}}{\sqrt{d\pi}} \qquad\qquad \text{Lemma G.6}$$

$$= \frac{(2\epsilon - \epsilon^2)^{\frac{d-1}{2}}}{\sqrt{2d\pi}}$$

The upper bound is similar. This time we replace the cap with the surface of a hemisphere with the same boundary. We have

$$\sigma_\epsilon \leq \frac{A_d\left((1 - (1-\epsilon)^2)^{\frac{1}{2}}\right)}{2A_d(1)} = \frac{(1 - (1-\epsilon)^2)^{\frac{d-1}{2}}}{2} \leq (2\epsilon - \epsilon^2)^{\frac{d-1}{2}}$$

$\square$

We will also need upper and lower bounds on a discretized version of the incomplete gamma function.

**Definition G.2.** *Denote by $\gamma(d, \alpha, m)$ the expression $\gamma(d, \alpha, m) = \sum_{i=1}^{m} i^d e^{-\alpha i}$.*

We have the following

**Lemma G.3.** *For $d > 5, 1 \leq \alpha \leq 2$, the incomplete Gamma function is bounded as*

$$
\begin{cases}
m^d e^{-\alpha m - 1/2} \leq \gamma(d, \alpha, m) \leq m^{d+1} e^{-\alpha m - 1/2} & m < d + \sqrt{d} \\
\frac{\Gamma(d+1)}{2\alpha^{d+1}} \leq \gamma(d, \alpha, m) \leq \frac{2\Gamma(d+1)}{\alpha^{d+1}} & m \geq d + \sqrt{d}
\end{cases}
$$

*Proof.* We compare with the Gamma function

$$
\Gamma(d+1) = \int_0^\infty t^d e^{-t} dt.
$$

Note that $\int_0^\infty t^d e^{-\alpha t} dt = \frac{1}{\alpha^{d+1}} \int_0^\infty t^d e^{-t} dt = \frac{1}{\alpha^{d+1}} \Gamma(d+1)$. Because the function $t^d e^{-\alpha t}$ is uni-modal with maximum $\left(\frac{d}{\alpha e}\right)^d$, we have from Lemma I.1

$$
\sum_{i=1}^{m} i^d e^{-\alpha i} + \left(\frac{d}{\alpha e}\right)^d + \sum_{i=m}^{\infty} i^d e^{-\alpha i} \geq \int_0^\infty t^d e^{-\alpha t} dt = \frac{1}{\alpha^{d+1}} \Gamma(d+1)
$$

Now suppose $m \geq \frac{d+\sqrt{d}}{\alpha}$. Then we have

$$
\sum_{i=m}^{\infty} i^d e^{-\alpha i} \leq \sum_{i=\frac{d+\sqrt{d}}{\alpha}}^{\infty} i^d e^{-\alpha i}
$$

$$
= \sum_{i=\frac{d+\sqrt{d}}{\alpha}}^{\infty} \left(\frac{d+\sqrt{d}}{\alpha}\right)^d e^{-(d+\sqrt{d})} \prod_{j=0}^{i-\frac{d+\sqrt{d}}{\alpha}} \left[\frac{1}{e^\alpha} \left(\frac{\frac{d+\sqrt{d}}{\alpha}+j+1}{\frac{d+\sqrt{d}}{\alpha}+j}\right)^d\right]
$$

$$
\leq \sum_{i=\frac{d+\sqrt{d}}{\alpha}}^{\infty} \left(\frac{d+\sqrt{d}}{\alpha}\right)^d e^{-(d+\sqrt{d})} \prod_{j=0}^{i-\frac{d+\sqrt{d}}{\alpha}} \left[\frac{1}{e^\alpha} \left(\frac{\frac{d+\sqrt{d}}{\alpha}+1}{\frac{d+\sqrt{d}}{\alpha}}\right)^d\right]
$$

$$
\leq \sum_{i=\frac{d+\sqrt{d}}{\alpha}}^{\infty} \left(\frac{d+\sqrt{d}}{\alpha}\right)^d e^{-(d+\sqrt{d})} \left(e^{-\frac{\alpha\sqrt{d}}{d+\sqrt{d}}}\right)^{i-\frac{d+\sqrt{d}}{\alpha}}
$$

$$
= \left(\frac{d+\sqrt{d}}{\alpha}\right)^d e^{-(d+\sqrt{d})} \frac{1}{1 - e^{-\alpha\sqrt{d}/(d+\sqrt{d})}} \leq \left(\frac{d}{\alpha e}\right)^d \frac{2\sqrt{d}}{\alpha}
$$

the first inequality follows because $\frac{d+\sqrt{d}}{\alpha} \leq m$, the second follows because $\frac{2d+1}{2d} \geq \frac{2d+j+1}{2d+j}$, the last follows because $\left(1 + \frac{\sqrt{d}}{d\alpha}\right)^d \leq e^{\frac{\sqrt{d}}{\alpha}}$ and $\frac{1}{1-e^{x-x}} \leq 2x$ for $x \leq 2$. Over all, we have

$$
\sum_{i=1}^{m} i^d e^{-i} + \left(2\frac{\sqrt{d}}{\alpha} + 1\right) \left(\frac{d}{\alpha e}\right)^d \geq \int_0^\infty t^d e^{-t} dt = \frac{\Gamma(d+1)}{\alpha^{d+1}}
$$

While for the upper bound we have

$$
\sum_{i=1}^{m} i^d e^{-\alpha i} - \left(\frac{d}{\alpha e}\right)^d \leq \int_0^\infty t^d e^{-\alpha t} dt = \frac{\Gamma(d+1)}{\alpha^{d+1}}
$$

Finally, we use Lemma G.3, specifically that $\left(\frac{d}{\alpha e}\right)^d \leq \frac{1}{\alpha^{d+1}} \sqrt{2\pi d} \left(\frac{d}{e}\right)^d \leq \frac{\Gamma(d+1)}{\alpha^{d+1}}$ to yield the desired result.

For $m < \frac{d + \sqrt{d}}{\alpha}$, we have from Lemma G.7 that $m^d e^{-\alpha m} \geq \frac{1}{\sqrt{e}} i^d e^{-\alpha i}$ so

$$\sum_{i=0}^{m} i^d e^{-\alpha i} \geq m^d e^{-\alpha m - \frac{1}{2}}$$

and

$$\sum_{i=0}^{m} i^d e^{-\alpha i} \leq m^{d+1} e^{-\alpha m - \frac{1}{2}}$$

$\square$

## G.2 Bounds on $g_p(r)$

**Lemma G.4.** *Suppose $\{x_i\}$ are drawn independently and uniformly from the unit hypersphere. For $\frac{n}{\log n} \geq 45\sqrt{d} r^{\frac{d}{2}}, n > 5, d > 2, p \leq 2$, we have $g_p(r) = \sum_{i=1}^{n} \| x_i - x \|^p e^{-r \| x_i^\top - x \|^2}$ satisfies*

$$(1 - e^{\frac{p}{2} - 2}) \frac{n 2^{\frac{p}{2}}}{\sqrt{8 e^4 \pi d}} \left( \frac{1}{r} \right)^{\frac{d}{2} + \frac{p}{2}} \gamma(\frac{d}{2} + \frac{p}{2}, 2, r) \leq g_p(r) \leq 3n \left( \frac{2}{r} \right)^{\frac{d}{2} + \frac{p}{2}} \gamma(\frac{d}{2} + \frac{p}{2}, 2, r)$$

*with probability at least $1 - \frac{1}{2n}$*

*Proof.* For $0 \leq i \leq r$ let $N_i$ denote the number, and $S_i$ denote the set, of points satisfying $1 - \frac{i}{r} \leq x_i^\top x \iff \| x_i - x \| \leq \left( \frac{2i}{r} \right)^{\frac{1}{2}}$. Also denote by $N_{-1}$ the points satisfying $x_i^\top x < 0$, and let $S_{-1}$ denote this set. Note that

$$g_p(r) = \sum_{i=0}^{n} \| x_i^\top - x \|^p e^{-r \| x_i^\top - x \|^2}$$

$$= \sum_{i=0}^{r-1} \sum_{j \in S_{i+1} \setminus S_i} \| x_i^\top - x \|^p e^{-r \| x_j^\top - x \|^2} + \sum_{j \in S_{-1}} \| x_i^\top - x \|^p e^{-r \| x_j^\top - x \|^2}$$

$$\leq \sum_{i=0}^{r-1} \left( \frac{2(i+1)}{r} \right)^{\frac{p}{2}} e^{-2i} (N_{i+1} - N_i) + 2^p e^{-2r} N_{-1}$$

Similarly,

$$h(r) \geq \sum_{i=0}^{r-1} \left( \frac{2i}{r} \right)^{\frac{p}{2}} e^{-2(i+1)} (N_{i+1} - N_i)$$

Note that because $N_i > 0$,

$$\sum_{i=0}^{r-1} \left( \frac{2(i+1)}{r} \right)^{\frac{p}{2}} N_{i+1} e^{-2i} \geq \sum_{i=0}^{r-1} \left( \frac{2(i+1)}{r} \right)^{\frac{p}{2}} (N_{i+1} - N_i) e^{-2i}$$

And similarly,

$$\sum_{i=0}^{r-1} \left( \frac{2i}{r} \right)^{\frac{p}{2}} N_{i+1} e^{-2i} = \sum_{i=1}^{r-1} \left( \frac{2i}{r} \right)^{\frac{p}{2}} \sum_{j=0}^{i} (N_{j+1} - N_j) e^{-2i} \qquad \because i = 0 \implies \frac{2i}{r} = 0$$

$$= \sum_{j=1}^{r-1} (N_{j+1} - N_j) \sum_{i=j}^{r-1} \left( \frac{2i}{r} \right)^{\frac{p}{2}} e^{-2i}$$

$$\leq \sum_{j=1}^{r-1} (N_{j+1} - N_j) \sum_{i=j}^{\infty} \left( \frac{2i}{r} \right)^{\frac{p}{2}} e^{-2i}$$

$$\leq \sum_{j=1}^{r-1} (N_{j+1} - N_j) \sum_{i=j}^{\infty} \left(\frac{2j}{r}\right)^{\frac{p}{2}} e^{-2j} \left(\frac{\left(\frac{j+1}{j}\right)^{\frac{p}{2}}}{e^2}\right)^{i-j} \qquad \because i < j\left(\frac{j+1}{j}\right)^{i-j}$$

$$\leq \sum_{j=1}^{r-1} (N_{j+1} - N_j) \sum_{i=j}^{\infty} \left(\frac{2j}{r}\right)^{\frac{p}{2}} e^{-2j} \left(e^{\frac{p}{2j}-2}\right)^{i-j} \qquad \because 1 + x \leq e^x$$

$$\leq \sum_{j=1}^{r-1} (N_{j+1} - N_j) \sum_{i=j}^{\infty} \left(\frac{2j}{r}\right)^{\frac{p}{2}} e^{-2j} \left(e^{\frac{p}{2}-2}\right)^{i-j} \qquad \because j \geq 1$$

$$\leq \sum_{j=1}^{r-1} (N_{j+1} - N_j) \left(\frac{2j}{r}\right)^{\frac{p}{2}} e^{-2j} \frac{1}{1 - e^{\frac{p}{2}-2}} \qquad \because p < 4$$

and so

$$\left(1 - e^{\frac{p}{2}-2}\right) \sum_{i=0}^{r-1} \left(\frac{2i}{r}\right)^{\frac{p}{2}} N_{i+1} e^{-2(i+1)} \leq \sum_{j=0}^{r-1} (N_{j+1} - N_j) \left(\frac{2i}{r}\right)^{\frac{p}{2}} e^{-2(i+1)}$$

By a Chernoff bound for Binomial random variables, we have with probability $1 - \frac{r}{n^2}$:

$$N_i = n\sigma_{\frac{i}{r}} \leq n\sigma_{\frac{i}{r}} + \sqrt{6n \log n\sigma_{\frac{i}{r}}} \leq 2n\sigma_{\frac{i}{r}} \ \forall r$$

and

$$N_i = n\sigma_{\frac{i}{r}} \geq n\sigma_{\frac{i}{r}} - \sqrt{4n \log n\sigma_{\frac{i}{r}}} \leq \frac{1}{2}n\sigma_{\frac{i}{r}}$$

Whenever

$$n\sigma_{\frac{i}{r}} \geq 16 \log n \ \forall i \leftarrow \frac{1}{\sqrt{2\pi d}} \left(\frac{1}{r}\right)^{\frac{d}{2}} \geq \frac{16 \log n}{n}$$

and

$$N_{-1} \leq n$$

Over all we have with probability $1 - \frac{r}{n^2}$

$$h(r) \leq \sum_{i=0}^{r-1} \left(\frac{2(i+1)}{r}\right)^{\frac{p}{2}} N_{i+1} e^{-2i} + 2^p N_{-1} e^{-2r}$$

$$\leq n \sum_{i=0}^{r-1} 2e^{-2i} \left(\frac{2(i+1)}{r}\right)^{\frac{d}{2}+\frac{p}{2}} + 2^p e^{-2r} n$$

$$= 2ne^2 \left(\frac{2}{r}\right)^{\frac{d}{2}+\frac{p}{2}} \sum_{i=1}^{r} i^{\frac{d}{2}+\frac{p}{2}} e^{-2i} + 2^p e^{-2r} n$$

$$= 2ne^2 \left(\frac{2}{r}\right)^{\frac{d}{2}+\frac{p}{2}} \gamma(\frac{d}{2} + \frac{p}{2}, 2, r) + 2^p e^{-2r} n \qquad \text{Definition G.2}$$

We always have for $p \leq 2$

$$2ne^2 \left(\frac{2}{r}\right)^{\frac{d}{2}+\frac{p}{2}} \gamma(\frac{d}{2} + \frac{p}{2}, 2, r) \geq 2^p e^{-2r} n$$

$$\leftarrow \left(\frac{d}{2}\right)^{\frac{d}{2}} e^{2r} 2^{-p} \geq r^{\frac{d+p}{2}}$$

So at last, we have

$$g_p(r) \leq 16n \left(\frac{2}{r}\right)^{\frac{d}{2}+\frac{p}{2}} \gamma(\frac{d}{2} + \frac{p}{2}, 2, r)$$

We obtain a lower bound in the same way.

$$h(r) \geq (1 - e^{\frac{p}{2}-2}) \sum_{i=0}^{r-1} \left(\frac{2i}{r}\right)^{\frac{p}{2}} e^{-2(i+1)} \frac{n}{2\sqrt{2\pi d}} \left(\frac{i}{r}\right)^{\frac{d}{2}}$$

$$\geq (1 - e^{\frac{p}{2}-2}) \frac{n 2^{\frac{p}{2}}}{\sqrt{8 e^4 \pi d}} \left(\frac{1}{r}\right)^{\frac{d}{2}+\frac{p}{2}} \sum_{i=0}^{r-1} e^{-2i} i^{\frac{d}{2}+\frac{p}{2}}$$

$$\geq (1 - e^{\frac{p}{2}-2}) \frac{n 2^{\frac{p}{2}}}{\sqrt{8 e^4 \pi d}} \left(\frac{1}{r}\right)^{\frac{d}{2}+\frac{p}{2}} \gamma(\frac{d}{2} + \frac{p}{2}, 2, r)$$

$$(1 - e^{\frac{p}{2}-2}) \frac{n 2^{\frac{p}{2}}}{\sqrt{8 e^4 \pi d}} \left(\frac{1}{r}\right)^{\frac{d}{2}+\frac{p}{2}} \gamma(\frac{d}{2} + \frac{p}{2}, 2, r) \leq h(r) \leq 3n \left(\frac{2}{r}\right)^{\frac{d}{2}+\frac{p}{2}} \gamma(\frac{d}{2} + \frac{p}{2}, 2, r)$$

with probability $1 - \frac{r}{n^2} \geq 1 - \frac{1}{2n}$ when $\frac{n}{\log n} \geq 45\sqrt{d} r^{\frac{d}{2}}$  □

It will be useful to simplify this bound in regimes that we are interested in

**Corollary G.5.** *Suppose $\{x_i\}$ are drawn independently and uniformly from the unit hypersphere. For $\frac{n}{\log n} \geq 45\sqrt{d} r^{\frac{d}{2}}, n > 5, p \leq 2 \leq d$, we have $g_p(r) = \sum_{i=1}^{n} \| x_i - x \|^p e^{-r\| x_i^\top - x \|^2}$ satisfies with probability $1 - \frac{1}{2n}$*

$$\begin{cases} g_p(r) = \Theta\left(\frac{n}{r^{\frac{d+p}{2}}}\right) & r \geq \frac{d+\sqrt{d}}{2} \\ g_p(r) = \Theta\left(n e^{-2r}\right) & r < \frac{d+\sqrt{d}}{2} \end{cases}$$

The following bounds are known for the Gamma function.

**Lemma G.6.** *The Gamma function satisfies*

1. $\sqrt{2\pi d} \left(\frac{d}{e}\right)^d \leq \Gamma(d+1) \leq e\sqrt{2\pi d} \left(\frac{d}{e}\right)^d$

2. $\frac{\Gamma(x+\frac{1}{2})}{\Gamma(x+1)} \geq \frac{1}{\sqrt{x+0.5}}$

*Proof.*    1. Please see [64].

2. Please see [65].

□

**Lemma G.7.** *The following inequality holds:*

$$\left(1 + \frac{1}{\sqrt{d}}\right)^d e^{-\sqrt{d}} \geq e^{-\frac{1}{2}} \tag{20}$$

*Proof.* Take the logarithm of both sides, we have that this is equivalent to

$$d \log \left(1 + \frac{1}{\sqrt{d}}\right) \geq \sqrt{d} - \frac{1}{2}$$

A Taylor series expansion of $\log(1+x)$ demonstrates that $\log\left(1 + \frac{1}{\sqrt{d}}\right) = \sum_i (-1)^{i+1} \frac{1}{i\sqrt{d}^i}$. For $d > 1$, these terms are decreasing in absolute value beyond $i = 2$, so we can upper bound the log with just the first two terms: $\log\left(1 + \frac{1}{\sqrt{d}}\right) \geq \frac{1}{\sqrt{d}} - \frac{1}{2d}$.  □

# H   Attention Window Captures Appropriate Directions

In this section we prove Theorem 4.4, which entails showing that if the Lipschitzness of the function class is zero in some directions, one-layer self-attention learns to ignore these directions when the function class consists of linear functions. First, we give a brief sketch of the proof.

## H.1 Proof Sketch

We briefly sketch the proof of Theorem 4.4. WLOG we write $\mathbf{M} = \mathbf{B}\mathbf{F} + \mathbf{B}_\perp \mathbf{G}$ where $\mathbf{F} := \mathbf{B}^\top \mathbf{M}$ and $\mathbf{G} := \mathbf{B}_\perp^\top \mathbf{M}$. Lemma H.2 leverages the rotational symmetry of $\mathcal{F}_\mathbf{B}$ in $\mathrm{col}(\mathbf{B})$ to show that the loss is minimized over $(\mathbf{F}, \mathbf{G})$ at $(\mathbf{F}, \mathbf{G}) = (c\mathbf{B}^\top, c'\mathbf{B}_\perp)$ for some constants $c, c'$. It remains to show that $\mathcal{L}(c\mathbf{B}\mathbf{B}^\top + c'\mathbf{B}_\perp\mathbf{B}_\perp^\top) > \mathcal{L}(c\mathbf{B}\mathbf{B}^\top)$ whenever $c'$ is nonzero. Intuitively, if the attention estimator incorporates the closeness of $\mathbf{B}_\perp^\top \boldsymbol{x}_i$ and $\mathbf{B}_\perp^\top \boldsymbol{x}_{n+1}$ into its weighting scheme via nonzero $\mathbf{Q}$, this may improperly up- or down-weight $f(\boldsymbol{x}_i)$, since projections of $\boldsymbol{x}_i$ onto $\mathrm{col}(\mathbf{B}_\perp)$ do not carry any information about the closeness of $f(\boldsymbol{x}_i)$ and $f(\boldsymbol{x}_{n+1})$.

Using this intuition, we show that for any fixed $c'$ and $\{\mathbf{v}_i\}_i$ such that $c'\mathbf{v}_i^\top \mathbf{v}_{n+1} \neq \mathbf{v}_{i'}^\top \mathbf{Q}\mathbf{v}_{n+1}$ for some $i, i'$, the attention estimator improperly up-weights $f(\boldsymbol{x}_1)$, where $1 \in \arg\max_i c'\mathbf{v}_i^\top \mathbf{v}_{n+1}$ WLOG. In particular, the version of the pretraining population loss (ICL) with expectation over $\mathbf{a}$, $\{\mathbf{u}_i\}_i$ and $\{\epsilon_i\}_i$ is reduced by reducing $c'\mathbf{v}_1^\top \mathbf{v}_{n+1}$. The only way to ensure all $\{c'\mathbf{v}_i^\top \mathbf{v}_{n+1}\}_i$ are equal for all instances of $\{\mathbf{v}_i\}_i$ is to set $c' = 0$, so this $c'$ must be optimal.

To show that reducing $c'\mathbf{v}_1^\top \mathbf{v}_{n+1}$ reduces the loss with fixed $\{\mathbf{v}_i\}_i$, we define $\alpha_i := e^{c_\mathbf{v} c'\mathbf{v}_i^\top \mathbf{v}_{n+1}}$ for all $i \in [n]$ and show the loss' partial derivative with respect to $\alpha_1$ is positive, i.e.

$$\frac{\partial}{\partial \alpha_1}\left(\tilde{\mathcal{L}}(c, \{\alpha_i\}_i) := \mathbb{E}_{\mathbf{a}, \{\mathbf{u}_i\}_i, \{\epsilon_i\}_i}\left[\left(\frac{\sum_{i=1}^n (\mathbf{a}^\top \mathbf{u}_i - \mathbf{a}^\top \mathbf{u}_{n+1} + \epsilon_i)e^{cc_\mathbf{u}^2 \mathbf{u}_i^\top \mathbf{u}}\alpha_i}{\sum_{i=1}^n e^{cc_\mathbf{u}^2 \mathbf{u}_i^\top \mathbf{u}_{n+1}}\alpha_i}\right)^2\right]\right) > 0. \quad (21)$$

This requires a careful symmetry-based argument as the expectation over $\{\mathbf{u}_i\}_i$ cannot be evaluated in closed-form. To overcome this, we fix all $\mathbf{u}_i$ but $\mathbf{u}_1$ and one other $\mathbf{u}_{i'} \neq \mathbf{u}_{n+1}$ with $\alpha_{i'} < \alpha_1$. We show the expectation over $(\mathbf{u}_1, \mathbf{u}_{i'})$ can be written as an integral over $(\mathbf{y}_1, \mathbf{y}_2) \in \mathbb{S}^{k-1} \times \mathbb{S}^{k-1}$ of a sum of the derivatives at each of the four assignments of $(\mathbf{u}_1, \mathbf{u}_{i'})$ to $(\mathbf{y}_1, \mathbf{y}_2)$, and show that this sum is always positive. Intuitively, any "bad" assignment for which increasing $\alpha_1$ reduces the loss is outweighed by the other assignments, which favor smaller $\alpha_1$. For example, if $\mathbf{y}_1 = \mathbf{u}_{n+1} \neq \mathbf{y}_2$, and $\mathbf{u}_1 = \mathbf{y}_1$ and $\mathbf{u}_{i'} = \mathbf{y}_2$, we observe from (21) that increasing $\alpha_1$ can reduce the loss. However, the cumulative increase in the loss on the other three assignments due to increasing $\alpha_1$ is always greater.

## H.2 Full Proof

We now prove Theorem 4.4 in full detail.

**Lemma H.1.** *For any $\mathbf{u} \in \mathbb{S}^{k-1}$ and $\alpha_1, \ldots, \alpha_n$ such that $\min_i \alpha_i > 0$, and any $c_a, c_u \in \mathbb{R} \setminus \{0\}$, define*

$$J(c) := c_a^2 c_u^2 \mathbb{E}_{\{\mathbf{u}_i\}_{i \in [n]}}\left[\frac{\sum_{i=1}^n \sum_{j=1}^n (\mathbf{u}_i - \mathbf{u})^\top (\mathbf{u}_j - \mathbf{u})e^{c_u^2 c\mathbf{u}_i^\top \mathbf{u} + c_u^2 c\mathbf{u}_j^\top \mathbf{u}}\alpha_i \alpha_j}{(\sum_{i=1}^n e^{c_u^2 c\mathbf{u}_i^\top \mathbf{u}}\alpha_i)^2}\right]$$
$$+ \sigma^2 \mathbb{E}_{\{\mathbf{u}_i\}_{i \in [n]}}\left[\frac{\sum_{i=1}^n e^{2c_u^2 c\mathbf{u}_i^\top \mathbf{u}}\alpha_i^2}{(\sum_{i=1}^n e^{c_u^2 c\mathbf{u}_i^\top \mathbf{u}}\alpha_i)^2}\right]$$

*Then for any $\delta > 0$, $0 \notin \arg\min_{0 \leq c \leq \delta} J(c)$.*

*Proof.* We show that there exists some arbitrarily small $\epsilon > 0$ such that $J(\epsilon) < J(0)$ by showing $\frac{dJ(c)}{dc}\big|_{c=0} < 0$. We have

$$\frac{dJ(c)}{dc}$$
$$= 2c_u^4 \mathbb{E}_{\{\mathbf{u}_i\}_{i \in [n]}}\left[\sum_{i=1}^n \sum_{i'=1}^n \sum_{i''=1}^n (\mathbf{u}_i - \mathbf{u})^\top (\mathbf{u}_{i'} - \mathbf{u})(\mathbf{u}_i^\top \mathbf{u} - \mathbf{u}_{i''}^\top \mathbf{u})\frac{e^{c_u^2 c(\mathbf{u}_i + \mathbf{u}_{i'} + \mathbf{u}_{i''})^\top \mathbf{u}}\alpha_i \alpha_{i'} \alpha_{i''}}{(\sum_{i=1}^n e^{c_u^2 c\mathbf{u}_i^\top \mathbf{u}}\alpha_i)^3}\right]$$
$$+ 2\sigma^2 c_u^2 \mathbb{E}_{\{\mathbf{u}_i\}_{i \in [n]}}\left[\sum_{i=1}^n \sum_{i'=1}^n \sum_{i''=1}^n (\mathbf{u}_i^\top \mathbf{u} - \mathbf{u}_{i'}^\top \mathbf{u})\frac{e^{c_u^2 c(2\mathbf{u}_i + \mathbf{u}_{i'} + \mathbf{u}_{i''})^\top \mathbf{u}}\alpha_i^2 \alpha_{i'} \alpha_{i''}}{(\sum_{i=1}^n e^{c_u^2 c\mathbf{u}_i^\top \mathbf{u}}\alpha_i)^4}\right]$$

Setting $c = 0$ results in

$$\frac{dJ(c)}{dc}\bigg|_{c=0} = \frac{2c_a^2 c_u^4}{(\sum_{i=1}^n \alpha_i)^3} \sum_{i=1}^n \sum_{i'=1}^n \sum_{i''=1}^n \mathbb{E}_{\{\mathbf{u}_i\}_{i \in [n]}}\left[(\mathbf{u}_i - \mathbf{u})^\top (\mathbf{u}_{i'} - \mathbf{u})(\mathbf{u}_i^\top \mathbf{u} - \mathbf{u}_{i''}^\top \mathbf{u})\alpha_i \alpha_{i'} \alpha_{i''}\right]$$

$$+ \frac{2\sigma^2 c_u^2}{(\sum_{i=1}^n \alpha_i)^4} \sum_{i=1}^n \sum_{i'=1}^n \sum_{i''=1}^n \mathbb{E}_{\{\mathbf{u}_i\}_{i\in[n]}} \left[ (\mathbf{u}_i^\top \mathbf{u} - \mathbf{u}_{i'}^\top \mathbf{u}) \alpha_i^2 \alpha_{i'} \alpha_{i''} \right]$$

$$= \frac{2c_a^2 c_u^4}{(\sum_{i=1}^n \alpha_i)^3} \sum_{i=1}^n \sum_{i'=1}^n \sum_{i''=1}^n \mathbb{E}_{\{\mathbf{u}_i\}_{i\in[n]}} \left[ (\mathbf{u}_i - \mathbf{u})^\top (\mathbf{u}_{i'} - \mathbf{u})(\mathbf{u}_i^\top \mathbf{u} - \mathbf{u}_{i''}^\top \mathbf{u}) \alpha_i \alpha_{i'} \alpha_{i''} \right] \tag{22}$$

$$= \frac{2c_a^2 c_u^4}{(\sum_{i=1}^n \alpha_i)^3} \sum_{i=1}^n \sum_{i'=1}^n \sum_{i''=1}^n \mathbb{E}_{\{\mathbf{u}_i\}_{i\in[n]}} \left[ (\mathbf{u}_i^\top \mathbf{u}_{i'} + 1)(\mathbf{u}_i^\top \mathbf{u} - \mathbf{u}_{i''}^\top \mathbf{u}) \alpha_i \alpha_{i'} \alpha_{i''} \right]$$

$$- \frac{2c_u^4}{(\sum_{i=1}^n \alpha_i)^3} \sum_{i=1}^n \sum_{i'=1}^n \sum_{i''=1}^n \mathbb{E}_{\{\mathbf{u}_i\}_{i\in[n]}} \left[ (\mathbf{u}^\top \mathbf{u}_{i'} + \mathbf{u}_i^\top \mathbf{u})(\mathbf{u}_i^\top \mathbf{u} - \mathbf{u}_{i''}^\top \mathbf{u}) \alpha_i \alpha_{i'} \alpha_{i''} \right]$$

$$= -\frac{2c_a^2 c_u^4}{(\sum_{i=1}^n \alpha_i)^3}$$

$$\times \sum_{i=1}^n \sum_{i'=1}^n \sum_{i''=1}^n \mathbb{E}_{\{\mathbf{u}_i\}_{i\in[n]}} \left[ (\mathbf{u}^\top \mathbf{u}_{i'} + \mathbf{u}_i^\top \mathbf{u})(\mathbf{u}_i^\top \mathbf{u} - \mathbf{u}_{i''}^\top \mathbf{u}) \alpha_i \alpha_{i'} \alpha_{i''} \right] \tag{23}$$

$$= -\frac{2c_a^2 c_u^4}{(\sum_{i=1}^n \alpha_i)^3} \sum_{i'=1}^n \alpha_{i'}$$

$$\times \left( \sum_{i=1}^n \sum_{i''=1}^n \mathbb{E}_{\{\mathbf{u}_i\}_{i\in[n]}} \left[ \mathbf{u}^\top \mathbf{u}_{i'} \mathbf{u}_i^\top \mathbf{u} \alpha_i \alpha_{i''} \right] - \sum_{i=1}^n \sum_{i''=1}^n \mathbb{E}_{\{\mathbf{u}_i\}_{i\in[n]}} \left[ \mathbf{u}^\top \mathbf{u}_{i'} \mathbf{u}_{i''}^\top \mathbf{u} \alpha_i \alpha_{i''} \right] \right)$$

$$- \frac{2c_a^2 c_u^4}{(\sum_{i=1}^n \alpha_i)^3} \sum_{i=1}^n \sum_{i'=1}^n \sum_{i''=1}^n \mathbb{E}_{\{\mathbf{u}_i\}_{i\in[n]}} \left[ \mathbf{u}_i^\top \mathbf{u} (\mathbf{u}_i^\top \mathbf{u} - \mathbf{u}_{i''}^\top \mathbf{u}) \alpha_i \alpha_{i'} \alpha_{i''} \right]$$

$$= -\frac{2c_a^2 c_u^4}{(\sum_{i=1}^n \alpha_i)^3} \sum_{i=1}^n \sum_{i'=1}^n \sum_{i''=1}^n \mathbb{E}_{\{\mathbf{u}_i\}_{i\in[n]}} \left[ \mathbf{u}_i^\top \mathbf{u} (\mathbf{u}_i^\top \mathbf{u} - \mathbf{u}_{i''}^\top \mathbf{u}) \alpha_i \alpha_{i'} \alpha_{i''} \right] \tag{24}$$

$$= -\frac{2c_a^2 c_u^4}{(\sum_{i=1}^n \alpha_i)^3} \sum_{i=1}^n \sum_{i'=1}^n \sum_{i''=1}^n \alpha_i \alpha_{i'} \alpha_{i''} \mathbb{E}_{\{\mathbf{u}_i\}_{i\in[n]}} \left[ \mathbf{u}^\top \mathbf{u}_i \mathbf{u}_i^\top \mathbf{u} \right]$$

$$+ \frac{2c_a^2 c_u^4}{(\sum_{i=1}^n \alpha_i)^3} \sum_{i=1}^n \sum_{i'=1}^n \sum_{i''=1}^n \alpha_i \alpha_{i'} \alpha_{i''} \mathbb{E}_{\{\mathbf{u}_i\}_{i\in[n]}} \left[ \mathbf{u}^\top \mathbf{u}_i \mathbf{u}_{i''}^\top \mathbf{u} \right]$$

$$= -\frac{2c_a^2 c_u^4}{k} + \frac{2c_u^4}{k(\sum_{i=1}^n \alpha_i)^3} \sum_{i=1}^n \sum_{i'=1}^n \alpha_i^2 \alpha_{i'} \tag{25}$$

$$= -\frac{2c_a^2 c_u^4}{k} \left( 1 - \frac{\sum_{i=1}^n \alpha_i^2}{(\sum_{i=1}^n \alpha_i)^2} \right)$$

$$< 0 \tag{26}$$

where (22) follows since $\mathbb{E}[\mathbf{u}_i] = \mathbf{0}_k$, (23) similarly follows since odd moments of uniform random variables on the hypersphere are zero, (24) follows by the i.i.d.-ness of the $\mathbf{u}_i$'s, (25) follows since $\mathbb{E}[\mathbf{u}_i \mathbf{u}_i^\top] = \frac{1}{k}\mathbf{I}_k$ and $\mathbf{u}^\top \mathbf{u} = 1$, and (26) follows since $\min_i \alpha_i > 0$. This completes the proof. $\qquad\square$

**Lemma H.2.** *Consider any* $\mathbf{B} \in \mathbb{O}^{d\times k}$ *and resulting function class* $\mathcal{F}_{\mathbf{B}}^{lin}$. *Consider the training population loss* $\mathcal{L}$ *defined in* (ICL), *and tasks drawn from* $D(\mathcal{F}_{\mathbf{B}}^{lin})$ *such that* $\mathbb{E}_{\mathbf{a}}[\mathbf{a}\mathbf{a}^\top] = c_a^2 \mathbf{I}_k$ *for some* $c_a \neq 0$ *and let* $\mathbf{M} := \mathbf{M}_K^\top \mathbf{M}_Q$ *be optimized over the domain* $\mathcal{M}_{\hat{c}} := \{\mathbf{M} \in \mathbb{R}^{d\times d} : \mathbf{M} = \mathbf{M}^\top, \|\mathbf{B}^\top \mathbf{M}\mathbf{B}\|_2 \leq \frac{\hat{c}}{c_u^2}\}$ *for any* $\hat{c} > 0$. *Then any*

$$\mathbf{M}^* \in \arg\min_{\mathbf{M}\in\mathcal{M}_{\hat{c}}} \mathcal{L}(\mathbf{M}) \tag{27}$$

*satisfies* $\mathbf{M}^* = c_1^* \mathbf{B}\mathbf{B}^T + c_2^* \mathbf{B}_\perp \mathbf{B}_\perp^\top$ *for some* $c_1^* : |c_1^*| \in (0, \frac{\hat{c}}{c_u^2}]$.

*Proof.* Without loss of generality (WLOG), we can decompose $\mathbf{M} = \mathbf{B}\mathbf{F} + \mathbf{B}_\perp \mathbf{G}$ where $\mathbf{F} := \mathbf{B}^\top \mathbf{M}$ and $\mathbf{G} := \mathbf{B}_\perp^\top \mathbf{M}$. Recall that for each $i \in [n+1]$, $\mathbf{x}_i = c_u \mathbf{B}\mathbf{u}_i + c_v \mathbf{B}_\perp \mathbf{v}_i$. Thus, for each $i \in [n]$,

we have

$$e^{\mathbf{x}_i^\top \mathbf{M}\mathbf{x}_{n+1}} = e^{\mathbf{x}_i^\top \mathbf{B}\mathbf{F}\mathbf{x}_{n+1}} e^{\mathbf{x}_i^\top \mathbf{B}_\perp \mathbf{G}\mathbf{x}_{n+1}}$$

$$= e^{c_u \mathbf{u}_i^\top \mathbf{F}\mathbf{x}_{n+1}} e^{c_v \mathbf{v}_i^\top \mathbf{G}\mathbf{x}_{n+1}}$$

$$= e^{c_u \mathbf{u}_i^\top \mathbf{F}\mathbf{x}_{n+1}} \alpha_i \qquad (28)$$

where, for each $i \in [n]$, $\alpha_i := e^{c_v \mathbf{v}_i^\top \mathbf{G}\mathbf{x}_{n+1}}$. For ease of notation, denote $\mathbf{x} = \mathbf{x}_{n+1}$.

We start by expanding the square and using the linearity of the expectation to re-write the population loss as:

$$\mathcal{L}(\mathbf{M})$$

$$= \mathbb{E}_{\mathbf{a},\mathbf{x},\{\mathbf{x}_i\}_{i\in[n]},\{\epsilon_i\}_{i\in[n]}}$$

$$\left[ \frac{\sum_{i=1}^n \sum_{j=1}^n (\mathbf{a}^\top \mathbf{B}^\top \mathbf{x}_i - \mathbf{a}^\top \mathbf{B}^\top \mathbf{x} + \epsilon_i)(\mathbf{a}^\top \mathbf{B}^\top \mathbf{x}_j - \mathbf{a}^\top \mathbf{B}^\top \mathbf{x} + \epsilon_j) e^{\mathbf{x}_i^\top \mathbf{M}\mathbf{x} + \mathbf{x}_j^\top \mathbf{M}\mathbf{x}}}{(\sum_{i=1}^n e^{\mathbf{x}_i^\top \mathbf{M}\mathbf{x}})^2} \right]$$

$$= c_u^2 \mathbb{E}_{\mathbf{a},\mathbf{x},\{\mathbf{u}_i\},\{\mathbf{v}_i\}_{i\in[n]}}$$

$$\left[ \frac{\sum_{i=1}^n \sum_{j=1}^n (\mathbf{a}^\top \mathbf{u}_i - \mathbf{a}^\top \mathbf{u})(\mathbf{a}^\top \mathbf{u}_j - \mathbf{a}^\top \mathbf{u}) e^{c_u \mathbf{u}_i^\top \mathbf{F}\mathbf{x} + c_u \mathbf{u}_j^\top \mathbf{F}\mathbf{x}} \alpha_i \alpha_j}{(\sum_{i=1}^n e^{c_u \mathbf{u}_i^\top \mathbf{F}\mathbf{x}} \alpha_i)^2} \right]$$

$$+ \sigma^2 \mathbb{E}_{u,\{\mathbf{u}_i\},\{\alpha_i\}_{i\in[n]},\{\epsilon_i\}_{i\in[n]}} \left[ \frac{\sum_{i=1}^n e^{2c_u \mathbf{u}_i^\top \mathbf{F}\mathbf{x}} \alpha_i^2}{(\sum_{i=1}^n e^{c_u \mathbf{u}_i^\top \mathbf{F}\mathbf{x}} \alpha_i)^2} \right]$$

$$= \mathbb{E}_{\mathbf{x}} \left[ \underbrace{c_a^2 c_u^2 \mathbb{E}_{\{\mathbf{u}_i\},\{\mathbf{v}_i\}_{i\in[n]}} \left[ \frac{\sum_{i=1}^n \sum_{j=1}^n (\mathbf{u}_i - \mathbf{u})^\top (\mathbf{u}_j - \mathbf{u}) e^{c_u \mathbf{u}_i^\top \mathbf{F}\mathbf{x} + c_u \mathbf{u}_j^\top \mathbf{F}\mathbf{x}} \alpha_i \alpha_j}{(\sum_{i=1}^n e^{c_u \mathbf{u}_i^\top \mathbf{F}\mathbf{x}} \alpha_i)^2} \right]}_{=:\tilde{\mathcal{L}}_{\text{signal}}(\mathbf{M},\mathbf{x})} \right.$$

$$\left. + \sigma^2 \underbrace{\mathbb{E}_{\{\mathbf{u}_i\},\{\mathbf{v}_i\}_{i\in[n]}} \left[ \frac{\sum_{i=1}^n e^{2c_u \mathbf{u}_i^\top \mathbf{F}\mathbf{x}} \alpha_i^2}{(\sum_{i=1}^n e^{c_u^2 \mathbf{u}_i^\top \mathbf{F}\mathbf{x}} \alpha_i)^2} \right]}_{=:\tilde{\mathcal{L}}_{\text{noise}}(\mathbf{M},\mathbf{x})} \right] \qquad (29)$$

WLOG we can write $\mathbf{F}\mathbf{x} = \mathbf{R}(\mathbf{F}\mathbf{x})\mathbf{u}\|\mathbf{F}\mathbf{x}\|_2$ for some rotation matrix $\mathbf{R}(\mathbf{F}\mathbf{x}) \in \mathbb{O}^{k\times k}$. Denote $C_1(\mathbf{F}\mathbf{x}) := \|\mathbf{F}\mathbf{x}\|_2$. Then we have

$$\tilde{\mathcal{L}}_{\text{signal}}(\mathbf{M},\mathbf{x})$$

$$= c_a^2 c_u^2 \mathbb{E}_{\{\mathbf{u}_i\},\{\mathbf{v}_i\}} \left[ \frac{\sum_{i=1}^n \sum_{j=1}^n (\mathbf{u}_i - \mathbf{u})^\top (\mathbf{u}_j - \mathbf{u}) e^{c_u C_1(\mathbf{F}\mathbf{x})\mathbf{u}_i^\top \mathbf{R}(\mathbf{F}\mathbf{x})\mathbf{u} + c_u C_1(\mathbf{F}\mathbf{x})\mathbf{u}_j^\top \mathbf{R}(\mathbf{F}\mathbf{x})\mathbf{u}} \alpha_i \alpha_j}{(\sum_{i=1}^n e^{c_u C_1(\mathbf{F}\mathbf{x})\mathbf{u}_i^\top \mathbf{R}(\mathbf{F}\mathbf{x})\mathbf{u}} \alpha_i)^2} \right]$$

$$= c_a^2 c_u^2 \mathbb{E}_{\{\mathbf{u}_i\},\{\mathbf{v}_i\}} \left[ \frac{\sum_{i=1}^n \sum_{j=1}^n (\mathbf{u}_i - \mathbf{u})^\top \mathbf{R}(\mathbf{F}\mathbf{x})\mathbf{R}(\mathbf{F}\mathbf{x})^\top (\mathbf{u}_j - \mathbf{u}) e^{c_u C_1(\mathbf{F}\mathbf{x})\mathbf{u}_i^\top \mathbf{R}(\mathbf{F}\mathbf{x})\mathbf{u} + c_u C_1(\mathbf{F}\mathbf{x})\mathbf{u}_j^\top \mathbf{R}(\mathbf{F}\mathbf{x})\mathbf{u}} \alpha_i \alpha_j}{(\sum_{i=1}^n e^{c_u C_1(\mathbf{F}\mathbf{x})\mathbf{u}_i^\top \mathbf{R}(\mathbf{F}\mathbf{x})\mathbf{u}} \alpha_i)^2} \right]$$

$$(30)$$

$$= c_a^2 c_u^2 \mathbb{E}_{\{\mathbf{u}_i\},\{\mathbf{v}_i\}} \left[ \frac{1}{(\sum_{i=1}^n e^{c_u C_1(\mathbf{F}\mathbf{x})\mathbf{u}_i^\top \mathbf{R}(\mathbf{F}\mathbf{x})\mathbf{u}} \alpha_i)^2} \right.$$

$$\times \sum_{i=1}^n \sum_{j=1}^n \left( (\mathbf{R}(\mathbf{F}\mathbf{x})^\top \mathbf{u}_i - \mathbf{R}(\mathbf{F}\mathbf{x})^\top \mathbf{u})^\top (\mathbf{R}(\mathbf{F}\mathbf{x})^\top \mathbf{u}_j - \mathbf{R}(\mathbf{F}\mathbf{x})^\top \mathbf{u}) \right.$$

$$\left. \left. \times e^{c_u C_1(\mathbf{F}\mathbf{x})\mathbf{u}_i^\top \mathbf{R}(\mathbf{F}\mathbf{x})\mathbf{u} + c_u C_1(\mathbf{F}\mathbf{x})\mathbf{u}_j^\top \mathbf{R}(\mathbf{F}\mathbf{x})\mathbf{u}} \alpha_i \alpha_j \right) \right]$$

$$= c_a^2 c_u^2 \mathbb{E}_{\{\mathbf{u}_i\},\{\mathbf{v}_i\}} \left[ \frac{\sum_{i=1}^n \sum_{j=1}^n (\mathbf{u}_i - \mathbf{R}(\mathbf{F}\mathbf{x})^\top \mathbf{u})^\top (\mathbf{u}_j - \mathbf{R}(\mathbf{F}\mathbf{x})^\top \mathbf{u}) e^{c_u C_1(\mathbf{F}\mathbf{x})\mathbf{u}_i^\top \mathbf{u} + c_u C_1(\mathbf{F}\mathbf{x})\mathbf{u}_j^\top \mathbf{u}} \alpha_i \alpha_j}{(\sum_{i=1}^n e^{c_u C_1(\mathbf{F}\mathbf{x})\mathbf{u}_i^\top \mathbf{u}} \alpha_i)^2} \right]$$

$$(31)$$

where (30) follows since $\mathbf{R}(\mathbf{Fx})\mathbf{R}(\mathbf{Fx})^\top = \mathbf{I}_k$ and (31) follows since the distribution of $\mathbf{u}_i$ is the same as the distribution of $\mathbf{R}(\mathbf{Fx})^\top \mathbf{u}_i$ for any rotation $\mathbf{R}(\mathbf{Fx})^\top$. Define

$$g(\mathbf{F}, \mathbf{u}, \mathbf{v}) := \mathbb{E}_{\{\mathbf{u}_i\}, \{\mathbf{v}_i\}} \left[ \frac{\sum_{i=1}^n \sum_{j=1}^n (\mathbf{u}_i - \mathbf{R}(\mathbf{Fx})^\top \mathbf{u})^\top (\mathbf{u}_j - \mathbf{R}(\mathbf{Fx})^\top \mathbf{u}) e^{c_u C_1(\mathbf{Fx})\mathbf{u}_i^\top \mathbf{u} + c_u C_1(\mathbf{Fx})\mathbf{u}_j^\top \mathbf{u}} \alpha_i \alpha_j}{(\sum_{i=1}^n e^{c_u C_1(\mathbf{Fx})\mathbf{u}_i^\top \mathbf{u}} \alpha_i)^2} \right]$$

for any $\mathbf{F} \in \mathbb{R}^{k \times d}$. We have $\mathcal{L}_{\mathrm{signal}}(\mathbf{M}) = c_a^2 c_u^2 \mathbb{E}_{\mathbf{u}, \mathbf{v}}[g(\mathbf{F}, \mathbf{u}, \mathbf{v})]$, and Note that if $\mathbf{F}' = c\mathbf{B}^\top$, then $\mathbf{R}_{\mathbf{F}'\mathbf{x}} = \mathbf{I}_k$ and $C_1(\mathbf{F}'\mathbf{x}) = c_u c$. Thus,

$$g(\mathbf{F}, \mathbf{u}, \mathbf{v}) - g(\frac{C_1(\mathbf{Fx})}{c_u}\mathbf{B}^\top, \mathbf{u}, \mathbf{v})$$

$$= \mathbb{E}_{\{\mathbf{u}_i\}, \{\mathbf{v}_i\}} \left[ \frac{\sum_{i=1}^n \sum_{j=1}^n (\mathbf{u}_i - \mathbf{R}(\mathbf{Fx})^\top \mathbf{u})^\top (\mathbf{u}_j - \mathbf{R}(\mathbf{Fx})^\top \mathbf{u}) e^{c_u C_1(\mathbf{Fx})\mathbf{u}_i^\top \mathbf{u} + c_u C_1(\mathbf{Fx})\mathbf{u}_j^\top \mathbf{u}} \alpha_i \alpha_j}{(\sum_{i=1}^n e^{c_u C_1(\mathbf{Fx})\mathbf{u}_i^\top \mathbf{u}} \alpha_i)^2} \right]$$

$$\quad - \mathbb{E}_{\{\mathbf{u}_i\}, \{\mathbf{v}_i\}} \left[ \frac{\sum_{i=1}^n \sum_{j=1}^n (\mathbf{u}_i - \mathbf{u})^\top (\mathbf{u}_j - \mathbf{u}) e^{c_u C_1(\mathbf{Fx})\mathbf{u}_i^\top \mathbf{u} + c_u C_1(\mathbf{Fx})\mathbf{u}_j^\top \mathbf{u}} \alpha_i \alpha_j}{(\sum_{i=1}^n e^{c_u C_1(\mathbf{Fx})\mathbf{u}_i^\top \mathbf{u}} \alpha_i)^2} \right]$$

$$= \mathbb{E}_{\{\mathbf{u}_i\}, \{\mathbf{v}_i\}}$$

$$\left[ \frac{\sum_{i=1}^n \sum_{j=1}^n (\mathbf{u}_i^\top \mathbf{u} - \mathbf{u}_i^\top \mathbf{R}(\mathbf{Fx})^\top \mathbf{u} + \mathbf{u}_j^\top \mathbf{u} - \mathbf{u}_j^\top \mathbf{R}(\mathbf{Fx})^\top \mathbf{u}) e^{c_u C_1(\mathbf{Fx})\mathbf{u}_i^\top \mathbf{u} + c_u C_1(\mathbf{Fx})\mathbf{u}_j^\top \mathbf{u}} \alpha_i \alpha_j}{(\sum_{i=1}^n e^{c_u C_1(\mathbf{Fx})\mathbf{u}_i^\top \mathbf{u}} \alpha_i)^2} \right]$$

$$= 2\mathbb{E}_{\{\mathbf{u}_i\}, \{\mathbf{v}_i\}} \left[ \frac{\sum_{i=1}^n (\mathbf{u}_i^\top \mathbf{u} - \mathbf{u}_i^\top \mathbf{R}(\mathbf{Fx})^\top \mathbf{u}) e^{c_u C_1(\mathbf{Fx})\mathbf{u}_i^\top \mathbf{u}} \alpha_i \sum_{j=1}^n e^{c_u C_1(\mathbf{Fx})\mathbf{u}_j^\top \mathbf{u}} \alpha_j}{(\sum_{i=1}^n e^{c_u C_1(\mathbf{Fx})\mathbf{u}_i^\top \mathbf{u}} \alpha_i)^2} \right]$$

$$= 2\mathbb{E}_{\{\mathbf{u}_i\}, \{\mathbf{v}_i\}} \left[ \frac{\sum_{i=1}^n (\mathbf{u}_i^\top \mathbf{u} - \mathbf{u}_i^\top \mathbf{R}(\mathbf{Fx})^\top \mathbf{u}) e^{c_u C_1(\mathbf{Fx})\mathbf{u}_i^\top \mathbf{u}} \alpha_i}{\sum_{i=1}^n e^{c_u C_1(\mathbf{Fx})\mathbf{u}_i^\top \mathbf{u}} \alpha_i} \right]$$

$$= 2(\mathbf{u}^\top - \mathbf{u}^\top \mathbf{R}(\mathbf{Fx}))\mathbb{E}_{\{\mathbf{u}_i\}, \{\mathbf{v}_i\}} \left[ \frac{\sum_{i=1}^n \mathbf{u}_i e^{c_u C_1(\mathbf{Fx})\mathbf{u}_i^\top \mathbf{u}} \alpha_i}{\sum_{i=1}^n e^{c_u C_1(\mathbf{Fx})\mathbf{u}_i^\top \mathbf{u}} \alpha_i} \right] \tag{32}$$

Define $\hat{\mathbf{u}} := \mathbb{E}_{\{\mathbf{u}_i\}, \{\mathbf{v}_i\}} \left[ \frac{\sum_{i=1}^n \mathbf{u}_i e^{c_u C_1(\mathbf{Fx})\mathbf{u}_i^\top \mathbf{u}} \alpha_i}{\sum_{i=1}^n e^{c_u C_1(\mathbf{Fx})\mathbf{u}_i^\top \mathbf{u}} \alpha_i} \right]$ and WLOG write $\mathbf{u}_i = \mathbf{p}_{\mathbf{u}_i} + \mathbf{q}_{\mathbf{u}_i}$, where $\mathbf{p}_{\mathbf{u}_i} :=$ $\mathbf{u}\mathbf{u}^\top \mathbf{u}_i$ and $\mathbf{q}_{\mathbf{u}_i} := (\mathbf{I}_k - \mathbf{u}\mathbf{u}^\top)\mathbf{u}_i$. Note that for any $\mathbf{u}_i = \mathbf{p}_{\mathbf{u}_i} + \mathbf{q}_{\mathbf{u}_i}$, $\mathbf{u}'_i := \mathbf{p}_{\mathbf{u}_i} - \mathbf{q}_{\mathbf{u}_i}$ occurs with equal probability, and flipping $\mathbf{q}_{\mathbf{u}_i}$ does not change any exponent or $\alpha_i$ in (32). Thus

$$\hat{\mathbf{u}} = \mathbb{E}_{\{(\mathbf{p}_{\mathbf{u}_i}, \mathbf{q}_{\mathbf{u}_i})\}_{i\in[n]}, \{\mathbf{v}_i\}} \left[ \frac{\sum_{i=1}^n (\mathbf{p}_{\mathbf{u}_i} + \mathbf{q}_{\mathbf{u}_i}) e^{c_u C_1(\mathbf{Fx})\mathbf{u}_i^\top \mathbf{u}} \alpha_i}{\sum_{i=1}^n e^{c_u C_1(\mathbf{Fx})\mathbf{u}_i^\top \mathbf{u}} \alpha_i} \right]$$

$$= \frac{1}{2}\mathbb{E}_{\{(\mathbf{p}_{\mathbf{u}_i}, \mathbf{q}_{\mathbf{u}_i})\}_i, \{\mathbf{v}_i\}} \left[ \frac{\sum_{i=1}^n (2\mathbf{p}_{\mathbf{u}_i} + \mathbf{q}_{\mathbf{u}_i} - \mathbf{q}_{\mathbf{u}_i}) e^{c_u C_1(\mathbf{Fx})\mathbf{u}_i^\top \mathbf{u}} \alpha_i}{\sum_{i=1}^n e^{c_u C_1(\mathbf{Fx})\mathbf{u}_i^\top \mathbf{u}} \alpha_i} \right]$$

$$= \mathbb{E}_{\{\mathbf{p}_{\mathbf{u}_i}\}_i, \{\mathbf{v}_i\}} \left[ \frac{\sum_{i=1}^n \mathbf{p}_{\mathbf{u}_i} e^{c_u C_1(\mathbf{Fx})\mathbf{u}_i^\top \mathbf{u}} \alpha_i}{\sum_{i=1}^n e^{c_u C_1(\mathbf{Fx})\mathbf{u}_i^\top \mathbf{u}} \alpha_i} \right] \tag{33}$$

$$= \tilde{c}\,\mathbf{u}$$

where $\tilde{c} := \mathbb{E}_{\{\mathbf{u}_i\}, \{\mathbf{v}_i\}} \left[ \frac{\sum_{i=1}^n \mathbf{u}_i^\top \mathbf{u}\, e^{c_u C_1(\mathbf{Fx})\mathbf{u}_i^\top \mathbf{u}} \alpha_i}{\sum_{i=1}^n e^{c_u C_1(\mathbf{Fx})\mathbf{u}_i^\top \mathbf{u}} \alpha_i} \right]$. Note that for any $\mathbf{u}_i$, $-\mathbf{u}_i$ occurs with equal probability, so

$$\tilde{c} = \sum_{i=1}^n \mathbb{E}_{\{\mathbf{u}_i\}, \{\mathbf{v}_i\}} \left[ \frac{\mathbf{u}^\top \mathbf{u}_i\, e^{c_u C_1(\mathbf{Fx})\mathbf{u}_i^\top \mathbf{u}} \alpha_i}{\sum_{j=1}^n e^{c_u C_1(\mathbf{Fx})\mathbf{u}_j^\top \mathbf{u}} \alpha_j} \right]$$

$$= \frac{1}{2}\sum_{i=1}^n \mathbb{E}_{\{\mathbf{u}_i\}, \{\mathbf{v}_i\}} \left[ \frac{\mathbf{u}_i^\top \mathbf{u}\, e^{c_u C_1(\mathbf{Fx})\mathbf{u}_i^\top \mathbf{u}} \alpha_i}{e^{c_u C_1(\mathbf{Fx})\mathbf{u}_i^\top \mathbf{u}} \alpha_i + \sum_{j=1, j\neq i}^n e^{c_u C_1(\mathbf{Fx})\mathbf{u}_j^\top \mathbf{u}} \alpha_j} \right.$$

$$-\frac{\mathbf{u}_i^\top \mathbf{u}\, e^{-c_u C_1(\mathbf{Fx})\mathbf{u}_i^\top \mathbf{u}}\alpha_i}{e^{-c_u C_1(\mathbf{Fx})\mathbf{u}_i^\top \mathbf{u}}\alpha_i + \sum_{j=1,j\neq i}^n e^{c_u C_1(\mathbf{Fx})\mathbf{u}_j^\top \mathbf{u}}\alpha_j}\Bigg]$$

$$= \frac{1}{2}\sum_{i=1}^n \mathbb{E}_{\{\mathbf{u}_i\},\{\mathbf{v}_i\}}\Bigg[\mathbf{u}_i^\top \mathbf{u}\Bigg(\frac{e^{c_u C_1(\mathbf{Fx})\mathbf{u}_i^\top \mathbf{u}}\alpha_i}{e^{c_u C_1(\mathbf{Fx})\mathbf{u}_i^\top \mathbf{u}}\alpha_i + \sum_{j=1,j\neq i}^n e^{c_u C_1(\mathbf{Fx})\mathbf{u}_j^\top \mathbf{u}}\alpha_j}$$
$$-\frac{e^{-c_u C_1(\mathbf{Fx})\mathbf{u}_i^\top \mathbf{u}}\alpha_i}{e^{-c_u C_1(\mathbf{Fx})\mathbf{u}_i^\top \mathbf{u}}\alpha_i + \sum_{j=1,j\neq i}^n e^{c_u C_1(\mathbf{Fx})\mathbf{u}_j^\top \mathbf{u}}\alpha_j}\Bigg)\Bigg]. \quad (34)$$

Since $\alpha_i > 0$ and $\bar{c}_{u,v} > 0$ by definition, $e^{c_u C_1(\mathbf{Fx})\mathbf{u}_i^\top \mathbf{u}}\alpha_i$ is monotonically increasing in $\mathbf{u}_i^\top \mathbf{u}$. Also, $f(x) := \frac{x}{x+c}$ is monotonically increasing for $x > 0$ for all $c > 0$. Thus we have that

$$\mathbf{u}_i^\top \mathbf{u} > 0$$
$$\iff \left(\frac{e^{c_u C_1(\mathbf{Fx})\mathbf{u}_i^\top \mathbf{u}}\alpha_i}{e^{c_u C_1(\mathbf{Fx})\mathbf{u}_i^\top \mathbf{u}}\alpha_i + \sum_{j=1,j\neq i}^n e^{c_u C_1(\mathbf{Fx})\mathbf{u}_j^\top \mathbf{u}}\alpha_j} - \frac{e^{-c_u C_1(\mathbf{Fx})\mathbf{u}_i^\top \mathbf{u}}\alpha_i}{e^{-c_u C_1(\mathbf{Fx})\mathbf{u}_i^\top \mathbf{u}}\alpha_i + \sum_{j=1,j\neq i}^n e^{c_u C_1(\mathbf{Fx})\mathbf{u}_j^\top \mathbf{u}}\alpha_j}\right) > 0,$$
$$(35)$$

and thereby $\tilde{c} > 0$. Therefore, $\arg\max_{\mathbf{u}'\in\mathbb{S}^{k-1}}(\mathbf{u}')^\top \hat{\mathbf{u}} = \mathbf{u}$, in particular $\mathbf{u}^\top \hat{\mathbf{u}} > \mathbf{u}^\top \mathbf{R}(\mathbf{Fx})^\top \hat{\mathbf{u}}$ whenever $\mathbf{R}(\mathbf{Fx})\mathbf{u} \neq \mathbf{u}$, so (32) is strictly positive if $\mathbf{R}(\mathbf{Fx})\mathbf{u} \neq \mathbf{u}$. Thus, for any $\mathbf{u}, \mathbf{v}$ such that $\mathbf{R}(\mathbf{Fx})\mathbf{u} \neq \mathbf{u}$, $g(\mathbf{F}, \mathbf{u}, \mathbf{v}) > g(\frac{C_1(\mathbf{Fx})}{c_u}\mathbf{B}^\top, \mathbf{u}, \mathbf{v})$. Also, for any $\mathbf{u}, \mathbf{v}$ such that $\mathbf{R}(\mathbf{Fx})\mathbf{u} = \mathbf{u}$, $g(\mathbf{F}, \mathbf{u}, \mathbf{v}) = g(\frac{C_1(\mathbf{Fx})}{c_u}\mathbf{B}^\top, \mathbf{u}, \mathbf{v})$.

Next we need to account for $\tilde{\mathcal{L}}_{\text{noise}}(\mathbf{M}, \mathbf{x})$. Again writing $\mathbf{Fx} = \mathbf{R}(\mathbf{Fx})\mathbf{u}\|\mathbf{Fx}\|_2$ and $C_1(\mathbf{Fx}) = \|\mathbf{Fx}\|_2$ and using the rotational invariance of $\mathbf{u}_i$, we obtain

$$\mathcal{L}_{\text{noise}}(\mathbf{M}) = \sigma^2 \mathbb{E}_{\mathbf{x},\{\mathbf{x}_i\}_{i\in[n]}}\left[\frac{\sum_{i=1}^n e^{2c_u \mathbf{u}_i^\top \mathbf{Fx}}\alpha_i^2}{(\sum_{i=1}^n e^{c_u \mathbf{u}_i^\top \mathbf{Fx}}\alpha_i)^2}\right]$$
$$= \sigma^2 \mathbb{E}_{\mathbf{u},\mathbf{v},\{\mathbf{u}_i\},\{\mathbf{v}_i\}}\left[\frac{\sum_{i=1}^n e^{2c_u C_1(\mathbf{Fx})\mathbf{u}_i^\top \mathbf{R}(\mathbf{Fx})\mathbf{u}}\alpha_i^2}{(\sum_{i=1}^n e^{c_u C_1(\mathbf{Fx})\mathbf{u}_i^\top \mathbf{R}(\mathbf{Fx})\mathbf{u}}\alpha_i)^2}\right]$$
$$= \sigma^2 \mathbb{E}_{\mathbf{u},\mathbf{v},\{\mathbf{u}_i\},\{\mathbf{v}_i\}}\left[\frac{\sum_{i=1}^n e^{2c_u C_1(\mathbf{Fx})\mathbf{u}_i^\top \mathbf{u}}\alpha_i^2}{(\sum_{i=1}^n e^{c_u C_1(\mathbf{Fx})\mathbf{u}_i^\top \mathbf{u}}\alpha_i)^2}\right] \quad (36)$$

where (36) follows using the rotational invariance of $\mathbf{u}_i$. So, returning to (29), we have

$$\mathcal{L}(\mathbf{M}) = \mathbb{E}_{\mathbf{x}}[\tilde{\mathcal{L}}_{\text{signal}}(\mathbf{BF} + \mathbf{B}_\perp \mathbf{G}, \mathbf{x}) + \tilde{\mathcal{L}}_{\text{noise}}(\mathbf{BF} + \mathbf{B}_\perp \mathbf{G}, \mathbf{x})]$$
$$\geq \mathbb{E}_{\mathbf{u},\mathbf{v}}\left[c_a^2 c_u^2 \mathbb{E}_{\{\mathbf{u}_i\},\{\mathbf{v}_i\}}\left[\frac{\sum_{i=1}^n \sum_{j=1}^n (\mathbf{u}_i - \mathbf{u})^\top (\mathbf{u}_j - \mathbf{u})e^{c_u C_1(\mathbf{Fx})\mathbf{u}_i^\top \mathbf{u} + c_u C_1(\mathbf{Fx})\mathbf{u}_j^\top \mathbf{u}}\alpha_i \alpha_j}{(\sum_{i=1}^n e^{c_u C_1(\mathbf{Fx})\mathbf{u}_i^\top \mathbf{u}}\alpha_i)^2}\right]\right.$$
$$\left.+ \sigma^2 \mathbb{E}_{\{\mathbf{u}_i\},\{\mathbf{v}_i\}}\left[\frac{\sum_{i=1}^n e^{2c_u C_1(\mathbf{Fx})\mathbf{u}_i^\top \mathbf{u}}\alpha_i^2}{(\sum_{i=1}^n e^{c_u C_1(\mathbf{Fx})\mathbf{u}_i^\top \mathbf{u}}\alpha_i)^2}\right]\right] \quad (37)$$

where (37) is strict if $\mathbf{R}(\mathbf{Fx})\mathbf{u} \neq \mathbf{u}$ for some $\mathbf{u}, \mathbf{v}$, which is equivalent to saying that $\mathbf{F} \notin \{c'\mathbf{B}^\top, c' > 0\}$.

Next, recall that we have defined $\alpha_i := e^{c_v \mathbf{v}_i^\top \mathbf{Gx}}$. Using a similar argument as earlier, by the rotational invariance of the $\mathbf{v}_i$'s, for any fixed $\mathbf{x}$, we can write $\alpha_i = e^{c_v C_2(\mathbf{Gx})\mathbf{v}_i^\top \mathbf{e}_1}$ where $C_2(\mathbf{Gx}) := \|\mathbf{Gx}\|_2$ and $\mathbf{e}_1$ is the first standard basis vector.

Next, for $c_1, c_2 \geq 0$ and some fixed $\mathbf{u}, \mathbf{v}$, define

$$H(\mathbf{u}, \mathbf{v}, c_1, c_2) := c_a^2 c_u^2 \mathbb{E}_{\{\mathbf{u}_i\},\{\mathbf{v}_i\}}\left[\frac{\sum_{i=1}^n \sum_{j=1}^n (\mathbf{u}_i - \mathbf{u})^\top (\mathbf{u}_j - \mathbf{u})e^{c_u c_1 \mathbf{u}_i^\top \mathbf{u} + c_u c_1 \mathbf{u}_j^\top \mathbf{u}}e^{c_v c_2 \mathbf{v}_i^\top \mathbf{e}_1 + c_v c_2 \mathbf{v}_j^\top \mathbf{e}_1}}{(\sum_{i=1}^n e^{c_u c_1 \mathbf{u}_i^\top \mathbf{u}}e^{c_v c_2 \mathbf{v}_i^\top \mathbf{e}_1})^2}\right]$$

$$+ \sigma^2 \mathbb{E}_{\{\mathbf{u}_i\},\{\mathbf{v}_i\}} \left[ \frac{\sum_{i=1}^n e^{2c_u c_1 \mathbf{u}_i^\top \mathbf{u}} e^{2c_v c_2 \mathbf{v}_i^\top \mathbf{e}_1}}{(\sum_{i=1}^n e^{c_u c_1 \mathbf{u}_i^\top \mathbf{u}} e^{c_v c_2 \mathbf{v}_i^\top \mathbf{e}_1})^2} \right] \tag{38}$$

and let

$$(C_1^*(\mathbf{u}, \mathbf{v}), C_2^*(\mathbf{u}, \mathbf{v})) \in \arg \min_{(c_1,c_2): 0 \le c_1 \le \frac{\hat{c}}{c_u^2}, c_2 \ge 0} H(\mathbf{u}, \mathbf{v}, c_1, c_2) \tag{39}$$

Since $H$ does not vary with $\mathbf{v}$, we have $(C_1^*(\mathbf{u}, \mathbf{v}), C_2^*(\mathbf{u}, \mathbf{v})) = (C_1^*(\mathbf{u}), C_2^*(\mathbf{u}))$ WLOG. In fact, $H$ does not vary with $\mathbf{u}$ either, due to the rotational invariance of the $\mathbf{u}_i$'s. So, we have $(C_1^*(\mathbf{u}, \mathbf{v}), C_2^*(\mathbf{u}, \mathbf{v})) = (C_1^*, C_2^*)$ WLOG, i.e. there is a single pair $(C_1^*, C_2^*)$ that minimizes $H(\mathbf{u}, \mathbf{v}, c_1, c_2)$ over $c_1, c_2$ for all $\mathbf{u} \in \mathbb{S}^{k-1}$ and $\mathbf{v} \in \mathbb{S}^{d-k-1}$.

Thus $\mathbf{F}^* = C_1^* \mathbf{B}^\top$ and $\mathbf{G}^*$ satisfies $\|\mathbf{G}^* \mathbf{x}\| = C_2^*$ for all $\mathbf{x}$, which implies, using also that $\mathbf{M}$ is symmetric, that $\mathbf{G}^* = C_2^* \mathbf{B}_\perp^\top$. $\qquad \square$

**Lemma H.3.** *Consider any* $\boldsymbol{\alpha} := [\alpha_1, \ldots, \alpha_n]$ *such that* $\alpha_1 = \max_i \alpha_i$ *and* $\alpha_1 > \min_i \alpha_i > 0$. *Further, let* $c \in (0, 2]$. *Define*

$$H_{signal}(\mathbf{u}, \boldsymbol{\alpha}) := \mathbb{E}_{\{\mathbf{u}_i\}_{i \in [n]}} \left[ \frac{\sum_{i=1}^n \sum_{j=1}^n (\mathbf{u}_i - \mathbf{u})^\top (\mathbf{u}_j - \mathbf{u}) e^{c\mathbf{u}_i^\top \mathbf{u} + c\mathbf{u}_j^\top \mathbf{u}} \alpha_i \alpha_j}{(\sum_{i=1}^n e^{c\mathbf{u}_i^\top \mathbf{u}} \alpha_i)^2} \right]. \tag{40}$$

*Then*

$$\frac{\partial H_{signal}(\mathbf{u}, \boldsymbol{\alpha})}{\partial \alpha_1} > 0.$$

*Proof.* We first compute $\frac{\partial H_{signal}(\mathbf{u}, \boldsymbol{\alpha})}{\partial \alpha_1}$. Using the linearity of the expectation and the quotient rule we obtain:

$$\frac{\partial H_{\text{signal}}(\mathbf{u}, \boldsymbol{\alpha})}{\partial \alpha_1}$$

$$= \mathbb{E}_{\{\mathbf{u}_i\}_{i \in [n]}} \left[ \frac{\partial}{\partial \alpha_1} \frac{\sum_{i=1}^n \sum_{j=1}^n (\mathbf{u}_i - \mathbf{u})^\top (\mathbf{u}_j - \mathbf{u}) e^{c\mathbf{u}_i^\top \mathbf{u} + c\mathbf{u}_j^\top \mathbf{u}} \alpha_i \alpha_j}{(\sum_{i=1}^n e^{c\mathbf{u}_i^\top \mathbf{u}} \alpha_i)^2} \right]$$

$$= 2\mathbb{E}_{\{\mathbf{u}_i\}_i} \left[ \frac{(\sum_{i=1}^n e^{c\mathbf{u}_i^\top \mathbf{u}} \alpha_i)^2 \left( \sum_{j=2}^n (\mathbf{u}_1 - \mathbf{u})^\top (\mathbf{u}_j - \mathbf{u}) e^{c\mathbf{u}_1^\top \mathbf{u} + c\mathbf{u}_j^\top \mathbf{u}} \alpha_j + \|\mathbf{u}_1 - \mathbf{u}\|_2^2 e^{2c\mathbf{u}_1^\top \mathbf{u}} \alpha_1 \right)}{(\sum_{i=1}^n e^{c\mathbf{u}_i^\top \mathbf{u}} \alpha_i)^4} \right]$$

$$\quad - 2\mathbb{E}_{\{\mathbf{u}_i\}_i} \left[ \frac{(\sum_{i=1}^n \sum_{j=1}^n (\mathbf{u}_i - \mathbf{u})^\top (\mathbf{u}_j - \mathbf{u}) e^{c\mathbf{u}_i^\top \mathbf{u} + c\mathbf{u}_j^\top \mathbf{u}} \alpha_i \alpha_j)(\sum_{i=1}^n e^{c\mathbf{u}_i^\top \mathbf{u}} \alpha_i) e^{c\mathbf{u}_1^\top \mathbf{u}}}{(\sum_{i=1}^n e^{c\mathbf{u}_i^\top \mathbf{u}} \alpha_i)^4} \right]$$

$$= 2\mathbb{E}_{\{\mathbf{u}_i\}_i} \left[ \frac{(\sum_{i=1}^n e^{c\mathbf{u}_i^\top \mathbf{u}} \alpha_i) \left( \sum_{j=1}^n (\mathbf{u}_1 - \mathbf{u})^\top (\mathbf{u}_j - \mathbf{u}) e^{c\mathbf{u}_1^\top \mathbf{u} + c\mathbf{u}_j^\top \mathbf{u}} \alpha_j \right)}{(\sum_{i'=1}^n e^{c\mathbf{u}_{i'}^\top \mathbf{u}} \alpha_{i'})^3} \right]$$

$$\quad - 2\mathbb{E}_{\{\mathbf{u}_i\}_i} \left[ \frac{(\sum_{i=1}^n \sum_{j=1}^n (\mathbf{u}_i - \mathbf{u})^\top (\mathbf{u}_j - \mathbf{u}) e^{c\mathbf{u}_i^\top \mathbf{u} + c\mathbf{u}_j^\top \mathbf{u}} \alpha_i \alpha_j) e^{c\mathbf{u}_1^\top \mathbf{u}}}{(\sum_{i'=1}^n e^{c\mathbf{u}_{i'}^\top \mathbf{u}} \alpha_{i'})^3} \right]$$

$$= 2 \sum_{i=2}^n \sum_{j=1}^n S_{i,j} \tag{41}$$

where

$$S_{i,j} := \alpha_i \alpha_j \mathbb{E}_{\{\mathbf{u}_{i'}\}_{i' \in [n]}} \left[ \frac{(\mathbf{u}_1 - \mathbf{u}_i)^\top (\mathbf{u}_j - \mathbf{u}) e^{c(\mathbf{u}_1^\top \mathbf{u} + \mathbf{u}_i^\top \mathbf{u} + \mathbf{u}_j^\top \mathbf{u})}}{(\sum_{i'=1}^n e^{c\mathbf{u}_{i'}^\top \mathbf{u}} \alpha_{i'})^3} \right].$$

Note that terms with $i = 1$ do not appear in (41). We analyze $S_{i,1} + S_{i,i}$ and each $S_{i,j}, j \notin \{1, i\}$ separately, and will ultimately show that each of these terms is positive. We start with the latter case as it is easier to handle. For $j \notin \{1, i\}$, we have

$$S_{i,j} = \alpha_i \alpha_j \mathbb{E}_{\{\mathbf{u}_{i'}\}_{i' \in [n]}} \left[ \frac{(\mathbf{u}_1 - \mathbf{u}_i)^\top (\mathbf{u}_j - \mathbf{u}) e^{c(\mathbf{u}_1^\top \mathbf{u} + \mathbf{u}_i^\top \mathbf{u} + \mathbf{u}_j^\top \mathbf{u})}}{(\sum_{i'=1}^n e^{c\mathbf{u}_{i'}^\top \mathbf{u}} \alpha_{i'})^3} \right]$$

$$
= \alpha_i \alpha_j \mathbb{E}_{\{\mathbf{u}_{i'}\}_{i' \in [n]}} \left[ \frac{(\mathbf{u}_1 - \mathbf{u}_i)^\top \mathbf{u}\mathbf{u}^\top (\mathbf{u}_j - \mathbf{u}) e^{c(\mathbf{u}_1^\top \mathbf{u} + \mathbf{u}_i^\top \mathbf{u} + \mathbf{u}_j^\top \mathbf{u})}}{(\sum_{i'=1}^n e^{c\mathbf{u}_{i'}^\top \mathbf{u}} \alpha_{i'})^3} \right]
$$

$$
+ \alpha_i \alpha_j \underbrace{\mathbb{E}_{\{\mathbf{u}_{i'}\}_{i' \in [n]}} \left[ \frac{\mathbf{u}_1^\top (\mathbf{I}_k - \mathbf{u}\mathbf{u}^\top)(\mathbf{u}_j - \mathbf{u}) e^{c(\mathbf{u}_1^\top \mathbf{u} + \mathbf{u}_i^\top \mathbf{u} + \mathbf{u}_j^\top \mathbf{u})}}{(\sum_{i'=1}^n e^{c\mathbf{u}_{i'}^\top \mathbf{u}} \alpha_{i'})^3} \right]}_{=0}
$$

$$
- \alpha_i \alpha_j \underbrace{\mathbb{E}_{\{\mathbf{u}_{i'}\}_{i' \in [n]}} \left[ \frac{\mathbf{u}_i^\top (\mathbf{I}_k - \mathbf{u}\mathbf{u}^\top)(\mathbf{u}_j - \mathbf{u}) e^{c(\mathbf{u}_1^\top \mathbf{u} + \mathbf{u}_i^\top \mathbf{u} + \mathbf{u}_j^\top \mathbf{u})}}{(\sum_{i'=1}^n e^{c\mathbf{u}_{i'}^\top \mathbf{u}} \alpha_{i'})^3} \right]}_{=0}
\tag{42}
$$

$$
= \alpha_i \alpha_j \mathbb{E}_{\{\mathbf{u}_{i'}\}_{i' \in [n]}} \left[ \frac{(\mathbf{u}_1^\top \mathbf{u} - \mathbf{u}_i^\top \mathbf{u})(\mathbf{u}_j^\top \mathbf{u} - 1) e^{c(\mathbf{u}_1^\top \mathbf{u} + \mathbf{u}_i^\top \mathbf{u} + \mathbf{u}_j^\top \mathbf{u})}}{(\sum_{i'=1}^n e^{c\mathbf{u}_{i'}^\top \mathbf{u}} \alpha_{i'})^3} \right]
$$

where the latter two terms in (42) are zero by the same argument as in (33): flipping the component of either $\mathbf{u}_1$ or $\mathbf{u}_i$ perpendicular to $\mathbf{u}$ does not change any of the values in any exponent, and each flip occurs with equal probability. Next, note that if $\alpha_i = \alpha_1$,

$$
\mathbb{E}_{\{\mathbf{u}_{i'}\}_{i' \in [n]}} \left[ \frac{\mathbf{u}_1^\top \mathbf{u}(\mathbf{u}_j^\top \mathbf{u} - 1) e^{c(\mathbf{u}_1^\top \mathbf{u} + \mathbf{u}_i^\top \mathbf{u} + \mathbf{u}_j^\top \mathbf{u})}}{(\sum_{i'=1}^n e^{c\mathbf{u}_{i'}^\top \mathbf{u}} \alpha_{i'})^3} \right] = \mathbb{E}_{\{\mathbf{u}_{i'}\}_{i' \in [n]}} \left[ \frac{\mathbf{u}_i^\top \mathbf{u}(\mathbf{u}_j^\top \mathbf{u} - 1) e^{c(\mathbf{u}_1^\top \mathbf{u} + \mathbf{u}_i^\top \mathbf{u} + \mathbf{u}_j^\top \mathbf{u})}}{(\sum_{i'=1}^n e^{c\mathbf{u}_{i'}^\top \mathbf{u}} \alpha_{i'})^3} \right]
$$

thus $S_{i,j} = 0$. Otherwise, $\alpha_i < \alpha_1$ by definition of $\alpha_1$, and there must be some such $\alpha_i$, since if not, there would be some $c' \in \mathbb{R}_+$ such that $\boldsymbol{\alpha} = c' \boldsymbol{\alpha}^*$. For the case $\alpha_i < \alpha_1$, we use a symmetry argument to show that $S_{i,j} > 0$.

First we define additional notations. Let $\bar{U}_{1,i} := \{\mathbf{u}_{i'}\}_{i' \in [n] \setminus \{1,i\}}$, and for any $(a,b) \in [-1,1]^2$, define

$$
f_{a,b}(\bar{U}_{1,i}) := \frac{(a-b)(\mathbf{u}_j^\top \mathbf{u} - 1) e^{c(a+b+\mathbf{u}_j^\top \mathbf{u})}}{(e^{ca}\alpha_1 + e^{cb}\alpha_i + \sum_{i' \neq 1,i} e^{c\mathbf{u}_{i'}^\top \mathbf{u}} \alpha_{i'})^3}.
$$

In particular, for any $a \in [-1,1]$, define $p_a := \mathbb{P}_{\mathbf{u}_1}[\mathbf{u}_1^\top \mathbf{u} = a]$. Since $\mathbf{u}_1$ and $\mathbf{u}_i$ are i.i.d., we have $\mathbb{P}_{\mathbf{u}_1,\mathbf{u}_i}[\mathbf{u}_1^\top \mathbf{u} = a, \mathbf{u}_i^\top \mathbf{u} = b] = \mathbb{P}_{\mathbf{u}_1,\mathbf{u}_i}[\mathbf{u}_1^\top \mathbf{u} = b, \mathbf{u}_i^\top \mathbf{u} = a] = p_a p_b$ for any $(a,b) \in [-1,1]^2$ Thus, by the law of total expectation we have

$$
S_{i,j} = \alpha_i \alpha_j \mathbb{E}_{\bar{U}_{1,i}} \left[ \int_{-1}^1 \int_{-1}^1 f_{a,b}(\bar{U}_{1,i}) p_a p_b \, da \, db \right]
$$

$$
= \frac{\alpha_i \alpha_j}{2} \mathbb{E}_{\bar{U}_{1,i}} \left[ \int_{-1}^1 \int_{-1}^1 (f_{a,b}(\bar{U}_{1,i}) + f_{b,a}(\bar{U}_{1,i})) p_a p_b \, da \, db \right]
\tag{43}
$$

Next we show that for any instance of $a, b$ and $\bar{U}_{1,i}$, $f_{a,b}(\bar{U}_{1,i}) + f_{b,a}(\bar{U}_{1,i})$ is positive. We have:

$$
f_{a,b}(\bar{U}_{1,i}) + f_{b,a}(\bar{U}_{1,i})
$$

$$
= (a-b)(\mathbf{u}_j^\top \mathbf{u} - 1) e^{c(a+b+\mathbf{u}_j^\top \mathbf{u})}
$$

$$
\times \left( \frac{1}{(e^{ca}\alpha_1 + e^{cb}\alpha_i + \sum_{i' \neq 1,i} e^{c\mathbf{u}_{i'}^\top \mathbf{u}} \alpha_{i'})^3} - \frac{1}{(e^{cb}\alpha_1 + e^{ca}\alpha_i + \sum_{i' \neq 1,i} e^{c\mathbf{u}_{i'}^\top \mathbf{u}} \alpha_{i'})^3} \right)
$$

$$
\geq 0
$$

with equality only if $a = b$ or $\mathbf{u}_j = \mathbf{u}$, since $\mathbf{u}_j^\top \mathbf{u} \leq 1$ with equality only if $\mathbf{u}_j = \mathbf{u}$, and

$$
a > b \iff (e^{ca}\alpha_1 + e^{cb}\alpha_i + \sum_{i' \neq 1,i} e^{c\mathbf{u}_{i'}^\top \mathbf{u}} \alpha_{i'})^3 > (e^{cb}\alpha_1 + e^{ca}\alpha_i + \sum_{i' \neq 1,i} e^{c\mathbf{u}_{i'}^\top \mathbf{u}} \alpha_{i'})^3
\tag{44}
$$

due to $\alpha_1 > \alpha_i$ and $\alpha_{i'} > 0$ for all $i'$. So we have $S_{i,j} > 0$.

Next we analyze $S_{i,1} + S_{i,i}$. In these cases we cannot immediately drop the components of $\mathbf{u}_1$ and $\mathbf{u}_i$ that are perpendicular to $\mathbf{u}$. We have:

$$
S_{i,1} + S_{i,i} = \alpha_i \alpha_1 \mathbb{E}_{\{\mathbf{u}_{i'}\}_{i' \in [n]}} \left[ \frac{(\mathbf{u}_1 - \mathbf{u}_i)^\top (\mathbf{u}_1 - \mathbf{u}) e^{c(2\mathbf{u}_1^\top \mathbf{u} + \mathbf{u}_i^\top \mathbf{u})}}{(\sum_{i'=1}^n e^{c\mathbf{u}_{i'}^\top \mathbf{u}} \alpha_{i'})^3} \right]
$$

$$+ \alpha_i^2 \mathbb{E}_{\{\mathbf{u}_{i'}\}_{i' \in [n]}} \left[ \frac{(\mathbf{u}_1 - \mathbf{u}_i)^\top (\mathbf{u}_i - \mathbf{u}) e^{c(\mathbf{u}_1^\top \mathbf{u} + 2\mathbf{u}_i^\top \mathbf{u})}}{(\sum_{i'=1}^n e^{c\mathbf{u}_{i'}^\top \mathbf{u}} \alpha_{i'})^3} \right]$$

$$= \alpha_i \alpha_1 \mathbb{E}_{\{\mathbf{u}_{i'}\}_{i' \in [n]}} \left[ \frac{(1 - \mathbf{u}_i^\top \mathbf{u}_1 - \mathbf{u}_1^\top \mathbf{u} + \mathbf{u}_i^\top \mathbf{u}) e^{c(2\mathbf{u}_1^\top \mathbf{u} + \mathbf{u}_i^\top \mathbf{u})}}{(\sum_{i'=1}^n e^{c\mathbf{u}_{i'}^\top \mathbf{u}} \alpha_{i'})^3} \right]$$

$$+ \alpha_i^2 \mathbb{E}_{\{\mathbf{u}_{i'}\}_{i' \in [n]}} \left[ \frac{(\mathbf{u}_i^\top \mathbf{u}_1 - 1 - \mathbf{u}_1^\top \mathbf{u} + \mathbf{u}_i^\top \mathbf{u}) e^{c(\mathbf{u}_1^\top \mathbf{u} + 2\mathbf{u}_i^\top \mathbf{u})}}{(\sum_{i'=1}^n e^{c\mathbf{u}_{i'}^\top \mathbf{u}} \alpha_{i'})^3} \right]$$

$$= \alpha_i \mathbb{E}_{\{\mathbf{u}_{i'}\}_{i' \in [n]}} \left[ \frac{(\mathbf{u}_i^\top \mathbf{u} - \mathbf{u}_1^\top \mathbf{u}) e^{c(\mathbf{u}_1^\top \mathbf{u} + \mathbf{u}_i^\top \mathbf{u})} (e^{c\mathbf{u}_1^\top \mathbf{u}} \alpha_1 + e^{c\mathbf{u}_i^\top \mathbf{u}} \alpha_i)}{(\sum_{i'=1}^n e^{c\mathbf{u}_{i'}^\top \mathbf{u}} \alpha_{i'})^3} \right]$$

$$+ \alpha_i \mathbb{E}_{\{\mathbf{u}_{i'}\}_{i' \in [n]}} \left[ \frac{(1 - \mathbf{u}_i^\top \mathbf{u}_1) e^{c(\mathbf{u}_1^\top \mathbf{u} + \mathbf{u}_i^\top \mathbf{u})} (e^{c\mathbf{u}_1^\top \mathbf{u}} \alpha_1 - e^{c\mathbf{u}_i^\top \mathbf{u}} \alpha_i)}{(\sum_{i'=1}^n e^{c\mathbf{u}_{i'}^\top \mathbf{u}} \alpha_{i'})^3} \right]$$

Now we can split $\mathbf{u}_i^\top \mathbf{u}_1$ into the product of the components of $\mathbf{u}_i$, $\mathbf{u}_1$ in the direction $\mathbf{u}$ and the product of their components in the perpendicular subspace as before. Doing so yields

$$S_{i,1} + S_{i,i} = \alpha_i \mathbb{E}_{\{\mathbf{u}_{i'}\}_{i' \in [n]}} \left[ \frac{(\mathbf{u}_i^\top \mathbf{u} - \mathbf{u}_1^\top \mathbf{u}) e^{c(\mathbf{u}_1^\top \mathbf{u} + \mathbf{u}_i^\top \mathbf{u})} (e^{c\mathbf{u}_1^\top \mathbf{u}} \alpha_1 + e^{c\mathbf{u}_i^\top \mathbf{u}} \alpha_i)}{(\sum_{i'=1}^n e^{c\mathbf{u}_{i'}^\top \mathbf{u}} \alpha_{i'})^3} \right]$$

$$+ \alpha_i \mathbb{E}_{\{\mathbf{u}_{i'}\}_{i' \in [n]}} \left[ \frac{(1 - \mathbf{u}_i^\top \mathbf{u} \mathbf{u}^\top \mathbf{u}_1) e^{c(\mathbf{u}_1^\top \mathbf{u} + \mathbf{u}_i^\top \mathbf{u})} (e^{c\mathbf{u}_1^\top \mathbf{u}} \alpha_1 - e^{c\mathbf{u}_i^\top \mathbf{u}} \alpha_i)}{(\sum_{i'=1}^n e^{c\mathbf{u}_{i'}^\top \mathbf{u}} \alpha_{i'})^3} \right]$$

$$- \alpha_i \mathbb{E}_{\{\mathbf{u}_{i'}\}_{i' \in [n]}} \left[ \frac{\mathbf{u}_i^\top (\mathbf{I}_k - \mathbf{u} \mathbf{u}^\top) \mathbf{u}_1 e^{c(\mathbf{u}_1^\top \mathbf{u} + \mathbf{u}_i^\top \mathbf{u})} (e^{c\mathbf{u}_1^\top \mathbf{u}} \alpha_1 - e^{c\mathbf{u}_i^\top \mathbf{u}} \alpha_i)}{(\sum_{i'=1}^n e^{c\mathbf{u}_{i'}^\top \mathbf{u}} \alpha_{i'})^3} \right]$$

$$= \alpha_i \mathbb{E}_{\{\mathbf{u}_{i'}\}_{i' \in [n]}} \left[ \frac{(\mathbf{u}_i^\top \mathbf{u} - \mathbf{u}_1^\top \mathbf{u}) e^{c(\mathbf{u}_1^\top \mathbf{u} + \mathbf{u}_i^\top \mathbf{u})} (e^{c\mathbf{u}_1^\top \mathbf{u}} \alpha_1 + e^{c\mathbf{u}_i^\top \mathbf{u}} \alpha_i)}{(\sum_{i'=1}^n e^{c\mathbf{u}_{i'}^\top \mathbf{u}} \alpha_{i'})^3} \right]$$

$$+ \alpha_i \mathbb{E}_{\{\mathbf{u}_{i'}\}_{i' \in [n]}} \left[ \frac{(1 - \mathbf{u}_i^\top \mathbf{u} \mathbf{u}^\top \mathbf{u}_1) e^{c(\mathbf{u}_1^\top \mathbf{u} + \mathbf{u}_i^\top \mathbf{u})} (e^{c\mathbf{u}_1^\top \mathbf{u}} \alpha_1 - e^{c\mathbf{u}_i^\top \mathbf{u}} \alpha_i)}{(\sum_{i'=1}^n e^{c\mathbf{u}_{i'}^\top \mathbf{u}} \alpha_{i'})^3} \right]$$

Next, define

$$g_{a,b}(\bar{U}_{1,i}) := \mathbb{E}_{\{\mathbf{u}_{i'}\}_{i' \in [n]}} \left[ \frac{(\mathbf{u}_i^\top \mathbf{u} - \mathbf{u}_1^\top \mathbf{u}) e^{c(\mathbf{u}_1^\top \mathbf{u} + \mathbf{u}_i^\top \mathbf{u})} (e^{c\mathbf{u}_1^\top \mathbf{u}} \alpha_1 + e^{c\mathbf{u}_i^\top \mathbf{u}} \alpha_i)}{(\sum_{i'=1}^n e^{c\mathbf{u}_{i'}^\top \mathbf{u}} \alpha_{i'})^3} \right]$$

$$+ \mathbb{E}_{\{\mathbf{u}_{i'}\}_{i' \in [n]}} \left[ \frac{(1 - \mathbf{u}_i^\top \mathbf{u} \mathbf{u}^\top \mathbf{u}_1) e^{c(\mathbf{u}_1^\top \mathbf{u} + \mathbf{u}_i^\top \mathbf{u})} (e^{c\mathbf{u}_1^\top \mathbf{u}} \alpha_1 - e^{c\mathbf{u}_i^\top \mathbf{u}} \alpha_i)}{(\sum_{i'=1}^n e^{c\mathbf{u}_{i'}^\top \mathbf{u}} \alpha_{i'})^3} \right] \quad (45)$$

We argue similarly as in the previous case, except that here we must include additional terms.

$$S_{i,1} + S_{i,i}$$
$$= \frac{\alpha_i}{2} \mathbb{E}_{\bar{U}_{1,i}} \left[ \int_{-1}^1 \int_{-1}^1 (g_{a,b}(\bar{U}_{1,i}) + g_{b,a}(\bar{U}_{1,i})) p_a p_b \, da \, db \right]$$
$$= \frac{\alpha_i}{2} \mathbb{E}_{\bar{U}_{1,i}} \left[ \int_{-1}^1 \int_{-1}^1 G_{a,b}(\bar{U}_{1,i}) p_a p_b \, da \, db \right] \quad (46)$$

where

$$G_{a,b}(\bar{U}_{1,i}) := g_{a,b}(\bar{U}_{1,i}) + g_{b,a}(\bar{U}_{1,i}) \quad (47)$$

We show that for any $(a, b) \in [-1, 1]^2$ and any $\bar{U}_{1,i}$, $G_{a,b}(\bar{U}_{1,i})$ is positive, which implies that $S_{i,1} + S_{i,i}$ is positive by (46).

First, note that if $b = a$ for any $a \in [-1, 1]$ and $\bar{U}_{1,i}$, we have

$$g_{a,a}(\bar{U}_{1,i}) = \mathbb{E}_{\{\mathbf{u}_{i'}\}_{i' \in [n]}} \left[ \frac{(1 - a^2) e^{3ca} (\alpha_1 - \alpha_i)}{((\alpha_1 + \alpha_i) e^{ca} + \sum_{i' \in [n] \setminus \{1, i\}} e^{c\mathbf{u}_{i'}^\top \mathbf{u}} \alpha_{i'})^3} \right] \geq 0 \quad (48)$$

since each term inside the expectation is nonnegative, as $a^2 \leq 1$ and $\alpha_1 > \alpha_i$. Note that this implies $G_{a,b} \geq 0$ when $a = b$, so WLOG we consider $b \neq a$ for the remainder of the proof. Now we focus on showing (61). Throughout, we will make use of the notation

$$d_{a,b} := e^{ca}\alpha_1 + e^{cb}\alpha_i + \sum_{i' \in [n] \setminus \{1,i\}} e^{c\mathbf{u}_{i'}^\top \mathbf{u}}\alpha_{i'} \tag{49}$$

which represents the cube root of the denominator in all terms when $\mathbf{u}_1^\top \mathbf{u} = a$ and $\mathbf{u}_i^\top \mathbf{u} = b$, and

$$\gamma_{a,b} := 1 - ab + a - b.$$

Using this notation, we can rewrite

$$g_{a,b}(\bar{U}_{1,i}) = e^{c(a+b)} \frac{e^{ca}\gamma_{b,a}\alpha_1 - e^{cb}\gamma_{a,b}\alpha_i}{d_{a,b}^3} \tag{50}$$

Therefore,

$$g_{a,b}(\bar{U}_{1,i}) + g_{b,a}(\bar{U}_{1,i})$$
$$= e^{c(a+b)} \frac{e^{ca}\gamma_{b,a}\alpha_1 - e^{cb}\gamma_{a,b}\alpha_i}{d_{a,b}^3} + e^{c(a+b)} \frac{e^{cb}\gamma_{a,b}\alpha_1 - e^{ca}\gamma_{b,a}\alpha_i}{d_{b,a}^3}$$
$$= e^{c(a+b)}d_{a,b}^{-3}d_{b,a}^{-3} \left( \alpha_1 \left( e^{ca}\gamma_{b,a}d_{b,a}^3 + e^{cb}\gamma_{a,b}d_{a,b}^3 \right) - \alpha_i \left( e^{ca}\gamma_{b,a}d_{a,b}^3 + e^{cb}\gamma_{a,b}d_{b,a}^3 \right) \right)$$

Note that $e^{c(a+b)}d_{a,b}^{-3}d_{b,a}^{-3} > 0$, so it remains to show that the term inside the parentheses is positive. This term can be rearranged as:

$$\alpha_1 \left( e^{ca}\gamma_{b,a}d_{b,a}^3 + e^{cb}\gamma_{a,b}d_{a,b}^3 \right) - \alpha_i \left( e^{ca}\gamma_{b,a}d_{a,b}^3 + e^{cb}\gamma_{a,b}d_{b,a}^3 \right)$$
$$= (\alpha_1 - \alpha_i) \left( e^{ca}\gamma_{b,a}d_{b,a}^3 + e^{cb}\gamma_{a,b}d_{a,b}^3 \right)$$
$$\quad + \alpha_i \left( e^{ca}\gamma_{b,a}d_{b,a}^3 + e^{cb}\gamma_{a,b}d_{a,b}^3 - e^{ca}\gamma_{b,a}d_{a,b}^3 - e^{cb}\gamma_{a,b}d_{b,a}^3 \right)$$
$$= \underbrace{(\alpha_1 - \alpha_i) \left( e^{ca}\gamma_{b,a}d_{b,a}^3 + e^{cb}\gamma_{a,b}d_{a,b}^3 \right)}_{=:T_1} + \underbrace{\alpha_i \left( d_{b,a}^3 - d_{a,b}^3 \right) \left( e^{ca}\gamma_{b,a} - e^{cb}\gamma_{a,b} \right)}_{=:T_2} \tag{51}$$

First we show that $T_1$ is positive by analyzing $\gamma_{a,b}$ and $\gamma_{b,a}$. For any $(a,b) \in [-1,1]^2$ such that $a \neq b$,

$$\frac{\partial}{\partial b}(\gamma_{a,b}) = \frac{\partial}{\partial b}(1 - ab + a - b) = -1 - a \leq 0 \tag{52}$$

with equality holding if and only if $a = -1$. If $a = -1$, we have $\gamma_{a,b} = 1 + b - 1 - b = 0$ for all $b \in [-1,1]$. Otherwise, (52) shows that $\gamma_{a,b}$ is strictly decreasing with $b$, so it is minimized over $b \in [-1,1]$ at $b = 1$. When $b = 1$, we have $\gamma_{a,b} = 1 - a + a - 1 = 0$ for all $a$. So, $\gamma_{a,b} \geq 0$ with equality holding if and only if $a = -1$ or $b = 1$. Note that by symmetry, this implies $\gamma_{b,a} \geq 0$ with equality holding if and only if $a = 1$ or $b = -1$. So, we can have both $\gamma_{a,b} = 0$ and $\gamma_{b,a} = 0$ if and only if $a = b = -1$ or $a = b = 1$. However, we have $a \neq b$, so at least one of $\gamma_{a,b}$ and $\gamma_{b,a}$ are strictly positive, and $T_1$ is strictly positive (using also that $\alpha_1 > \alpha_i$).

We next show that $T_2$ is positive. Observe that

$$d_{b,a}^3 - d_{a,b}^3 > 0 \iff b > a \tag{53}$$

since $\alpha_1 > \alpha_i$, so it remains to show

$$b > a \iff e^{ca}\gamma_{b,a} - e^{cb}\gamma_{a,b} > 0. \tag{54}$$

where

$$e^{ca}\gamma_{b,a} - e^{cb}\gamma_{a,b} = e^{ca}(1 - ab - a + b) - e^{cb}(1 - ab + a - b). \tag{55}$$

We first show the forward direction, namely $b > a \implies e^{ca}\gamma_{b,a} - e^{cb}\gamma_{a,b} > 0$.

Note that if $b = a$, $e^{ca}\gamma_{b,a} - e^{cb}\gamma_{a,b} = 0$. So, if we can show that for any fixed $a$, $e^{ca}\gamma_{b,a} - e^{cb}\gamma_{a,b}$ is increasing with $b$ as long as $b \geq a$, then we will have $e^{ca}\gamma_{b,a} - e^{cb}\gamma_{a,b} > 0$ for $b > a$. To show $e^{ca}\gamma_{b,a} - e^{cb}\gamma_{a,b}$ is increasing, we take its partial derivative with respect to $b$:

$$\frac{\partial}{\partial b}\left(e^{ca}\gamma_{b,a} - e^{cb}\gamma_{a,b}\right) = e^{ca}(1-a) + e^{cb}(1 + a + cb - ca - c + cab) \tag{56}$$

We would like to show that the RHS of (56) is nonnegative. To do so, we show that its partial derivative with respect to $a$ is positive, so it achieves minimum value at $a = -1$, at which point the value is positive. We have:

$$\frac{\partial}{\partial a}\left(\frac{\partial}{\partial b}\left(e^{ca}\gamma_{b,a} - e^{cb}\gamma_{a,b}\right)\right) = e^{ca}(c - ca - 1) + e^{cb}(1 - c + cb)$$
$$= q(b) - q(a) \tag{57}$$

where $q(x) := e^{cx}(1 + cx - c)$. Note that $q(x)$ is monotonically increasing in $x \in [-1, 1]$; to see this, observe that

$$\frac{\partial}{\partial x}q(x) = e^{cx}(1 + cx - c)c + e^{cx}c = e^{cx}(2 + cx - c)c \geq 0 \tag{58}$$

where the inequality follows since $c \in (0, 2]$ and $x \in [-1, 1]$. Therefore, since $b > a$, we have $q(b) - q(a) \geq 0$ and $\frac{\partial}{\partial a}\left(\frac{\partial}{\partial b}\left(e^{ca}\gamma_{b,a} - e^{cb}\gamma_{a,b}\right)\right) \geq 0$ from (57). As a result, $\frac{\partial}{\partial b}\left(e^{ca}\gamma_{b,a} - e^{cb}\gamma_{a,b}\right)$ achieves minimum value at $a = -1$. At this point, using (56) we have

$$\frac{\partial}{\partial b}\left(e^{ca}\gamma_{b,a} - e^{cb}\gamma_{a,b}\right) = 2e^{-c} + e^{cb}(cb + c - c - cb)$$
$$= 2e^{-c}$$
$$> 0$$

This implies that the minimum value of $e^{ca}\gamma_{b,a} - e^{cb}\gamma_{a,b}$ over $b \in [a, 1]$ is achieved at $b = a$, and we know this value is zero, so we have that $e^{ca}\gamma_{b,a} - e^{cb}\gamma_{a,b} > 0$ when $b - a$.

To show the backward direction of (54), namely $e^{ca}\gamma_{b,a} - e^{cb}\gamma_{a,b} > 0 \implies b > a$, note that the converse, namely $a > b \implies e^{ca}\gamma_{b,a} - e^{cb}\gamma_{a,b} < 0$, follows by the same argument as above with $a$ and $b$ swapped. Therefore, we have $T_2 > 0$ as desired. $\qquad\square$

**Lemma H.4.** *Consider any* $\boldsymbol{\alpha} := [\alpha_1, \alpha_2]$ *such that* $\alpha_1 > \alpha_2 > 0$. *Further, let* $c \in (0, 1]$. *Define*

$$H_{noise}(\mathbf{u}, \boldsymbol{\alpha}) := \mathbb{E}_{\mathbf{u}_1, \mathbf{u}_2}\left[\frac{e^{2c\mathbf{u}_1^\top \mathbf{u}}\alpha_1^2 + e^{2c\mathbf{u}_2^\top \mathbf{u}}\alpha_2^2}{(e^{c\mathbf{u}_1^\top \mathbf{u}}\alpha_1 + e^{c\mathbf{u}_2^\top \mathbf{u}}\alpha_2)^2}\right].$$

*Then*

$$\frac{\partial H_{noise}(\mathbf{u}, \boldsymbol{\alpha})}{\partial \alpha_1} > 0$$

*Proof.* We have

$$H_{\text{noise}}(\mathbf{u}, \boldsymbol{\alpha}) := \mathbb{E}_{\mathbf{u}_1, \mathbf{u}_2}\left[\frac{e^{2c\mathbf{u}_1^\top \mathbf{u}}\alpha_1^2 + e^{2c\mathbf{u}_2^\top \mathbf{u}}\alpha_2^2}{(e^{c\mathbf{u}_1^\top \mathbf{u}}\alpha_1 + e^{c\mathbf{u}_2^\top \mathbf{u}}\alpha_2)^2}\right]$$

Since $n = 2$, we have

$$\frac{\partial H_{\text{noise}}(\mathbf{u}, \boldsymbol{\alpha})}{\partial \alpha_1} = \mathbb{E}_{\mathbf{u}_1, \mathbf{u}_2}\left[\frac{\partial}{\partial \alpha_1}\frac{e^{2c\mathbf{u}_1^\top \mathbf{u}}\alpha_1^2 + e^{2c\mathbf{u}_2^\top \mathbf{u}}\alpha_2^2}{(e^{c\mathbf{u}_1^\top \mathbf{u}}\alpha_1 + e^{c\mathbf{u}_2^\top \mathbf{u}}\alpha_2)^2}\right]$$

$$= \mathbb{E}_{\mathbf{u}_1, \mathbf{u}_2}\left[\frac{2e^{2c\mathbf{u}_1^\top \mathbf{u}}\alpha_1(e^{c\mathbf{u}_1^\top \mathbf{u}}\alpha_1 + e^{c\mathbf{u}_2^\top \mathbf{u}}\alpha_2)^2}{(e^{c\mathbf{u}_1^\top \mathbf{u}}\alpha_1 + e^{c\mathbf{u}_2^\top \mathbf{u}}\alpha_2)^4}\right]$$

$$- \mathbb{E}_{\mathbf{u}_1, \mathbf{u}_2}\left[\frac{2(e^{c\mathbf{u}_1^\top \mathbf{u}}\alpha_1 + e^{c\mathbf{u}_2^\top \mathbf{u}}\alpha_2)e^{c\mathbf{u}_1^\top \mathbf{u}}(e^{2c\mathbf{u}_1^\top \mathbf{u}}\alpha_1^2 + e^{2c\mathbf{u}_1^\top \mathbf{u}}\alpha_2^2)}{(e^{c\mathbf{u}_1^\top \mathbf{u}}\alpha_1 + e^{c\mathbf{u}_2^\top \mathbf{u}}\alpha_2)^4}\right]$$

$$= 2\alpha_2 \mathbb{E}_{\mathbf{u}_1, \mathbf{u}_2} \left[ \frac{e^{c(\mathbf{u}_1^\top \mathbf{u} + \mathbf{u}_2^\top \mathbf{u})}(e^{c\mathbf{u}_1^\top \mathbf{u}}\alpha_1 - e^{c\mathbf{u}_2^\top \mathbf{u}}\alpha_2)}{(e^{c\mathbf{u}_1^\top \mathbf{u}}\alpha_1 + e^{c\mathbf{u}_2^\top \mathbf{u}}\alpha_2)^3} \right]$$

Define $N := \mathbb{E}_{\mathbf{u}_1, \mathbf{u}_2} \left[ \frac{e^{c(\mathbf{u}_1^\top \mathbf{u} + \mathbf{u}_2^\top \mathbf{u})}(e^{c\mathbf{u}_1^\top \mathbf{u}}\alpha_1 - e^{c\mathbf{u}_2^\top \mathbf{u}}\alpha_2)}{(e^{c\mathbf{u}_1^\top \mathbf{u}}\alpha_1 + e^{c\mathbf{u}_2^\top \mathbf{u}}\alpha_2)^3} \right]$, and

$$d_{a,b} := e^{ca}\alpha_1 + e^{cb}\alpha_2$$

$$h_{a,b} := e^{c(a+b)} \frac{e^{ca}\alpha_1 - e^{cb}\alpha_i}{d_{a,b}^3},$$

Now, we have

$$N = \int_{-1}^{1} \int_{-1}^{1} h_{a,b} p_a p_b \, da \, db$$

$$= \frac{1}{2} \int_{-1}^{1} \int_{-1}^{1} (h_{a,b} + h_{b,a}) p_a p_b \, da \, db$$

$$= \frac{1}{2} \int_{-1}^{1} \int_{-1}^{1} (h_{a,b} + h_{b,a}) p_a p_b \chi\{a \neq b\} \, da \, db + \frac{1}{2} \int_{-1}^{1} \int_{-1}^{1} (h_{a,b} + h_{b,a}) p_a p_b \chi\{a = b\} \, da \, db$$

$$= \frac{1}{2} \int_{-1}^{1} \int_{-1}^{1} (h_{a,b} + h_{b,a}) p_a p_b \chi\{a \neq b\} \, da \, db + \int_{-1}^{1} h_{a,a} p_a^2 \, da$$

$$= \frac{1}{2} \int_{-1}^{1} \int_{-1}^{1} (h_{a,b} + h_{b,a}) p_a p_b \chi\{a \neq b\} \, da \, db + \frac{1}{2} \int_{-1}^{1} h_{a,a} p_a^2 \, da + \frac{1}{2} \int_{-1}^{1} h_{b,b} p_b^2 \, db$$

$$= \frac{1}{2} \int_{-1}^{1} \int_{-1}^{1} (h_{a,b} + h_{b,a}) p_a p_b \chi\{a \neq b\} \, da \, db + \frac{1}{4} \int_{-1}^{1} \int_{-1}^{1} h_{a,a} p_a^2 \, da \, db$$

$$\quad + \frac{1}{4} \int_{-1}^{1} \int_{-1}^{1} h_{b,b} p_b^2 \, da \, db$$

$$= \frac{1}{2} \int_{-1}^{1} \int_{-1}^{1} (h_{a,b} + h_{b,a}) p_a p_b \chi\{a \neq b\} \, da \, db + \frac{1}{4} \int_{-1}^{1} \int_{-1}^{1} (h_{a,a} p_a^2 + h_{b,b} p_b^2) \, da \, db$$

$$= \frac{1}{2} \int_{-1}^{1} \int_{-1}^{1} H_{a,b} \, da \, db \tag{59}$$

where

$$H_{a,b} := p_a p_b (h_{a,b} + h_{b,a}) + \frac{p_a^2}{2} h_{a,a} + \frac{p_b^2}{2} h_{b,b} \tag{60}$$

We will show that for any $(a, b) \in [-1, 1]^2$ and $(p_a, p_b) \in [0, 1]^2$, $H_{a,b}$ is positive, which implies that $N_i$ is positive by (59). To do this, assuming $h_{a,a}$ is nonnegative for any $a$, it is sufficient to show

$$\tilde{H}_{a,b} := h_{a,b} + h_{b,a} + \sqrt{h_{a,a} h_{b,b}} > 0, \tag{61}$$

since this implies $h_{a,b} + h_{b,a} > -\sqrt{h_{a,a} h_{b,b}}$ and thus, from (60),

$$H_{a,b} > -p_a p_b \sqrt{h_{a,a} h_{b,b}} + \frac{p_a^2}{2} h_{a,a} + \frac{p_b^2}{2} h_{b,b}$$

$$= \left( p_a \sqrt{\frac{h_{a,a}}{2}} - p_b \sqrt{\frac{h_{b,b}}{2}} \right)^2$$

$$\geq 0 \tag{62}$$

Before showing (61), we need to confirm that $h_{a,a}$ is not negative for all $a \in [-1, 1]$. We have

$$h_{a,a} = \frac{e^{3ca}(\alpha_1 - \alpha_2)}{d_{a,a}^3} \geq 0 \tag{63}$$

since each term inside the expectation is nonnegative, as $\alpha_1 > \alpha_2$. Note that this implies $H_{a,b} \geq 0$ when $a = b$, so WLOG we consider $a > b$ for the remainder of the proof.

Note that

$$h_{a,a}h_{b,b} = \frac{e^{3c(a+b)}(\alpha_1 - \alpha_i)^2}{e^{3c(a+b)}(\alpha_1 + \alpha_2)^6} = \frac{(\alpha_1 - \alpha_2)^2}{(\alpha_1 + \alpha_2)^6} \tag{64}$$

Using this, we have

$$\begin{aligned}
\tilde{H}_{a,b} &= h_{a,b} + h_{b,a} + \sqrt{h_{a,a}h_{b,b}} \\
&= \frac{e^{2ca+cb}\alpha_1 - e^{2cb+ca}\alpha_2}{d_{a,b}^3} + \frac{e^{2cb+ca}\alpha_1 - e^{2ca+cb}\alpha_2}{d_{b,a}^3} + \frac{\alpha_1 - \alpha_2}{(\alpha_1 + \alpha_2)^3} \\
&= d_{a,b}^{-3}d_{b,a}^{-3}e^{c(a+b)}(\alpha_1 + \alpha_2)^3 \\
&\quad \times \Big( \underbrace{(e^{ca}\alpha_1 - e^{cb}\alpha_2)d_{b,a}^3(\alpha_1 + \alpha_2)^3 + (e^{cb}\alpha_1 - e^{ca}\alpha_2)d_{a,b}^3(\alpha_1 + \alpha_2)^3}_{=:P} \\
&\qquad \underbrace{+ e^{-c(a+b)}d_{a,b}^3 d_{b,a}^3(\alpha_1 - \alpha_2)}_{=:P} \Big)
\end{aligned} \tag{65}$$

To show that $\tilde{H}_{a,b}$ is positive, we need to show that $P$ is positive. Without loss of generality we can consider $\alpha_1 = 1$ and $\alpha_2 \in (0, 1)$ by dividing the numerator and denominator of $H_{\text{noise}}$ by $\alpha_1^2$. Thus, for the remainder of the proof we treat $\alpha_1$ as 1 and write $\alpha := \alpha_2$ for ease of notation. Using this notation we can expand $P$ as follows:

$$\begin{aligned}
P &= (e^{ca} - e^{cb}\alpha)d_{b,a}^3(1+\alpha)^3 + (e^{cb} - e^{ca}\alpha)d_{a,b}^3(1+\alpha)^3 + e^{-c(a+b)}d_{a,b}^3 d_{b,a}^3(1-\alpha) \\
&= (e^{ca} - e^{cb}\alpha)(e^{cb} + e^{ca}\alpha)^3(1+\alpha)^3 + (e^{cb} - e^{ca}\alpha)(e^{ca} + e^{cb}\alpha)^3(1+\alpha)^3 \\
&\quad + e^{-c(a+b)}(e^{ca} + e^{cb}\alpha)^3(e^{cb} + e^{ca}\alpha)^3(1-\alpha) \\
&= (e^{5ca-cb} + e^{5cb-ca})\left(\alpha^3(1-\alpha)\right) \\
&\quad + (e^{4ca} + e^{4cb})\left(-\alpha - 5\alpha^3 + 5\alpha^4 + \alpha^6\right) \\
&\quad + (e^{3ca+cb} + e^{3cb+ca})\left(1 + 6\alpha + 10\alpha^3 - 10\alpha^4 - 6\alpha^6 - \alpha^7\right) \\
&\quad + e^{2ca+2cb}\left(1 + 5\alpha + 27\alpha^2 + 3\alpha^3 - 3\alpha^4 - 27\alpha^5 - 5\alpha^6 - \alpha^7\right) \\
&= (1-\alpha) \times \Bigg( (e^{5ca-cb} + e^{5cb-ca})\alpha^3 \\
&\quad + (e^{4ca} + e^{4cb})\left(-\alpha - \alpha^2 - 6\alpha^3 - \alpha^4 - \alpha^5\right) \\
&\quad + (e^{3ca+cb} + e^{3cb+ca})\left(1 + 7\alpha + 7\alpha^2 + 17\alpha^3 + 7\alpha^4 + 7\alpha^5 + \alpha^6\right) \\
&\quad + e^{2ca+2cb}\left(1 + 6\alpha + 33\alpha^2 + 36\alpha^3 + 33\alpha^4 + 6\alpha^5 + \alpha^6\right) \Bigg)
\end{aligned}$$

Recall that $1 - \alpha > 0$, so we need to show that the sum of the remaining terms is positive. These terms can be written as a polynomial in $y := e^{c(a-b)}$ as follows:

$$\begin{aligned}
P(1-\alpha)^{-1}e^{ca-5cb} &= y^6\alpha^3 \\
&\quad + y^5\left(-\alpha - \alpha^2 - 6\alpha^3 - \alpha^4 - \alpha^5\right) \\
&\quad + y^4\left(1 + 7\alpha + 7\alpha^2 + 17\alpha^3 + 7\alpha^4 + 7\alpha^5 + \alpha^6\right) \\
&\quad + y^3\left(1 + 6\alpha + 33\alpha^2 + 36\alpha^3 + 33\alpha^4 + 6\alpha^5 + \alpha^6\right) \\
&\quad + y^2\left(1 + 7\alpha + 7\alpha^2 + 17\alpha^3 + 7\alpha^4 + 7\alpha^5 + \alpha^6\right) \\
&\quad + y\left(-\alpha - \alpha^2 - 6\alpha^3 - \alpha^4 - \alpha^5\right) \\
&\quad + \alpha^3
\end{aligned} \tag{66}$$

We know that $y^6 > y^5 > \cdots > 1$ since $a > b$. We also have that $\alpha < 1$. Using these facts we next show that the sum of the third and smaller-order terms in the RHS of (66) is positive.

$$
\begin{aligned}
(*) := \; & y^3 \left(1 + 6\alpha + 33\alpha^2 + 36\alpha^3 + 33\alpha^4 + 6\alpha^5 + \alpha^6\right) \\
& + y^2 \left(1 + 7\alpha + 7\alpha^2 + 17\alpha^3 + 7\alpha^4 + 7\alpha^5 + \alpha^6\right) \\
& + y \left(-\alpha - \alpha^2 - 6\alpha^3 - \alpha^4 - \alpha^5\right) \\
& + \alpha^3 \\
> \; & y \left(1 + 6\alpha + 33\alpha^2 + 36\alpha^3 + 33\alpha^4 + 6\alpha^5 + \alpha^6\right) \\
& + y \left(1 + 7\alpha + 7\alpha^2 + 17\alpha^3 + 7\alpha^4 + 7\alpha^5 + \alpha^6\right) \\
& + y \left(-\alpha - \alpha^2 - 6\alpha^3 - \alpha^4 - \alpha^5\right) \\
& + \alpha^3 \\
> \; & y \left(2 + 12\alpha + 39\alpha^2 + 47\alpha^3 + 39\alpha^4 + 12\alpha^5 + 1\alpha^6\right) \\
> \; & 0
\end{aligned}
$$

Next we show that the sum of the sixth-, fifth-, and fourth-order terms is positive. Let $a_6 := \alpha^3$, $a_5 := \alpha + \alpha^2 + 6\alpha^3 + \alpha^4 + \alpha^5$, and $a_4 := 1 + 7\alpha + 7\alpha^2 + 17\alpha^3 + 7\alpha^4 + 7\alpha^5 + \alpha^6$, so the sum of the sixth-, fifth-, and fourth-order terms is $y^6 a_6 - y^5 a_5 + y^4 a_4$. Note that $32 a_6 < a_4$ since $\alpha < 1$, and

$$
\begin{aligned}
a_5 - 4a_6 &= \alpha + \alpha^2 + 2\alpha^3 + \alpha^4 + \alpha^5 \\
&= \frac{1}{7.5} \left(7.5\alpha + 7.5\alpha^2 + 15\alpha^3 + 7.5\alpha^4 + 7.5\alpha^5\right) \\
&< \frac{1}{7.5} \left(1 + 7\alpha + 7\alpha^2 + 17\alpha^3 + 7\alpha^4 + 7\alpha^5 + \alpha^6\right) \\
&= \frac{a_4}{7.5}
\end{aligned}
\tag{67}
$$

thus $a_5 < \frac{a_4}{7.5} + 4a_6$. Also, $y = e^{c(a-b)} \leq e^2 < 7.5$ since $c \leq 1$. Therefore,

$$
\begin{aligned}
y^6 a_6 - y^5 a_5 + y^4 a_4 &= y^4 \left(y^2 a_6 - y a_5 + a_4\right) \\
&> y^4 \left(y^2 a_6 - 4 y a_6 - y \frac{a_4}{7.5} + a_4\right) \\
&> y^4 \left(y^2 a_6 - 4 y a_6 + a_4 \underbrace{\left(1 - \frac{y}{7.5}\right)}_{>0 \text{ since } y < 7.5}\right) \\
&> y^4 \left(y^2 a_6 - 4 y a_6 + 32 a_6 \left(1 - \frac{y}{7.5}\right)\right) \\
&= y^4 a_6 \left(y^2 - \frac{62}{7.5} y + 32\right) \\
&> y^4 a_6 \left(-\frac{1}{4} \left(\frac{62}{7.5}\right)^2 + 32\right) \tag{68} \\
&> 0 \tag{69}
\end{aligned}
$$

where (68) follows by minimizing the terms inside the parentheses over $y$. Thus, we have $\tilde{H}_{a,b} > 0$, which completes the proof. $\qquad\square$

Now we can finally prove Theorem 4.4. We prove a slightly stronger result, formally stated as follows.

**Theorem H.5.** *Consider any* $\mathbf{B} \in \mathbb{O}^{d \times k}$ *and the corresponding function class* $\mathcal{F}_{\mathbf{B}}^{lin}$ *as defined in* (4.2). *Suppose tasks are drawn from* $D(\mathcal{F}_{\mathbf{B}}^{lin})$ *and Assumption 4.3 holds. Recall the pretraining population loss:*

$$
\mathcal{L}(\mathbf{M}) = \mathbb{E}_{f, \{\mathbf{x}_i\}_{i \in [n+1]}, \{\epsilon_i\}_{i \in [n]}} \left[\left(\frac{\sum_{i=1}^{n} (f(\boldsymbol{x}_i) - f(\mathbf{x}_{n+1}) + \epsilon_i) e^{\mathbf{x}_i^\top \mathbf{M} \mathbf{x}_{n+1}}}{\sum_{i=1}^{n} e^{\mathbf{x}_i^\top \mathbf{M} \mathbf{x}_{n+1}}}\right)^2\right]. \tag{70}
$$

*Consider two cases:*

- *Case 1: $\sigma = 0$, $n > 1$. Then define $C_p := 2$.*
- *Case 2: $\sigma > 0$, $n = 2$. Then define $C_p := 1$.*

*Then in each case, among all $\mathbf{M} \in \mathcal{M} := \{\mathbf{M} \in \mathbb{R}^{d \times d} : \mathbf{M} = \mathbf{M}, \|\mathbf{B}^\top \mathbf{M} \mathbf{B}\|_2 \leq \frac{C_p}{c_u^2}\}$, any minimizer $\mathbf{M}^*$ of (70) satisfies $\mathbf{M}^* = c\mathbf{B}\mathbf{B}^\top$ for some $c \in (0, \frac{C_p}{c_u^2}]$.*

*Proof.* From Lemma H.2, we have $\mathbf{M}^* = c_p \mathbf{B}\mathbf{B}^\top + \tilde{c} mathbf{B}_\perp \mathbf{B}_\perp^\top$ for some $\tilde{c} \in \mathbb{R}$ and some $c_p \in (0, \frac{C_p}{c_u^2}]$, where $C_p = 2$ in Case 1 and $C_p = 1$ in Case 2. Suppose that $\tilde{c} \neq 0$. Then it remains to show that $\mathcal{L}(c_p \mathbf{B}\mathbf{B}^\top + \tilde{c}\mathbf{B}_\perp \mathbf{B}_\perp^\top) > \mathcal{L}(c_p \mathbf{B}\mathbf{B}^\top)$.

We start by establishing the same notations as in the proof of Lemma H.2. For each $i \in [n+1]$, $\mathbf{x}_i = c_u B\mathbf{u}_i + c_v \mathbf{B}_\perp \mathbf{v}_i$. Thus, for each $i \in [n]$, we have

$$
\begin{aligned}
e^{\mathbf{x}_i^\top \mathbf{M}\mathbf{x}_{n+1}} &= e^{c_p \mathbf{x}_i^\top \mathbf{B}\mathbf{B}^\top \mathbf{x}_{n+1}} e^{c' \mathbf{x}_i^\top \mathbf{B}_\perp \mathbf{B}_\perp^\top \mathbf{x}_{n+1}} \\
&= e^{c_p c_u^2 \mathbf{u}_i^\top \mathbf{u}_{n+1}} e^{c_v^2 \tilde{c} \mathbf{v}_i^\top \mathbf{v}_{n+1}} \\
&= e^{c_p c_u^2 \mathbf{u}_i^\top \mathbf{u}_{n+1}} \alpha_i
\end{aligned}
\tag{71}
$$

where, for each $i \in [n]$, $\alpha_i := e^{c_v^2 \tilde{c} \mathbf{v}_i^\top \mathbf{v}_{n+1}}$. For ease of notation, denote $\mathbf{x} = \mathbf{x}_{n+1}$, $\mathbf{u} := \mathbf{u}_{n+1}$ and $c = c_p c_u^2$. Also, note that for any $\mathbf{x}_i$, $f(\mathbf{x}_i) = \mathbf{a}^\top \mathbf{B}^\top \mathbf{x}_i = c_u \mathbf{a}^\top \mathbf{u}_i$, and that drawing $f \sim D(\mathcal{F}_\mathbf{B}^{\text{lin}})$ is equivalent to drawing $\mathbf{a} \sim D_\mathbf{a}$ for some distribution $D_\mathbf{a}$ over $\mathbb{R}^k$ such that $\mathbb{E}_{\mathbf{a} \sim D_\mathbf{a}}[\mathbf{a}\mathbf{a}^\top] = c_a^2 \mathbf{I}_k$. Using this, we have:

$$
\mathcal{L}(c_p \mathbf{B}\mathbf{B}^\top + \tilde{c}\mathbf{B}_\perp \mathbf{B}_\perp^\top)
$$

$$
= \mathbb{E}_{\mathbf{a}, \mathbf{u}, \{\mathbf{u}_i\}_{i \in [n]}, \{\alpha_i\}_{i \in [n]}, \{\epsilon_i\}_{i \in [n]}} \left[ \frac{\left(\sum_{i=1}^n (c_u \mathbf{a}^\top \mathbf{u}_i - c_u \mathbf{a}^\top \mathbf{u} + \epsilon_i) e^{c\mathbf{u}_i^\top \mathbf{u}} \alpha_i\right)^2}{\left(\sum_{i=1}^n e^{c\mathbf{u}_i^\top \mathbf{u}} \alpha_i\right)^2} \right]
$$

$$
= \mathbb{E}_{u, \{\mathbf{u}_i\}_{i \in [n]}, \{\alpha_i\}_{i \in [n]}}
$$
$$
\left[ \frac{\sum_{i=1}^n \sum_{j=1}^n \mathbb{E}_{\mathbf{a}, \{\epsilon_i\}_{i \in [n]}}[(c_u \mathbf{a}^\top \mathbf{u}_i - c_u \mathbf{a}^\top \mathbf{u} + \epsilon_i)(c_u \mathbf{a}^\top \mathbf{u}_j - c_u \mathbf{a}^\top \mathbf{u} + \epsilon_j)] e^{c\mathbf{u}_i^\top \mathbf{u} + c\mathbf{u}_j^\top \mathbf{u}} \alpha_i \alpha_j}{(\sum_{i=1}^n e^{c\mathbf{u}_i^\top \mathbf{u}})^2} \right]
$$

$$
= \mathbb{E}_{u, \{\mathbf{u}_i\}_{i \in [n]}, \{\alpha_i\}_{i \in [n]}}
$$
$$
\left[ c_a^2 c_u^2 \frac{\sum_{i=1}^n \sum_{j=1}^n (\mathbf{u}_i - \mathbf{u})^\top (\mathbf{u}_j - \mathbf{u}) e^{c\mathbf{u}_i^\top \mathbf{u} + c\mathbf{u}_j^\top \mathbf{u}} \alpha_i \alpha_j}{(\sum_{i=1}^n e^{c\mathbf{u}_i^\top \mathbf{u}})^2} + \sigma^2 \frac{\sum_{i=1}^n e^{2c\mathbf{u}_i^\top \mathbf{u}} \alpha_i \alpha_j}{(\sum_{i=1}^n e^{c\mathbf{u}_i^\top \mathbf{u}})^2} \right]
$$

$$
= \mathbb{E}_{\mathbf{u}, \boldsymbol{\alpha}} [H(\mathbf{u}, \boldsymbol{\alpha})]
$$

where $\boldsymbol{\alpha} := [\alpha_1, \ldots, \alpha_n]$ and

$$
H(\mathbf{u}, \boldsymbol{\alpha})
$$
$$
:= \mathbb{E}_{\{\mathbf{u}_i\}_{i \in [n]}} \left[ c_a^2 c_u^2 \frac{\sum_{i=1}^n \sum_{j=1}^n (\mathbf{u}_i - \mathbf{u})^\top (\mathbf{u}_j - \mathbf{u}) e^{c\mathbf{u}_i^\top \mathbf{u} + c\mathbf{u}_j^\top \mathbf{u}} \alpha_i \alpha_j}{(\sum_{i=1}^n e^{c\mathbf{u}_i^\top \mathbf{u}} \alpha_i)^2} + \sigma^2 \frac{\sum_{i=1}^n e^{2c\mathbf{u}_i^\top \mathbf{u}} \alpha_i^2}{(\sum_{i=1}^n e^{c\mathbf{u}_i^\top \mathbf{u}} \alpha_i)^2} \right].
$$
$$
\tag{72}
$$

Define $\boldsymbol{\alpha}^* = [1, \ldots, 1] \in \mathbb{R}^n$. We proceed by showing that for any $\mathbf{u} \in \mathbb{S}^{d-1}$, all $\boldsymbol{\alpha} \in \mathbb{R}_+^n$ satisfy

- (i)  if $\boldsymbol{\alpha} = c' \boldsymbol{\alpha}^*$ for some $c' \in \mathbb{R}_+$, then $H(\mathbf{u}, \boldsymbol{\alpha}) = H(u, \boldsymbol{\alpha}^*)$
- (ii)  if $\boldsymbol{\alpha} \neq c' \boldsymbol{\alpha}^*$ for any $c' \in \mathbb{R}_+$, then $H(\mathbf{u}, \boldsymbol{\alpha}) > H(u, \boldsymbol{\alpha}^*)$

This implies $\mathcal{L}(c_p \mathbf{B}\mathbf{B}^\top + \tilde{c}\mathbf{B}_\perp \mathbf{B}_\perp^\top) > \mathcal{L}(c_p \mathbf{B}\mathbf{B}^\top)$, since

$$
\mathbb{P}_{\boldsymbol{\alpha}}(\{\boldsymbol{\alpha} = c' \boldsymbol{\alpha}^* \text{ for some } c' \in \mathbb{R}_+\}) = 1 \iff \tilde{c} = 0,
$$

which implies that $\tilde{c} = 0$ is the unique argument that achieves the minimal value of $\mathcal{L}(c_p \mathbf{B}\mathbf{B}^\top + \tilde{c}\mathbf{B}_\perp \mathbf{B}_\perp^\top)$ over $\tilde{c} \in \mathbb{R}$ (and this value is $\mathbb{E}_\mathbf{u}[H(\mathbf{u}, \boldsymbol{\alpha}^*)]$).

Proving $(i)$ is trivial as it can be easily checked that $H(\mathbf{u}, \boldsymbol{\alpha}) = H(\mathbf{u}, c'\boldsymbol{\alpha})$ for all $\mathbf{u} \in \mathbb{S}^{d-1}$, $\boldsymbol{\alpha} \in \mathbb{R}_+^n$, and $c' \in \mathbb{R}_+$.

Proving $(ii)$ is more involved. Consider any $\boldsymbol{\alpha} \neq c'\boldsymbol{\alpha}^*$ for any $c' \in \mathbb{R}_+$. WLOG let $1 \in \arg\max_i \alpha_i$. We show that the partial derivative of $H(\mathbf{u}, \boldsymbol{\alpha})$ with respect to $\alpha_1$ is strictly positive, which means that $H(\mathbf{u}, \boldsymbol{\alpha})$ can be reduced by reducing $\alpha_1$ by some $\epsilon > 0$. We can repeat this argument, repeatedly reducing $\max_i \alpha_i$ at each step and thereby reducing the loss, until we reach an $\boldsymbol{\alpha}'$ satisfying $\boldsymbol{\alpha}' = c'\boldsymbol{\alpha}^*$. Since the loss is reduced at each step, we have that $H(\mathbf{u}, \boldsymbol{\alpha}) > H(\mathbf{u}, \boldsymbol{\alpha}^*)$.

To show that the partial derivative of $H(\mathbf{u}, \boldsymbol{\alpha})$ with respect to $\alpha_1$ is strictly positive, we decompose $\frac{\partial H(\mathbf{u}, \boldsymbol{\alpha})}{\partial \alpha_1} = \frac{\partial H_{\text{signal}}(\mathbf{u}, \boldsymbol{\alpha})}{\partial \alpha_1} + \frac{\partial H_{\text{noise}}(\mathbf{u}, \boldsymbol{\alpha})}{\partial \alpha_1}$, where

$$H_{\text{signal}}(\mathbf{u}, \boldsymbol{\alpha}) := c_a^2 c_u^2 \mathbb{E}_{\{\mathbf{u}_i\}_{i \in [n]}} \left[ \frac{\sum_{i=1}^n \sum_{j=1}^n (\mathbf{u}_i - \mathbf{u})^\top (\mathbf{u}_j - \mathbf{u}) e^{c\mathbf{u}_i^\top \mathbf{u} + c\mathbf{u}_j^\top \mathbf{u}} \alpha_i \alpha_j}{(\sum_{i=1}^n e^{c\mathbf{u}_i^\top \mathbf{u}} \alpha_i)^2} \right]$$

$$H_{\text{noise}}(\mathbf{u}, \boldsymbol{\alpha}) := \sigma^2 \mathbb{E}_{\{\mathbf{u}_i\}_{i \in [n]}} \left[ \frac{\sum_{i=1}^n e^{2c\mathbf{u}_i^\top \mathbf{u}} \alpha_i^2}{(\sum_{i=1}^n e^{c\mathbf{u}_i^\top \mathbf{u}} \alpha_i)^2} \right]$$

By Lemma H.3, we have $\frac{\partial H_{\text{signal}}(\mathbf{u}, \boldsymbol{\alpha})}{\partial \alpha_1} > 0$. If $\sigma = 0$ we are done, otherwise we have $n = 2$ and $\frac{\partial H_{\text{noise}}(\mathbf{u}, \boldsymbol{\alpha})}{\partial \alpha_1} > 0$ by Lemma H.4. This completes the proof. $\qquad\square$

# I   Additional Lemmas

**Lemma I.1.** *Consider a continuous unimodal function $f$. Then we have*

$$\sum_{i=0}^\infty f(i) - \max f \leq \int_0^\infty f(t)dt \leq \sum_{i=1}^\infty f(i) + \max f$$

*Proof.* Let $T$ denote the point that achieves the maximum of $f$. Then we know that $f(t) \geq f(\lfloor t \rfloor)$ for $t < T$, while $f(t) \geq f(\lceil t \rceil)$ for $t > T$. This means $\int_{i-1}^i f(t)dt \leq f(i) \leq \int_i^{i+1} f(t)dt$ for $t \leq \lfloor T \rfloor$ and $\int_{i-1}^i f(t)dt \geq f(i) \geq \int_i^{i+1} f(t)dt$ for $t \geq \lceil T \rceil$ So

$$\sum_{i=0}^\infty f(i) = \sum_{i=0}^{\lfloor T \rfloor} f(i) + \sum_{i=\lceil T \rceil}^\infty f(i)$$

$$\leq \sum_{i=0}^{\lfloor T \rfloor} \int_i^{i+1} f(t)dt + \sum_{\lceil T \rceil}^\infty \int_{i-1}^i f(t)dt$$

$$\leq \sum_{i=0}^\infty \int_i^{i+1} f(t)dt + \int_{\lfloor T \rfloor}^{\lceil T \rceil} f(t)dt$$

$$\leq \int_0^\infty f(t)dt + \max f$$

Similarly we have

$$\sum_{i=1}^\infty f(i) = \sum_{i=1}^{\lfloor T \rfloor} f(i) + \sum_{i=\lceil T \rceil}^\infty f(i)$$

$$\leq \sum_{i=1}^{\lfloor T \rfloor} \int_{i-1}^i f(t)dt + \sum_{\lceil T \rceil}^\infty \int_i^{i+1} f(t)dt$$

$$\leq \sum_{i=1}^{\infty} \int_{i-1}^{i} f(t)dt - \int_{\lfloor T \rfloor}^{\lceil T \rceil} f(t)dt$$

$$\leq \int_{0}^{\infty} f(t)dt - \max f$$

$\square$

**Lemma I.2.** *If $f$ and $g$ are nonnegative measurable real functions, then*

$$\int f(x)g(x)dx \leq \int f^*(x)g^*(x)dx$$

*where $f^*, g^*$ are the symmetric decreasing rearrangements of $f$ and $g$.*

*Proof.* Please see [66] or [67]. $\square$

**Lemma I.3.** *Suppose $\{a_i\}, \{b_i\}$ are sorted the same way, $a_i > a_j \iff b_i > b_j$. Then we have*

$$\frac{\sum a_i^2}{\left(\sum a_i\right)^2} < \frac{\sum a_i^2 b_i^2}{\left(\sum a_i b_i\right)^2}.$$

*Proof.* Cross multiplying and expanding, we have

$$\left(\sum a_i^2\right)\left(\sum a_i b_i\right)^2 < \left(\sum a_i^2 b_i^2\right)\left(\sum a_i\right)^2$$

$$\iff \sum_{i,j,k} a_i b_i a_j b_j a_k^2 < \sum_{i,j,k} a_i^2 b_i^2 a_j a_k$$

$$\iff \frac{1}{3}\sum_{i,j,k} a_i b_i a_j b_j a_k^2 + a_j b_j a_k b_k a_i^2 + a_k b_k a_i b_i a_j^2 < \frac{1}{3}\sum_{i,j,k} a_i^2 b_i^2 a_j a_k + a_j^2 b_j^2 a_k a_i + a_k^2 b_k^2 a_i a_j$$

$$\iff \frac{1}{3}\sum_{i,j,k} a_i^2 b_i^2 a_j a_k + a_j^2 b_j^2 a_k a_i + a_k^2 b_k^2 a_i a_j - \left(a_i b_i a_j b_j a_k^2 + a_j b_j a_k b_k a_i^2 + a_k b_k a_i b_i a_j^2\right) > 0$$

$$\iff \frac{1}{3}\sum_{i,j,k} a_i a_j a_k \left(a_i b_i^2 + a_j b_j^2 + a_k b_k^2 - a_i b_j b_k - a_j b_k b_i - a_k b_i b_j\right) > 0$$

The last of which follows from the rearrangement inequality [67]. $\square$

## J   Additional Experiments and Details

All experiments were run in Google Colab in a CPU runtime. We used a random seed of 0 in all cases. All training was executed in PyTorch with the Adam optimizer. We tuned learning rates in $\{10^{-3}, 10^{-2}, 10^{-1}\}$ separately for linear and softmax attention, and we initialized $\mathbf{M}_K$ and $\mathbf{M}_Q$ by setting each to $0.001\mathbf{I}_d$, and tie the weights of $\mathbf{M}_K$ and $\mathbf{M}_Q$ to speed up training.

**Figure 1.** The upper row depicts our functions, which increase in Lipschitzness from left to right. The black curve depicts the ground truth, while the gray dots depict the noisy training samples. The shaded region represents the attention window. The middle row depicts the attention weights for softmax and linear attention. We remark that the softmax is able to adapt to the Lipschitzness while linear is not. The bottom row depicts the ICL error as a function of the context length $n$ for Linear and ReLU pretraining using Linear and Softmax attention. That is, at each iteration, a context is drawn from a non-linear regression (defined below) consisting of a randomly phase shifted cosine function. The ICL task is to predict the function value at a randomly chosen query on the unit circle. Each point in the plot depicts the ICL error of a pretrained attention unit (using softmax (blue) or linear (orange) activation) at the end of 15000 iterations with learning rate $10^{-3}$. We use $d = 2$ and a distribution $D(\mathcal{F}_{\nu,\text{hills}})$. Here we define

$$\mathcal{F}_{\nu,\text{hills}} = \{\nu \cos(\theta - b)\}$$

and a distribution $D(\mathcal{F}_{\nu,\text{hills}})$ is induced by drawing $b$ uniformly from $[-\pi, \pi]$. We use $\nu = 0, 1.5, 6$ for the left, middle and right plots in the bottom row, respectively.

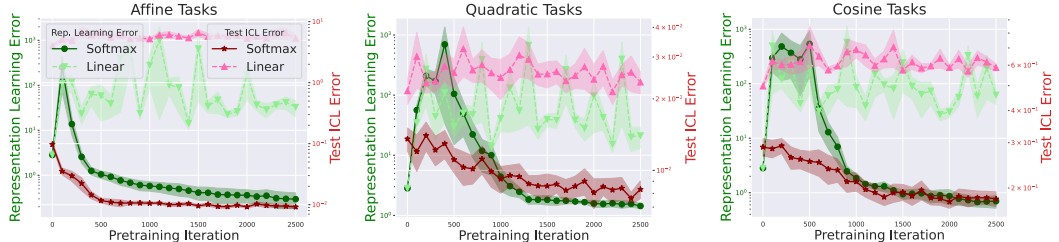

Figure 9: Representation learning error ($\rho(\mathbf{M}, \mathbf{B})$) and test ICL error (mean squared error) during pretraining softmax and linear attention on tasks from **Left:** $\mathcal{F}_{\mathbf{B}}^{\text{aff}}$, **Center:** $\mathcal{F}_{\mathbf{B}}^2$, and **Right:** $\mathcal{F}_{\mathbf{B}}^{\text{cos}}$.

**Figures 3, 4, 5.** In all cases, we use an exponentially decaying learning rate schedule with factor 0.999. In Figures 3 and 5 we use initial learning rate 0.1 and in Figure 4 we use an initial learning rate 0.01. Moreover, in all cases besides those with varying $n$ in Figure 4, we compute gradients with respect to the ICL loss evaluated on $N := \lfloor\sqrt{n}\rfloor$ query samples per task (that is, each context input to the attention unit has $n + N$ samples, of which $n$ are labeled, and the other $N$ labels are inferred). When $n$ varies in Figure 4, we use $N = 1$. In Figure 5 we show smoothed test ICL errors with smoothing rate 0.01.

## J.1    Low-Rank Experiments

Due to our results in Section 3 showing that softmax attention can learn an appropriate attention window scale when pretrained on nonlinear tasks, we hypothesize that it can also learn the appropriate *directions* during pretraining on nonlinear tasks. To test this, we consider tasks drawn from low-rank versions of affine, quadratic and cosine function classes, in particular: $\mathcal{F}_{\mathbf{B}}^{\text{aff}} := \{f : f(\boldsymbol{x}) = \mathbf{a}^\top \mathbf{B}^\top \boldsymbol{x} + 2, \mathbf{a} \in \mathbb{S}^{k-1}\}$, $\mathcal{F}_{\mathbf{B}}^2 := \{f : f(\boldsymbol{x}) = (\mathbf{a}^\top \mathbf{B}^\top \boldsymbol{x})^2, \mathbf{a} \in \mathbb{S}^{k-1}\}$ and $\mathcal{F}_{\mathbf{B}}^{\text{cos}} := \{f : f(\boldsymbol{x}) = \cos(4\mathbf{a}^\top \mathbf{B}^\top \boldsymbol{x}), \mathbf{a} \in \mathbb{S}^{k-1}\}$. Each task distribution $D(\mathcal{F}_{\mathbf{B}}^{\text{aff}}), D(\mathcal{F}_{\mathbf{B}}^2), D(\mathcal{F}_{\mathbf{B}}^{\text{cos}})$ is induced by drawing $\mathbf{a} \sim \mathcal{U}^k$. We train $\mathbf{M}_K$ and $\mathbf{M}_Q$ with Adam with learning rate tuned separately for softmax and linear attention. We set $d = 10$, $k = 2$, $n = 50$, and $\sigma = 0.01$. We draw $\{\boldsymbol{x}_i\}_{i=1}^{n+1}$ i.i.d. from a non-uniform distribution on $\mathbb{S}^{d-1}$ for each task, and draw one task per training iteration. We draw $\mathbf{B}$ randomly at the start of each trial, and repeat each trial 5 times and plots means and standard deviations over the 5 trials. We capture the extent to which the learned $\mathbf{M} = \mathbf{M}_K^\top \mathbf{M}_Q$ recovers $\text{col}(\mathbf{B})$ via the metric $\rho(\mathbf{M}, \mathbf{B}) := \frac{\|\mathbf{B}_\perp^\top \mathbf{M} \mathbf{B}_\perp\|_2}{\sigma_{\min}(\mathbf{B}^\top \mathbf{M} \mathbf{B})}$, where $\sigma_{\min}(\mathbf{A})$ is the minimum singular value of $\mathbf{A}$. For test error, we compute the average squared error on 500 random tasks drawn from the same distribution as the (pre)training tasks. Please see Appendix J for more details.

We randomly generate $\mathbf{B}$ on each trial by first sampling each element of $\hat{\mathbf{B}}$ i.i.d. from the standard normal distribution, then take its QR decomposition to obtain $\mathbf{B}$. To draw the covariates, we draw a random matrix $\tilde{\mathbf{J}} \in \mathbb{R}^{d \times d}$ by sampling each element i.i.d. from the standard normal distribution. Then, we compute $\mathbf{J} = (\tilde{\mathbf{J}}^\top \tilde{\mathbf{J}})^{1/2}$. Then we draw $\tilde{\boldsymbol{x}}_i \sim \mathcal{N}(\mathbf{0}_d, \mathbf{I}_d)$ and set $\boldsymbol{x}_i = \frac{\mathbf{J}\tilde{\boldsymbol{x}}_i}{\|\mathbf{J}\tilde{\boldsymbol{x}}_i\|}$.

**Results.** Figure 9 shows that softmax attention recovers the low-rank structure when tasks are drawn from each of the three function classes, which leads to test error improving with the quality of the learned subspace. In contrast, linear attention does not learn any meaningful structure in these cases.

