# OpenReview forum: "In-Context Learning with Transformers: Softmax Attention Adapts to Function Lipschitzness"
_NeurIPS.cc/2024/Conference — NeurIPS 2024 spotlight_

### Official Review · Reviewer_ftLf · 2024-07-05

**Soundness:** 4
**Presentation:** 4
**Contribution:** 4
**Rating:** 8
**Confidence:** 3

**Summary:**

This study examines the impact of softmax attention on ICL regression, advancing beyond the typical linear treatment of the topic. In particular, the authors find that 1) softmax attention leads the model to adapt to the target function's Lipschitz constant and 2) enables the model to recover low dimensional structure in the target.

**Strengths:**

The paper is very well-written and easy to read. It's great to see work that pushes beyond the usual linear treatment of attention, and explicitly considers the role of the softmax nonlinearity. The connection to kernel smoothing is very neat, and the resulting interpretation of softmax attention as an adaptive process sensitive to the target's Lipschitz constant is very insightful. I was particularly surprised that Lipschitz adaptation is both necessary and sufficient to illustrate the role of softmax attention in ICL regression, and that swapping function classes did not matter so long as the Lipschitz constant remained the same. Very cool work overall. Well done!

**Weaknesses:**

I found no substantive weaknesses in the analysis, but do have some follow-up question listed below. As with any theory paper, there's always more that can be done, but I think the level of work demonstrated in this paper is more than sufficient to merit a strong accept.

**Questions:**

- Could your results transfer in any way to an ICL classification setting, for instance like the one in [Reddy 2024](https://arxiv.org/abs/2312.03002) among others? Your Lipschitz adaptation argument, particularly as it controls a "window size" over inputs, sounds obliquely related to the way softmax attention is sometimes discussed as implementing "selection" or "copy" operations over context examples in ICL classification. I'm curious if there could be a deeper connection here?
- When you train models in practice, do you find that the QKV matrices of the final model become like the M-parameterization of the matrices you highlight in eq. 4?

**Limitations:**

The authors adequately address their limitations.

---

> ### Author Rebuttal · Authors · 2024-08-06
>
> We thank the reviewer for their valuable feedback. Please find the response below, and a global response above.
>
> 1. **Do trained models have the same decomposition for $\mathbf{W}_K^\top \mathbf{W}_Q$ as in
>     Equation (4)?** We indeed observe this decomposition  empirically, please see the attached pdf in the global rebuttal for examples of $\mathbf{W}_K^\top \mathbf{W}_Q$ trained on ReLU functions of varying ambient dimensions.
>
> 2. **Classification problems and selection.**
>     Please see our global response for some discussion of how our results may be extended to classification.
>     Regarding the selection idea: One framework that is used to interpret ICL in language is that of the ``induction head", which allows a transformer to implement an algorithm that searches for a previous occurrence of a token and repeating its following token. Concretely, if the context provided is $\{... \texttt{[A]}, \texttt{[B]}, ... \texttt{[A']}\}$, the network outputs $\texttt{[B']}$ such that $\texttt{[A]}$ is to $\texttt{[B]}$ analogously what $\texttt{[A']}$ is to $\texttt{[B']}$. While this is outside the scope of our work, one of the intuitions from our work is that the soft-max attention can help inform this analogy in two ways: (1) by informing a notion of "distance" that constitutes a significance of match between $[\texttt{A}]$ and $[\texttt{A'}]$, and (2) by selecting a subspace within which to calculate attention (in case such structure exists).

---

> > ### Comment · Reviewer_ftLf · 2024-08-08
> >
> > Thank you for your rebuttal and clarifying details. Well done overall! I continue to recommend acceptance, and maintain my current score.

---

### Official Review · Reviewer_BJxR · 2024-07-12

**Soundness:** 3
**Presentation:** 2
**Contribution:** 3
**Rating:** 7
**Confidence:** 3

**Summary:**

This paper analyzes how softmax attention learns to perform in-context learning (ICL) through pretraining. The authors show that softmax attention adapts its "attention window" based on the Lipschitzness and noise characteristics of the pretraining tasks. They provide theoretical analysis for affine and ReLU-based function classes, demonstrating that softmax attention learns an optimal trade-off between bias and variance. The paper also explores how softmax attention can recover low-dimensional structure in the input space. Experiments are conducted to validate the theoretical findings.

**Strengths:**

- The paper addresses an important and timely topic in understanding the mechanisms behind in-context learning in transformer models. This is a crucial area of research given the widespread adoption of large language models. The paper makes a novel connection between softmax attention and nearest neighbor regression, providing an intuitive interpretation of how ICL works in this setting.
- The analysis is clean and the main message is clear: softmax activation plays a critical role in enabling effective in-context learning. For example, Theorem 3.4 provides concrete bounds on how the attention window scales with task Lipschitzness, noise level, and context size. Also, I like Theorem 3.5, whose test data are any arbitrary L−Lipschitz task, which is pretty general ICL setting.
- The experimental results generally support the theoretical analysis. Figure 3 nicely illustrates how the spectral norm of M (representing the inverse of the attention window size) increases with the Lipschitz constant L, aligning with the theory.

**Weaknesses:**

- There's a potential gap between the paper's main message and practical ICL scenarios. The analysis is based on a nearest-neighbor interpretation, but in real-world applications, LLMs often perform well even when there are no "close" examples in the context.
- Some of the experimental writing and presentation could be improved. Figures 4 and 5 are particularly hard to follow without careful reading of the appendix. It would be helpful to have more detailed captions and clearer explanations in the main text.
- The theoretical analysis is limited to relatively simple function classes (affine and ReLU-based). While this provides valuable insights, it's not clear how well these results generalize to the more complex functions learned by large language models in practice.

**Questions:**

- How do you think your analysis might extend to more complex, hierarchical function classes that might better represent the capabilities of large language models? How can it be generally applied when there is no neighbor data?
- Have you considered how your findings might relate to or explain the emergent capabilities observed in large language models as they scale?

**Limitations:**

The authors acknowledge some limitations in Section 5, including that their model only considers the output of a single layer of attention and that establishing a mathematical framework for priming effects in LLMs remains an open challenge. These are important limitations to note, as they highlight the gap between this analysis and the full complexity of modern language models.

---

> ### Author Rebuttal · Authors · 2024-08-06
>
> We thank the reviewer for their valuable feedback. Please find the response below, and a global response above.
>
> 1. **No ``close" examples, simple function classes.** Please see the global rebuttal (Points 1 and 2). To summarize, it is possible that the intermediate layers of the model can learn the notion of closeness that it needs, but this is outside the scope of our analysis of a single layer.
>
> 2. **Writing improvements.** We thank the reviewer for this feedback. Should the paper be accepted, we will make these revisions in the camera-ready version with the extra space provided.
>
> 3. **How can our results be extended to hierarchical function classes?** Our analysis in Section 4 begins to answer this question. When the ground-truth functions are a hierarchy consisting of a linear projection followed by a linear link function, the attention weights learn the direction-wise Lipschitzness of the class, that is, they learn this projection. Based on our intuition and experiments in Figure 10 in Appendix J, we suspect that when the functions may have nonlinear link functions (which are still preceded by a shared linear projection), the attention weights again learn the projection and implement a nearest neighbors regressor in the range of the projection with appropriate neighborhood size based on the Lipschitzness of the link functions and noise level. More complex hierarchies likely require additional layers to learn; this is an interesting direction for future work.

---

> > ### Comment · Reviewer_BJxR · 2024-08-08
> > **Thanks. Update score from 6 to 7**
> >
> > I have checked all the rebuttals. They address all my concerns well, and I have updated my score from 6 to 7.

---

### Official Review · Reviewer_hHKc · 2024-07-15

**Soundness:** 4
**Presentation:** 4
**Contribution:** 3
**Rating:** 7
**Confidence:** 5

**Summary:**

This paper explores how softmax attention in transformer models enables in-context learning (ICL), where a model can adapt to solve new tasks using only a few input examples without additional training. The authors focus specifically on regression tasks, where the model must predict a continuous value given some input features. They show that during pretraining on a distribution of ICL regression tasks, softmax attention learns to implement a nearest neighbors predictor that is adapted to properties of the pretraining task distribution.

 The key insight is that softmax attention learns an "attention window" - a neighborhood around each input query point that determines which other input points influence the prediction. The size and shape of this attention window adapts based on properties of the pretraining tasks, specifically their Lipschitzness (how quickly function values can change) and the amount of label noise. Importantly, the authors demonstrate that learning this adapted attention window is crucial for generalization. The authors also prove that softmax attention can learn to project inputs onto a relevant low-dimensional subspace when the pretraining tasks depend only on projections of the inputs onto this subspace.

To validate their theoretical results, the authors conduct experiments on synthetic regression tasks with varying Lipschitzness, noise levels, and input dimensionality. Empirically, the authors demonstrate that softmax attention learns appropriate attention window scales across a range of nonlinear function classes, including ReLU networks and trigonometric functions.

**Strengths:**

A first strength is that the paper is not only well written but also excellently presented, which helps in understanding its mathematical content and putting it into a better light. The clarity of exposition is therefore a first great point. In terms of originality, it starts off with the fairly known/commonplace insight that there exists a connection between self-attention and Nadaraya-Watson kernel regression, thus establishing that learning the bandwidth of that estimator across multiple tasks is a necessity for ICL. In this sense and if it stopped there, the contribution wouldn't be particularly novel, as this is intuitive (if thinking of self-attention as learning a summary of the autocovariance function of data and therefore of its characteristic length) and fairly well understood in the literature already (see the cited works of Tosatto et al). One can argue - as in the paper - that Tosatto et al only give an upper bound on the bias, and this work provides a lower bound as well.

However where the paper takes off in my opinion is when these arguments move away from purely the 'frequency cutoff / Lipschitzness' length-related realm, to then move into the *directional*, via using concentration arguments on the hypersphere. This is Theorem 4.4 which formalizes the intuition that ICL in transformers identifies low-dimensional subspaces shared by training tasks, an argument more typically found in the analysis of (non-contrastive) self supervised learning. The mathematical method of proof is elegant as well. The authors derive novel concentration inequalities for functionals of points uniformly distributed on high-dimensional spheres, and use a careful symmetry argument to show that any non-zero component in the orthogonal subspace increases the loss. Overall this represents a standout technical contribution well worthy of publication in my view.

**Weaknesses:**

- A small weakness in presentation is I believe that Theorem 4.4 should be emphasized, as to my knowledge this is the more novel part of the contribution.

- Similarly, the theoretical guarantees are provided for relatively simple function classes (affine and ReLU-based), which may not represent the full range of tasks where ICL is effective.

- The limited scope of experiments is understandable given theoretical assumptions (single-layer) but also important enough that it becomes a weakness, IMHO.

- Finally, the paper could benefit from a more extensive comparison to other theoretical frameworks for understanding ICL.

**Questions:**

Would there be (not extremely involved) ways of moving towards more realistic settings ? i.e. moving to infinite width settings ? Would classification tasks with a cross-entropy loss be somewhat tractable if making the right Gaussian (concentration) assumptions ?

**Limitations:**

There are some obvious, if hard to tackle, limitations in this work:

a. Simplicity of setting: The analysis focuses on single-layer models and synthetic tasks, which may limit its immediate applicability to more complex real-world scenarios.

b. Gap to large language models: While insightful, the work doesn't fully explain emergent ICL in large language models trained on natural data.

---

> ### Author Rebuttal · Authors · 2024-08-06
>
> We thank the reviewer for their valuable feedback. Please find the response below, and a global response above.
>
> 1. **Simple function class.** Please see Point 1 of the global rebuttal.
>
> 2. **Extensions** Please see Point 3 of the global rebuttal.
>
> 3. **Emphasis on Theorem 4.4.** We thank the reviewer for appreciating Theorem 4.4 as a novel, significant and technically impressive contribution. If the paper is accepted, we will allocate some of the additional space provided in the camera-ready version towards further discussing Theorem 4.4's importance in the introduction.
>
> 4. **Comparison to other frameworks.**
> We will add a section in the appendix dedicated to induction heads and the framework comparing ICL to gradient based "mesa" optimizers. Please see Appendix A for an extended discussion of related works. We will refine this discussion and include it in the main body given the additional space if accepted.

---

> > ### Comment · Reviewer_hHKc · 2024-08-14
> >
> > Thanks for your reply and rebuttal. I will maintain my score towards acceptance. Nice work !

---

### Official Review · Reviewer_3LPG · 2024-07-17

**Soundness:** 2
**Presentation:** 3
**Contribution:** 2
**Rating:** 6
**Confidence:** 3

**Summary:**

The paper studies in-context learning (ICL) of one-layer attention-only transformers in a regression task. The paper argues that the product of query and key projection matrix is associated with the Liptchitzness of input data. The notion of attention windows is introduced based on this. The paper shows that attention windows will adapt to the Lipchitness and the noise level of data and perform dimensional reduction. Experiments are conducted to support theoretical claims.

**Strengths:**

The paper provides a rigorous approach to the problem both with proof and empirical simulations.

**Weaknesses:**

- Although explaining ICL is important, the setting seems to be less realistic (see question 1).
- The approach of understanding ICL through the lens of function Lipschitzness is somewhat limiting. For example, Figure 5 demonstrates a decline in generalization error, which is primarily attributed to the function's structure. While the Lipschitz property of a function is indeed a valuable aspect in studying ICL, it doesn't provide a comprehensive explanation or solution for ICL as a whole.

**Questions:**

- The assumption in Equation 4 can be understood as the optimal weights of values $W_V$ will ignore covariates $x$ while the attention will only consider covariates to decide where to attend in the sequence while disregarding predictor $f$. I wonder if this makes sense for hidden states of LLMs. In particular, there is no separation between covariates and predictors in LLMS.  Also, why is the optimal $W_V$ in the proof of Lemma B.1 assumed to have first columns to be zeros (in between Line 577-578)?


- I agree that $x^\top M y$ can be understood as the distance between $x$ and $y$. It might not straightforward to change $x^\top M y$ into $||x - y||$ like used in Appendix. Can you clarify this more on how to obtain Equation (8) in Appendix from Equation (ICL) in the main text.

**Limitations:**

Please see above.

---

> ### Author Rebuttal · Authors · 2024-08-07
>
> We thank the reviewer for their valuable feedback. Please find the response below, and a global response above.
>
> 1. **Closeness of the setting to reality.**
> The data model we consider in the paper is widely studied (Ahn et al. 2023, Akyürek et al. 2023, Mahankali et al. 2023, Garg et al. 2022, von Oswald et al. 2023, etc.) as a setting that demonstrates a form of in-context learning. In that sense, our work brings a common framework for understanding ICL closer to practice using nonlinear function classes and softmax activations.
> We address the reviewer's specific points related to the closeness of the setting we study to reality below.
>
>
>     * **Assumption that the optimal $\mathbf{W}_V$ ignores the covariates.** At a high level, it makes sense that the optimal $\mathbf{W}_V$ ignores the covariates because the value embeddings should live in label space rather than input (covariate) space. For example, suppose each task $t$ entails translating a word in Language $A_t$ to Language $B_t$. A natural prediction of the translation of the query word is some mixture of the example translated words in Language $B_t$, weighted by the closeness of the corresponding word in Language $A_t$ to the query. Languages $A_t$ and $B_t$ might be very different, so the prediction should not include any mixture of the example words in Language $A_t$ (as the prediction must be in Language $B_t$).
>
>         Returning to our setting, we prove that the optimal $\mathbf{W}_V$ leads to a prediction that ignores the covariates; this is not an assumption. First, the estimator does not depend on the first $d$ *rows* of $\mathbf{W}_V$ (not *columns*, as we mistakenly said in line 135), since these only affect the top $d$ entries of the output token which does not affect the loss, which is the standard loss studied by e.g. Ahn et al. 2023, Zhang et al. 2023a, and Mahankali et al. 2023.
>         Second, and more importantly, we prove in  Lemma B.1 that the optimal value of the first $d$ elements in the $(d+1)$-th row of $\mathbf{W}_V$ is zero. Concretely, suppose $\textbf{W}_V = \begin{bmatrix} \mathbf{A}&\mathbf{b} \\\\ \mathbf{v}^\top & c \end{bmatrix}$ and let the attention weight on the $i$-th token be $\beta_i$ (as in the paper). Then the estimator, which is the $(d+1)$-st coordinate of the output token, is $\sum_i \left(cf(\mathbf{x}_i)\beta_i + \mathbf{v}^\top \mathbf{x}_i\right)$. As mentioned above, this does not depend on $\mathbf{A}$ or $\mathbf{b}$ (the first $d$ rows of $\mathbf{W}_V$ only affect the first $d$ rows of the output), and we show in  Lemma B.1 that the optimal value of $\mathbf{v}$ is $\mathbf{0}$. Please see the proof of Lemma B.1 for additional technical details.
>
>     * **Assumption that the optimal  attention parameters $\mathbf{W}_K^\top \mathbf{W}_Q$ ignore the predictor $f$.** Again using the translation task example, it makes sense that the attention weights should only depend on the similarities of the query word in Language $A_t$ with the example inputs in Language $A_t$, since the query lives in a different space than the example labels (which are in Language $B_t$) and thus cannot be compared with them. In the model we use, we follow prior works by setting the last rows of $\mathbf{W}_K$ and $\mathbf{W}_Q$ to zero; for example please see Equation (2) in Ahn et al. 2023 and Equation (11) in Mahankali et al. 2023.
>
>
> * **Clarification on how to derive $\Vert \mathbf{x}-\mathbf{y}\Vert$ from $\mathbf{x}^\top \mathbf{My}$.** We would  like to clarify that we show $e^{\mathbf{x}^\top \mathbf{My}}$ simplifies to $e^{-w||\mathbf{x}-\mathbf{y}||^2}$. To do this, we show all optimal $\mathbf{M}$ must be of the form $\mathbf{M}=w\mathbf{I}_d$. Then, we complete the square, and use the fact that all covariates are on the unit sphere to cancel the squared terms in both numerator and denominator, leaving us only with exponents raised to the cross terms. Please see lines 593, 594, Equation (7), and Lemma B.5.
>
> * **Understanding in-context learning via Lipschitzness.** In the model that we consider, we show how a transformer can exploit a shared Lipschitzness in pretraining for ICL. Figure 5 shows that this is necessary and sufficient for generalization: the attention layer fails to generalize to tasks that would be considered as sharing a common structure to those it sees in pretraining. A model that is pretrained on Affine, ReLU and Cosine functions performs equally well for inference on any particular one of those tasks despite not sharing any common structure other than lipshitzness (Right), meanwhile, pretraining on tasks with a different lipshitzness leads to poor test performance. While this is one important aspect, we agree that there could be other aspects of the data and training that contribute to ICL. For instance, in Section 4 we show that attention can adapt to a ``low rank" structure, essentially picking out specific directions along which to consider distance.

---

> > ### Comment · Reviewer_3LPG · 2024-08-13
> >
> > Thank you for clarification. I have raised the score accordingly.

---

### Author Rebuttal · Authors · 2024-08-06

We would like to thank all reviewers for their thoughtful and detailed feedback. Some reviewers raised common concerns and questions; we will address these next here.

1. **Restricted function class (Reviewers BJxR, ftLf).** We expect that our  results will hold for quite  general function classes, in particular,  those that satisfy Assumption B.4. To instantiate specific bounds in the paper we work with specific function classes, which nevertheless comprises the main technical difficulty. Specifically, we have Lemmas C.3 and C.7 that show that our classes satisfy Assumption B.4. The upper bounds are shown in Lemmas C.5 and C.9. It is necessary that the bias of the estimator should increase with an increase in the size of the attention window (to counteract the decrease in variance), and we establish this for these two function classes specifically in Lemmas C.5 and C.9. The lemmas in Appendix C relate the correlation between the function values of a sample of neighbouring points and the bias of the estimator that is built from such a random sample of neighbours, and to derive these correlation we need to work with a specific class.

2. **Interpreting ``closeness" (Reviewers BJxR, ftLf).** Our paper is a study of what a single layer of attention does; a formal exploration of the notion of attending to "close" points similar to a Nadaraya-Watson (NW) kernel (Nadaraya 1964) (which is a consequence of the particular model we study, but also an intuition that many practitioners carry). We hypothesize that with more layers the hidden token embeddings at intermediate layers reflect some learned metric that this type of estimator can exploit for a notion of closeness beyond closeness of the input tokens. The groundwork for this is laid in Section 4 which shows that in the presence of a low rank structure, the kernel projects out the invariant components of the tokens. Suppose for instance that there are two layers. Then the first layer could learn to project onto the relevant subspace by placing all of its attention on tokens in that subspace, and for the second layer, distances are computed only within this subspace.

3. **How can our results be extended to more realistic settings (Reviewer hHKc) and/or settings that capture emergent properties of transformers as they scale (Reviewer BJxR)? (cc. Reviewer 3PLG)**
The regression model for in-context learning that we consider has been studied extensively (Garg et al. 2022, Akyurek et al. 2023, Ahn et al. 2023,  Zhang et al. 2023a, von Oswald et al. 2023a, Raventos et al. 2023, Fu et al. 2023, etc.) as a setting that demonstrates the in-context learning capabilities of transformer models. Our work brings this framework closer to practice due to its consideration of softmax-activated attention and nonlinear and low-dimensional target functions. Below are possible avenues to extend our results closer to practice (though out of scope of the current paper).

    * *Classification.* The nature of the attention estimate will not change based on the loss function, so it will again be a nearest neighbors estimate in the classification setting. Thus, we expect that the optimal attention weights will again scale with the label noise level and inversely with the Lipschitzness of the labels, for an analogous notion of Lipschitzness defined for the classification setting (e.g. inverse of minimum KL divergence between cluster distributions, or using Fourier coefficients for functions on the hypercube). Technical details need to be resolved regarding analyzing the cross entropy loss, but these should not be prohibitive.
    * *More layers.* Additional layers introduce the following difficulty: the covariates themselves change across layers, so a deeper transformer does not simply implement a kernel estimator with a different kernel. We believe the post-attention multi-layer perceptron (MLP) block and/or layer normalization has to be incorporated into the model and analysis to prevent the tokens from collapsing to a single point similar to what happens in a power iteration (a version of this phenomenon is noted in (Geshkovski et al. 2024)).
    With that being said, it is conceivable that the first $l-1$ layers of an $l$-layer transformer can be interpreted as mapping the input tokens into an appropriate space in which to do nearest-neighbors regression in the $l$-th layer. This is an interesting direction to explore in future work.

    While plausible, these extensions present significant technical challenges that we believe can be topics for separate works. Regarding *infinite width* (suggested by Reviewer hHKc), increasing the inner dimension of $\mathbf{M}_K^\top \mathbf{M}_Q$ will not change the optimal value of $\mathbf{M}=\mathbf{M}_K^\top \mathbf{M}_Q$, so our results will not change.

**References**

Geshkovski, Borjan, et al. "The emergence of clusters in self-attention dynamics." Advances in Neural Information Processing Systems 36 (2024).

---

### Decision · Program_Chairs · 2024-09-25

**Decision:**

Accept (spotlight)

**Comment:**

The paper analyzes how softmax attention enables in-context learning during pre-training, focusing on regression tasks. The authors show that, during pre-training, softmax attention learns a nearest-neighbor predictor tailored to the pre-training task distribution: the attention window adapts based on the Lipschitzness and the label noise of the pre-training tasks. They demonstrate that attention learns to balance the trade-off between bias and variance, and conditions generalization.

This is primarily a theoretical paper that focuses on a simplified yet widely adopted framework for analyzing in-context learning (ICL). All reviewers agree on the originality and relevance of the contribution to a timely topic. They consider the developed theoretical analysis to provide significant and interesting insights into the understanding of how ICL works, and they all express their appreciation for the contribution.